# DR-Submodular Maximization with Stochastic Biased Gradients: Classical and Quantum Gradient Algorithms

**Shengminjie Chen**[1,2]**, Xiaoming Sun**[1,2,*]**Wenguo Yang**[3]**, Jialin Zhang**[4,2]**, Zihan Zhao**[1,2]

1. State Key Lab of Processors, Institute of Computing Technology, Chinese Academy of Sciences
2. School of Computer Science and Technology, University of Chinese Academy of Sciences
3. School of Mathematical Sciences, University of Chinese Academy of Science
4. Institute of Computing Technology, Chinese Academy of Sciences

## Abstract

In this work, we investigate DR-submodular maximization using stochastic biased gradients, which is a more realistic but challenging setting than stochastic unbiased gradients. We first generalize the Lyapunov framework to incorporate biased stochastic gradients, characterizing the adverse impacts of bias and noise. Leveraging this framework, we consider not only conventional constraints but also a novel constraint class: convex sets with a largest element, which naturally arises in applications such as resource allocations. For this constraint, we propose an $1/e$ approximation algorithm for non-monotone DR-submodular maximization, surpassing the hardness result $1/4$ for general convex constraints. As a direct application of stochastic biased gradients, we consider zero-order DR-submodular maximization and introduce both classical and quantum gradient estimation algorithms. In each constraint we consider, while retaining the same approximation ratio, the iteration complexity of our classical zero-order algorithms is $O(\epsilon^{-3})$, matching that of stochastic unbiased gradients; our quantum zero-order algorithms reach $O(\epsilon^{-1})$ iteration complexity, on par with classical first-order algorithms, demonstrating quantum acceleration and validated in numerical experiments.

## 1 Introduction

Diminishing Returns submodular (DR-submodular) maximization has garnered significant interest in theoretical computer science and operations research. While early studies predominantly addressed discrete settings, the emergence of continuous methods has greatly advanced algorithm design for such problems. Consequently, continuous DR-submodular maximization has become an active research area, particularly given its broad applications in machine learning (Bian et al., 2019), social network (Chen et al., 2020; 2021; 2023b), non-convex/non-concave quadratic programming (Staib & Jegelka, 2017), and among others.

Fundamental properties of DR-submodular functions were initially studied through ultramodular functions (Marinacci & Montrucchio, 2005): if $F$ is ultramodular, then $-F$ is DR-submodular. Bian et al. (2017b) directly characterized similar properties for DR-submodular functions in continuous domains. While minimizing continuous submodular functions can be done in polynomial time (Bach, 2019), DR-submodular maximization is NP-hard. For monotone cases, approximation algorithms include Frank-Wolfe variants achieving $(1-e^{-1})$-approximation subject to down-closed convex constraints (Bian et al., 2017b) that also can be generalized to convex constraints and projected gradient ascent with $1/2$-approximation over general convex sets (Hassani et al., 2017). For non-monotone functions, various Frank-Wolfe variant methods have been proposed, offering approximations $1/4$ with general convex constraints (Du, 2022) that is also tight (Mualem & Feldman, 2023) and $1/e$ with down-closed convex constraints (Bian et al., 2017a). In addition, the Lyapunov framework in (Du, 2022) unifies the design and analysis of aforementioned algorithms. Further improvements for down-closed convex constraints achieve ratios up to 0.385 (Chen et al., 2023a),

---

*Corresponding Author: Xiaoming Sun, sunxiaoming@ict.ac.cn

Table 1: DR-submodular Maximization with Stochastic Biased Gradient Estimation

| Funs/Cons | Approx Ratio | Hardness | Iteration Complexity | | | |
|---|---|---|---|---|---|---|
| | | | Quan 0-order | Class 0-order | Class, Det 1-order | Class, Ran 1-order |
| Monotone/ Convex Set | $1 - e^{-1}$ | $1 - e^{-1}$ (Vondrák, 2009) | $O(\epsilon^{-1})$ Thm. 3 | $O(\epsilon^{-3})$ Thm. 5 | $O(\epsilon^{-1})$ (Bian et al., 2017b) | $O(\epsilon^{-2})$ (Zhang et al., 2022) |
| Non-Monotone/ Convex Set | $\frac{\|\mathbf{1}-\mathbf{x}(0)\|_\infty}{4}$ | $\frac{\|\mathbf{1}-\mathbf{x}(0)\|_\infty}{4}$ (Mualem & Feldman, 2023) | | | $O(\epsilon^{-1})$ (Du, 2022) | $O(\epsilon^{-3})$ (Pedramfar et al., 2023) |
| Non-Monotone / Down-Closed Convex Set | 0.385 | 0.478 (Qi, 2022) | | | $O(\epsilon^{-1})$ (Chen et al., 2023a) | $O(\epsilon^{-3})$ Thm. 5 |
| Non-Monotone / Convex Set with a Largest Element | $\frac{\|\mathbf{1}-\mathbf{x}(0)\|_\infty}{e}$ | Unknown | | | $O(\epsilon^{-1})$ Thm. 1 | $O(\epsilon^{-3})$ Thm. 5 |

0.401 (Buchbinder & Feldman, 2024), which are still a gap for the hardness results 0.478 (Qi, 2022). For box constraints, a tight 1/2-approximation is proposed by Niazadeh et al. (2020). While these constraints cover numerous application problems, there exist some application problems with other constraint properties, such as containing a largest element. For example, in some limited resource allocation, e.g., wireless network resource allocation, our goal is to allocate resources to users to maximize the utility function of resource allocation while satisfying the lower-bound constraints on the resources each user receives, e.g., $\mathcal{C} = \{\mathbf{x} \in [0,1]^d | \mathbf{Ax} \geq \mathbf{b}\}$, $\mathbf{A} \in \mathbb{R}_{\geq 0}^{m \times d}$, $\mathbf{b} \in \mathbb{R}_{\geq 0}^m$. Whether algorithms with higher approximation ratios exist under such constraints remains an open question.

However, the aforementioned approximation algorithms are based on the exact gradient oracle. In practical scenarios such as influence maximization and resource allocation in wireless networks, obtaining exact gradients is often challenging due to inherent randomness. To address this challenge, gradient estimation via sampling has emerged as a standard approach, driving the development of approximation algorithms for DR-submodular maximization problems with stochastic unbiased gradients. For stochastic gradient ascents, Hassani et al. (2017) proposed an 1/2 approximation with $O(\epsilon^{-2})$ iteration complexity for monotone DR-submodular functions under convex constraints. This was improved by Zhang et al. (2022) to an $(1 - e^{-1})$ approximation by incorporating boosting and non-oblivious functions. For non-monotone functions, a modified non-oblivious function enables an 1/4 approximation for stochastic projected gradient ascents (Zhang et al., 2024). Mokhtari et al. (2020) proposed a stochastic continuous greedy with the momentum technique to reduce variances, yielding $(1 - e^{-1})$ and $e^{-1}$ approximation with $O(\epsilon^{-3})$ iteration complexity for monotone cases and non-monotone cases, respectively. Under high-order smooth assumptions, Hassani et al. (2020) retained the $(1 - e^{-1})$ approximation ratio with less iteration complexity $O(\epsilon^{-2})$. Further extensions include non-monotone functions under down-closed constraints (Pedramfar et al., 2023) and variance-reduced methods (Lian et al., 2024a). On the other hand, Hazan et al. (2016) introduced a smoothing method $\hat{F}(\mathbf{x}) = \mathbb{E}_{\mathbf{u} \sim \mathbb{B}}[F(\mathbf{x} + r\mathbf{u})]$, which provides an unbiased gradient estimation for $\nabla \hat{F}(\mathbf{x})$ but yields a biased estimation for $\nabla F(\mathbf{x})$, i.e., $\|\nabla \hat{F}(\mathbf{x}) - \nabla F(\mathbf{x})\|_2 \neq 0$. When applying approximation algorithms with stochastic unbiased gradients, the effective objective becomes $\hat{F}(\mathbf{x})$, whose deviation from $F(\mathbf{x})$ can be quantified. Subsequently, this smooth technique is applied to zero-sum games with submodular functions (Wilder, 2018), zero-order stochastic DR-submodular maximization (Pedramfar et al., 2023), and zero-order optimization for the finite-sum of DR-submodular functions (Lian et al., 2024b). In addition, (Wan et al., 2023) also adopted the ellipsoid gradient estimator for DR-submodular maximization. However, the transformation of the objective function $F$ to $\hat{F}$ would be unnecessary if an analysis method tailored to stochastic biased gradients existed.

Given the widespread use of gradient estimators in real-world scenarios, often due to privacy, sampling constraints, robust, or value oracle only, gradient estimates typically exhibit bias. While the smoothing technique introduced in (Hazan et al., 2016) provides an alternative approach to handle stochastic biased gradients via the smoothing function $\hat{F}(\mathbf{x})$, a direct theoretical analysis is of significant theoretical and practical value. Although previous works (Ajalloeian & Stich, 2020; Driggs et al., 2022) have examined stochastic gradient descent under biased gradients in convex optimization settings, yielding important insights (Bhaskara et al., 2025; Augustino et al., 2025), the directly systematic investigation of DR-submodular maximization with stochastically biased gradients remains an open challenge.

**Contribution.** In this work, we investigate the DR-submodular maximization problem with stochastic biased gradients subject to several constraints. The main contributions are as follows:

- Extend the Lyapunov framework for algorithm design on DR-submodular maximization to stochastic biased gradients, characterizing adverse impacts of bias and variance.
- Propose an $1/e$ approximation algorithm for non-monotone DR-submodular maximization subject to convex sets with a largest element, surpassing the previous hardness result $1/4$ under general convex constraints by incorporating properties for the largest element.
- Illustrate quantum zero-order algorithms achieve the same approximation ratio and convergence rate as classical first-order algorithms, exhibiting substantial quantum acceleration.

**Organization.** Section 2 is problem settings and some important inequalities for algorithm design. In Section 3, we extend the Lyapunov framework from exact gradients to stochastic biased gradients and propose some approximation algorithms for DR-submodular maximization under several constraints. As a direct application, we consider zero-order algorithms with quantum and classical gradient estimations in Section 4. Section 5 is numerical experiments. Conclusion is in Section 6.

## 2 PRELIMINARY

In this section, we first explicitly define the problem setting, the gradient setting, and introduce some inequalities of DR-submodular functions for approximation algorithms design. Detailed proofs in this section can be referred in Appendix.A.

**DR-submodular Functions.** The function $F$ is defined over $[0,1]^d$. If $F$ is DR-submodular, for $\forall \mathbf{x} \leq \mathbf{y} \in [0,1]^d$, $\forall k \in \mathbb{R}_{\geq 0}$ and $\forall i \in [d]$ such that $\mathbf{x} + k\mathbf{1}_i, \mathbf{y} + k\mathbf{1}_i \in [0,1]^d$, the inequality holds $F(\mathbf{x} + k\mathbf{1}_i) - F(\mathbf{x}) \geq F(\mathbf{y} + k\mathbf{1}_i) - F(\mathbf{y})$. If $F$ is twice-differentiable, then, for $\forall i, j \in [d]$, $\frac{\partial^2 F(\mathbf{x})}{\partial x_i \partial x_j} \leq 0$. For sake of the following description, there are explanations of some notations: $\theta(t) = \|\mathbf{x}(t)\|_{\infty} = \max_i |x_i(t)|$, $D = \max_{\mathbf{x}, \mathbf{y} \in \mathcal{C}} \|\mathbf{x} - \mathbf{y}\|_2^2$, $\mathbf{x} \vee \mathbf{y} = (\max\{x_i, y_i\})_{i \in [d]}$ and $\mathbf{x} \wedge \mathbf{y} = (\min\{x_i, y_i\})_{i \in [d]}$, the optimal solution $\mathbf{x}^*$ for DR-submodular maximization. Here, we illustrate the inequality used for algorithm design under convex constraints with a largest element. Other inequalities related to algorithm design for general convex constraints and down-closed convex constraints are provided in Appendix.A.

Given that all entries of the Hessian matrix are non-positive, the function exhibits concavity along nonnegative directions, which implies the following upper bound for the optimal solution.

**Lemma 1.** *If $F$ is a non-monotone DR-submodular function, then*

$$(1 - \theta(t))F(\mathbf{x}^*) \leq F(\mathbf{x}^* \vee \mathbf{x}(t)) \leq F(\mathbf{x}(t)) + \langle \nabla F(\mathbf{x}(t)), \mathbf{x}(t) \vee \mathbf{x}^* - \mathbf{x}(t) \rangle \qquad (1)$$

**Problem Setting.** Our objective in this work is to identify a feasible solution within a given constraint set that maximizes a DR-submodular function. The formal statement is as follows:

$$\max_{\mathbf{x} \in \mathcal{C} \subseteq [0,1]^d} F(\mathbf{x}) \qquad (2)$$

where $F$ denotes a DR-submodular function that is $L_0$-continuous and $L_1$-smooth and $\mathcal{C}$ is the feasible set. We mainly focus on the following constraints: (1) $\mathcal{C}$ is a general convex set, i.e., $\forall \mathbf{x}, \mathbf{y} \in \mathcal{C}$ and $\forall \lambda \in [0,1]$, $\lambda \mathbf{x} + (1 - \lambda)\mathbf{y} \in \mathcal{C}$; (2) $\mathcal{C}$ is a down-closed and convex set, i.e., $\mathcal{C}$ is convex and if $\mathbf{y} \in \mathcal{C}$, then $\mathbf{x} \leq \mathbf{y}$ implies $\mathbf{x} \in \mathcal{C}$; (3) $\mathcal{C}$ is a convex set with a largest element, i.e., $\mathcal{C}$ is convex and there is a $\mathbf{x}_{large} \in \mathcal{C}$ such that $\forall \mathbf{x} \in \mathcal{C}$, $\mathbf{x} \leq \mathbf{x}_{large}$.

**Gradient Setting.** The algorithm in this work is built upon a stochastic biased gradient oracle, representing a more practically motivated yet challenging setting. Similar settings have already been considered in convex optimization. The formal statement is as follows:

$$\mathbf{g}(\mathbf{x}(t)) = \nabla F(\mathbf{x}(t)) + \mathbf{b}(\mathbf{x}(t)) + \mathbf{n}(\mathbf{x}(t), \xi) \tag{3}$$

where $\mathbf{b}(\mathbf{x}(t))$ is the bias and $\mathbf{n}(\mathbf{x}(t), \xi)$ represents the noise following a prescribed distribution, satisfying $\mathbb{E}_\xi[\mathbf{n}(\mathbf{x}(t), \xi)] = 0$ and $\mathbb{V}_\xi[\mathbf{n}(\mathbf{x}(t), \xi)] \neq 0$, i.e., $\mathbb{E}[\mathbf{g}(\mathbf{x}(t))|\mathbf{x}(t)] = \nabla F(\mathbf{x}(t)) + \mathbf{b}(\mathbf{x}(t))$. For notational convenience, the dependence on $\xi$ will be omitted in subsequent descriptions and $\mathbb{E}_t[\mathbf{g}(\mathbf{x}(t))] = \mathbb{E}[\mathbf{g}(\mathbf{x}(t))|\mathbf{x}(t)]$. Because boundless bias and noise will cause the algorithm to fail to converge, in our settings, we assume the bias and the noise is bounded as follows:

**Assumption 1** $((m, \eta_b)$-bounded bias)**.**

$$\|\mathbf{b}(\mathbf{x}(t))\|_2 \leq m \|\nabla F(\mathbf{x}(t))\|_2 + \eta_b(t) \tag{4}$$

**Assumption 2** $((M, \eta_n)$-bounded noise)**.**

$$\mathbb{E}[\|\mathbf{n}(\mathbf{x}(t))\|_2] \leq M \|\nabla F(\mathbf{x}(t))\|_2 + \eta_n(t) \tag{5}$$

Therefore, in the algorithm design, the total variance between stochastic biased gradients and exact gradients can be bounded as follows. For notational convenience, we denote the total variance as $A(\mathbf{x}(t), t)$.

$$\begin{aligned} \mathbb{E} \|\mathbf{g}(\mathbf{x}(t)) - \nabla F(\mathbf{x}(t))\|_2 = \mathbb{E} \|\mathbf{n}(\mathbf{x}(t)) + \mathbf{b}(\mathbf{x}(t))\|_2 &\leq \mathbb{E} \|\mathbf{n}(\mathbf{x}(t))\|_2 + \mathbb{E} \|\mathbf{b}(\mathbf{x}(t))\|_2 \\ &\leq (m + M) \|\nabla F(\mathbf{x}(t))\|_2 + \eta_b(t) + \eta_n(t) \triangleq A(\mathbf{x}(t), t) \end{aligned} \tag{6}$$

## 3 LYAPUNOV-BASED FRAMEWORK WITH STOCHASTIC BIASED GRADIENTS

In this section, we generalize the Lyapunov framework in DR-submodular maximization within exact gradients to stochastic biased gradients and illustrate implementation details of our proposed algorithms. For additional technical elaborations in Lyapunov framework and omitted proofs for the theoretical guarantee of algorithms, please refer to Appendix B.

### 3.1 LYAPUNOV-BASED FRAMEWORK WITH STOCHASTIC BIASED GRADIENTS

The primary objective of proposed algorithms is to commence from an initial solution and converge to a solution endowed with theoretical guarantees upon termination. Considering an iteration algorithm with the initial solution $\mathbf{x}(0)$, the algorithm moves the current solution along a direction with a certain step length in each iteration, i.e., $\mathbf{x}(t_{j+1}) - \mathbf{x}(t_j) = \alpha(t_j)\mathbf{v}(t_j)$. Potentially, this difference equation can be viewed as a numerical discretization for a continuous dynamic system $\dot{\mathbf{x}}(t) = \nabla \mathbf{x}(t) = \tilde{\mathbf{v}}(\mathbf{x}(t))$. This process can be conceptualized as identifying a trajectory originating from the initial solution, such that the final solution achieves theoretical guarantee performances while the trajectory length or computational complexity is as small as possible. Considering the algorithm based on the stochastic biased gradient oracle, our algorithm can be regarded as the following dynamic system:

$$\dot{\mathbf{x}}(t) = \nabla \mathbf{x}(t) = \mathbf{f}(\mathbf{x}(t), \mathbf{g}(\mathbf{x}(t)))$$

where $\mathbf{g}(\mathbf{x}(t))$ is the stochastic biased gradient. In DR-submodular maximization, the primary objective in approximation algorithm design is to maximize the expected approximation ratio, because the gradient is stochastic and biased. Consequently, we construct the following variational problem:

$$\begin{aligned} \max \quad & \frac{\mathbb{E}[F(\mathbf{x}(T))]}{F(\mathbf{x}^*)} \\ \text{s.t.} \quad & \dot{\mathbf{x}}(t) = \mathbf{f}(\mathbf{x}(t), \mathbf{g}(\mathbf{x}(t))) \\ & \mathbf{x}(T) \in \mathcal{C}, \mathbf{x}(0) = \mathbf{x}_{initial} \end{aligned} \tag{7}$$

where $T$ denotes the terminal time, $\mathbf{x}(0)$ is the initial solution, and $\mathbf{x}^*$ is the optimal solution for Eq.(2). The objective function in the above variational problem corresponds to the expected approximation ratio, defined as $\frac{\mathbb{E}[F(\mathbf{x}(T))]}{F(\mathbf{x}^*)}$. Solving such fractional optimization problems is challenging, as the optimal solution is generally unknown in advance. While an upper bound of the optimal solution can be derived based on the properties of the objective function and constraints and such a

bound typically relies only on information from the current solution, substituting this upper bound for the optimal solution remains the fractional problem. It remains intractable to solve directly. To address this challenge, we leverage the inherent connection between ratio optimization and difference optimization being generally more tractable and construct a Lyapunov function to streamline the analysis. In addition, we outline a systematic procedure to derive a feasible solution for the above variational problem via this Lyapunov function.

**Definition 1** (Lyapunov Function). *The Lyapunov function associated with a continuous-time algorithm $\mathbf{x}(t)$ for problem Eq.(2) with stochastic biased gradients is as follows:*

$$E(\mathbf{x}(t)) = a(t)\mathbb{E}[F(\mathbf{x}(t))] - b(t)F(\mathbf{x}^*) \tag{8}$$

*where $a(t)$, $b(t)$ are non-negative and non-decreasing functions depending on time $t$.*

**Assumption 3.** *The upper bound of the optimal solution $F(\mathbf{x}^*)$ in continuous time is as follows:*

$$\begin{aligned}
F(\mathbf{x}^*) \leq &p(\mathbf{x}(t))F(\mathbf{x}(t)) + q(\mathbf{x}(t))\mathbb{E}_t[\langle \mathbf{g}(\mathbf{x}(t)), v(\mathbf{x}(t))\rangle] \\
&+ q(\mathbf{x}(t))\mathbb{E}_t[\|\mathbf{n}(\mathbf{x}(t)) + \mathbf{b}(\mathbf{x}(t))\|_2 \|v(\mathbf{x}(t))\|_2]
\end{aligned} \tag{9}$$

*where $p(\mathbf{x}(t))$, $q(\mathbf{x}(t))$ are non-negative stochastic functions depending on the stochastic function $\mathbf{x}(t)$. The exact form of $p(\mathbf{x}(t)), q(\mathbf{x}(t)), v(\mathbf{x}(t))$ only relies on function properties and constraints, and can be obtained before designing the algorithm.*

Because the function value $F$ is non-negative, if we can ensure that the Lyapunov function is non-decreasing, then we have the approximation ratio for the algorithm $\mathbf{x}(t)$ is as follows:

$$E(\mathbf{x}(T)) \geq E(\mathbf{x}(0)) \Rightarrow \mathbb{E}[F(\mathbf{x}(T))] \geq \frac{b(T) - b(0)}{a(T)}F(\mathbf{x}^*) + \frac{a(0)}{a(T)}F(\mathbf{x}(0)) \geq \frac{b(T) - b(0)}{a(T)}F(\mathbf{x}^*)$$

Considering the derivative of the Lyapunov function and replacing the upper bound of the optimal solution, we can obtain a sufficient condition to ensure that the Lyapunov function is approximately non-decreasing. If the stochastic biased gradient is strengthened to the exact gradient, this approximate non-decreasing property of the Lyapunov function transitions to an exact non-decreasing one. The accumulation of errors arising from this approximate non-decreasing adversely impact the approximation ratio. Consequently, we reconstruct the variational optimization problem as follows:

$$\begin{aligned}
\max \quad &\left\{ \frac{b(T) - b(0)}{a(T)}, \; -\frac{1}{a(T)} \int_0^T 2\dot{b}(t)q(\mathbf{x}(t))\mathbb{E}\left[A(\mathbf{x}(t), t)\right] \|v(\mathbf{x}(t))\|_2 \, dt \right\} \\
\text{s.t.} \quad &\dot{a}(t) - \dot{b}(t) \cdot p(\mathbf{x}(t)) \geq 0 \\
&\dot{\mathbf{x}}(t) = \frac{\dot{b}(t)q(\mathbf{x}(t))}{a(t)}v(\mathbf{x}(t))
\end{aligned} \tag{10}$$

where $p(\mathbf{x}(t)), q(\mathbf{x}(t)), v(\mathbf{x}(t))$ is known in Eq.(9). The functions that need to be optimized are $a(t), b(t)$. The aforementioned procedure is formulated in the continuous time, which essentially serves as an informal statement for approximation algorithms. Next, we introduce the discrete time version. Considering the discrete counterpart of Lyapunov function namely potential function:

**Definition 2** (Potential Function). *The Potential function associated with a discrete-time algorithm $\mathbf{x}(t_j)$ for problem Eq.(2) with stochastic biased gradients is as follows:*

$$P(\mathbf{x}(t_j)) = a(t_j)\mathbb{E}[F(\mathbf{x}(t_j))] - b(t_j)F(\mathbf{x}^*) \tag{11}$$

*where $a(t_j)$, $b(t_j)$ are non-negative and non-decreasing sequences depending on time series $t_j$.*

**Assumption 4.** *The upper bound of the optimal solution $F(\mathbf{x}^*)$ in discrete time is as follows:*

$$\begin{aligned}
F(\mathbf{x}^*) \leq &p(\mathbf{x}(t_j))F(\mathbf{x}(t_j)) + q(\mathbf{x}(t_j))\mathbb{E}_{t_j}[\langle \mathbf{g}(\mathbf{x}(t_j)), v(\mathbf{x}(t_j))\rangle] \\
&- q(\mathbf{x}(t_j))\mathbb{E}_{t_j}[\|\mathbf{b}(\mathbf{x}(t_j)) + \mathbf{n}(\mathbf{x}(t_j))\|_2 \|v(\mathbf{x}(t_j))\|_2]
\end{aligned} \tag{12}$$

*where $a(t_j)$, $b(t_j)$ are non-negative and non-decreasing, $p(\mathbf{x}(t_j))$, $q(\mathbf{x}(t_j))$ are non-negative stochastic functions depending on the stochastic sequence $\mathbf{x}(t_j)$.*

Similarly, if we can ensure that the potential function $P(\mathbf{x}(t_j))$ is non-decreasing, we have:

$$P(\mathbf{x}(t_N)) \geq P(\mathbf{x}(t_0)) \Rightarrow \mathbb{E}[F(\mathbf{x}(t_N))] \geq \frac{b(t_N) - b(t_0)}{a(t_N)} F(\mathbf{x}^*) + \frac{a(0)}{a(t_N)} F(\mathbf{x}(0))$$

where $N$ is the total number of the sequence. Considering the difference between adjacent iterations in algorithm $\mathbf{x}(t_j)$, because the total variance Eq.(6) and the $L_1$-smooth condition cannot be neglected, there is a sufficient condition to ensure the potential function is approximate non-decreasing. The accumulation of such non-decreasing errors exerts adverse impacts on both the approximation ratio and the iteration complexity. Certain variance reduction techniques establish explicit relationships among the total variance, the number of iteration steps, and the step size. These relationships enable the mitigation of variance-induced effects on algorithm performances by increasing the number of iterations. Consequently, we reconstruct the sequence problem as follows:

$$\max \quad \left\{ \frac{b(t_N) - b(t_0)}{a(t_N)}, \frac{-\sum_{j=0}^{N-1} \left[ 2[b(t_{j+1}) - b(t_j)]q(\mathbf{x}(t_j))\mathbb{E}[A(\mathbf{x}(t_j), t_j)] \|v(\mathbf{x}(t_j))\|_2 \right.}{a(t_N)} \right.$$

$$\left. \frac{\frac{L_1}{2}\mathbb{E}[a(t_{j+1}) \|\mathbf{x}(t_{j+1}) - \mathbf{x}(t_j)\|_2^2]\right]}{a(t_N)} \right\} \tag{13}$$

$$\text{s.t. } a(t_{j+1}) - a(t_j) - [b(t_{j+1}) - b(t_j)] \cdot p(\mathbf{x}(t_j)) \geq 0$$

$$\mathbf{x}(t_{j+1}) - \mathbf{x}(t_j) = \frac{[b(t_{j+1}) - b(t_j)]q(\mathbf{x}(t_j))}{a(t_{j+1})} v(\mathbf{x}(t_j))$$

## 3.2 DR-submodular Maximization with Stochastic Biased Gradient

Leveraging inequalities in Section 2 and the Lyapunov framework, we design approximation algorithms tailored to DR-submodular maximization with stochastic biased gradients under several constraints. When gradients are exact or stochastically unbiased, these algorithms achieve theoretical guarantees identical to those established in (Du, 2022; Pedramfar et al., 2023; Chen et al., 2023a) for convex constraints and down-closed convex constraints. As a demonstration, the main text focuses on convex constraints with the largest element, which is newly introduced in this work. Details for DR-submodular maximization with stochastic biased gradients under convex constraints and down-closed convex constraints, please refer to Appendix B.2. Because a tight approximation $(1 - e^{-1})$ for monotone DR-submodular maximization are reached, our analysis of monotone cases is confined to general convex constraints.

**Theorem 1.** *[Informal Statement] For monotone DR-submodular maximization with stochastic biased gradients, there is an $(1 - e^{-1})$ approximation algorithm under general convex sets. When considering non-monotone DR-submodular functions, the approximation results under different constraints are as follows: $\frac{\|\mathbf{1} - \mathbf{x}(0)\|_\infty}{4}$ (general convex sets), $0.385$ (down-closed convex sets), and $\frac{\|\mathbf{1} - \mathbf{x}(0)\|_\infty}{e}$ (convex sets with the largest element). If the gradient becomes an exact gradient, the iteration complexity for aforementioned approximation algorithms is $O(\epsilon^{-1})$.*

Next, we illustrate details for convex constraints with the largest element. This is a part of proof for the above theorem. Considering the upper bound of the optimal solution in Eq.(1), we have:

$$F(\mathbf{x}^*) \leq \frac{F(\mathbf{x}(t)) + \mathbb{E}_t[\max_{\mathbf{v} \in \mathcal{C}} \langle \mathbf{g}(\mathbf{x}(t)), \mathbf{v} - \mathbf{x}(t) \rangle] - \mathbb{E}_t[\|\mathbf{n}(\mathbf{x}(t)) + \mathbf{b}(\mathbf{x}(t))\|_2]\sqrt{D}}{1 - \theta(t)}$$

where the inequality comes from Eq.(1) and $\mathbf{x} \vee \mathbf{x}^* \in \mathcal{C}$. The above inequality implies the explicit formula for $p(\mathbf{x}(t)), q(\mathbf{x}(t)), v(\mathbf{x}(t))$ as follows:

$$p(\mathbf{x}(t)) = q(\mathbf{x}(t)) = \frac{1}{1 - \theta(t)}, \mathbf{v}_{\max}(t) = \arg\max_{\mathbf{v} \in \mathcal{C}}\langle \mathbf{g}(\mathbf{x}(t)), \mathbf{v} \rangle, v(\mathbf{x}(t)) = \mathbf{v}_{\max}(t) - \mathbf{x}(t)$$

Replacing the above parameter functions into Eq.(10)

$$\begin{cases} \dot{a}(t) - \dot{b}(t) \cdot p(\mathbf{x}(t)) = 0 \\ \dot{\mathbf{x}}(t) = \frac{\dot{b}(t)q(\mathbf{x}(t))}{a(t)}v(\mathbf{x}(t)) \end{cases} \Rightarrow \dot{\mathbf{x}}(t) = \frac{\dot{a}(t)}{a(t)}v(\mathbf{x}(t)) \tag{14}$$

Formally, the above procedure is implemented into the continuous time. To demonstrate that the solution is a convex combination of feasible solutions, we use the following limit form.

$$\lim_{\epsilon \to 0} \mathbf{x}(t+\epsilon) - \mathbf{x}(t) = \lim_{\epsilon \to 0} \int_t^{t+\epsilon} \frac{\dot{a}(s)}{a(s)} v(\mathbf{x}(s)) ds = \lim_{\epsilon \to 0} \frac{\dot{a}(t)}{a(t)} [\mathbf{v}_{\max}(t) - \mathbf{x}(t)]\epsilon$$

Because the constraint is convex, the above inequality shows $\mathbf{x}(T)$ is a feasible solution. In addition, due to $\mathbf{v}_{\max}(t) \le \mathbf{1}$, the explicit formula for $\theta(t)$ can be determined as follows:

$$\dot{\mathbf{x}}(t) = \frac{\dot{a}(t)}{a(t)} v(\mathbf{x}(t)) \le \frac{\dot{a}(t)}{a(t)}(\mathbf{1} - \mathbf{x}(t)) \Rightarrow \mathbf{x}(t) \le \mathbf{1} - (\mathbf{1} - \mathbf{x}(0))e^{-\int_0^t \frac{\dot{a}(\tau)}{a(\tau)} d\tau} = \mathbf{1} - (\mathbf{1} - \mathbf{x}(0))\frac{a(0)}{a(t)}$$

Combining Eq.(14) with $p(\mathbf{x}(t)) = \frac{1}{\|\mathbf{1} - \mathbf{x}(0)\|_\infty \frac{a(0)}{a(t)}}$, the approximation ratio is as follows:

$$\|\mathbf{1} - \mathbf{x}(0)\|_\infty \dot{a}(t)\frac{a(0)}{a(t)} = \dot{b}(t) \Rightarrow \frac{b(T) - b(0)}{a(T)} = \|\mathbf{1} - \mathbf{x}(0)\|_\infty \frac{a(0)}{a(T)} \ln \frac{a(T)}{a(0)} \le \frac{\|\mathbf{1} - \mathbf{x}(0)\|_\infty}{e}$$

If we assume $T = 1$, the explicit formula is as follows:

$$a(t) = e^t, b(t) = \|\mathbf{1} - \mathbf{x}(0)\|_\infty t$$

The above analysis is based on continuous times. To adopt the potential function for deriving a discrete time algorithm and leverage the explicit formula for parameter functions in the Lyapunov function, we first claim that parameter functions in the Lyapunov function satisfy constraints in Eq.(13):

$$a(t_{j+1}) - a(t_j) - [b(t_{j+1}) - b(t_j)]p(\mathbf{x}(t_j)) = e^{t_{j+1}} - e^{t_j} - e^{t_j}[t_{j+1} - t_j] \ge 0$$

This implies that we only consider the second objective function in Eq.(13) to derive the complexity of the $\frac{\|\mathbf{1} - \mathbf{x}(0)\|_\infty}{e}$ approximation algorithm. If we set a constant time step $t_j = \frac{j}{N}$, we have:

$$\sum_{j=0}^{N-1} \left[ 2[b(t_{j+1}) - b(t_j)]\mathbb{E}[q(\mathbf{x}(t_j))A(\mathbf{x}(t_j), t_j) \|v(\mathbf{x}(t_j))\|_2] + \frac{L_1}{2}\mathbb{E}[a(t_{j+1}) \|\mathbf{x}(t_{j+1}) - \mathbf{x}(t_j)\|_2^2] \right]$$

$$\le \sum_{j=0}^{N-1} \frac{2e^{\frac{j}{N}}\sqrt{D}}{N} \mathbb{E}[A(\mathbf{x}(t_j), t_j)] + \frac{L_1 D(e-1)}{2N} = O(\epsilon)$$

When the first item is 0, i.e., the stochastic biased gradient is strengthened to the exact gradient, our algorithm yields an $\frac{\|\mathbf{1} - \mathbf{x}(0)\|_\infty}{e}$ approximation algorithm for DR-submodular maximization with the exact gradient subject to convex constraints with the largest element, after $N = O(\frac{1}{\epsilon})$ iterations. Therefore, we obtain an approximation algorithm with the following theoretical guarantees.

$$\begin{aligned}
\textbf{Approximation Ratio:} \quad & \frac{\|\mathbf{1} - \mathbf{x}(0)\|_\infty}{e} - O(\epsilon) \\
\textbf{Algorithm:} \quad & \mathbf{x}(t_{j+1}) = (1 - \tfrac{1}{N}e^{-\frac{1}{N}})\mathbf{x}(t_j) + \tfrac{1}{N}e^{-\frac{1}{N}}\mathbf{v}(t_j) \\
\textbf{Iteration Complexity:} \quad & \sum_{j=0}^{N-1} \frac{2e^{\frac{j}{N}}\sqrt{D}}{N}\mathbb{E}[A(\mathbf{x}(t_j), t_j)] + \frac{L_1 D(e-1)}{2eN} \le O(\epsilon)
\end{aligned}$$

# 4 ZERO-ORDER ALGORITHMS WITH GRADIENT ESTIMATION

As a direct application to stochastic biased gradients, this section introduces several zero-order algorithms based on gradient estimation techniques. First, we adopt a quantum gradient estimation method that integrates the Jordan algorithm with quantum state tomography. Subsequently, we introduce a classical gradient estimation approach that incorporates smoothing techniques and variance reduction methods. Detailed algorithms and proofs in this section can be referred in Appendix C.

## 4.1 QUANTUM ZERO-ORDER ALGORITHM

Quantum Jordan algorithm exhibits polynomial acceleration in problem dimensions compared to classical algorithms for estimating gradients. In this section, we employ an improved Jordan algorithm, proposed by van Apeldoorn et al. (2023) and characterize the variance of this algorithm when estimating stochastic biased gradients using biased function value oracles, which is motivated by the fact that, in finite-qubits, we obtain a $\beta$-approximation function value $F_\beta(\mathbf{x})$ of the exact function $F(\mathbf{x})$ for all $\mathbf{x} \in \mathbb{R}^d$. Several parameters in our setup differ from those in (Augustino et al., 2025).

**Theorem 2.** *Let $F : \mathbb{R}^d \to \mathbb{R}_{\geq 0}$ be a $L_0$-Lipschitz continuous and $L_1$-smooth function, let $\sigma \in \mathbb{R}_+$ with $\frac{\sigma}{L_0} \in (0, 1]$, and let $F_\beta(\mathbf{x})$ be the $\beta$-approximation function for $F(\mathbf{x})$ with $\forall \mathbf{x}$ in $\mathcal{B}_\infty^d(\mathbf{x}, \sqrt{\frac{2\beta}{L_1 d}})$. The gradient estimation $\mathbf{g}(\mathbf{x})$ returned by the quantum algorithm in Appendix C.1 satisfies*

$$\mathbb{E}\left[\|\mathbf{g}(\mathbf{x}) - \nabla F(\mathbf{x})\|_\infty\right] \leq \sigma$$

*when*

$$\beta \leq \frac{\sigma^2}{288\pi L_1 d(8\lceil \ln \frac{36 L_0 d}{\sigma}\rceil + 1)}$$

*The query complexity of the value oracle $F$ is $O(8\lceil \ln \frac{36 L_0 d}{\sigma}\rceil)$.*

For DR-submodular functions and $\forall \mathbf{x}, \mathbf{y} \in [0, 1]^d$, we have:

$$
\begin{aligned}
|F(\mathbf{x}) - F(\mathbf{y})| &= |\langle \nabla F(\mathbf{x} + \epsilon(\mathbf{y} - \mathbf{x})), \mathbf{y} - \mathbf{x}\rangle| \\
&\leq \|\nabla F(\mathbf{x} + \epsilon(\mathbf{y} - \mathbf{x}))\|_2 \|\mathbf{y} - \mathbf{x}\|_2 \leq (\|\nabla F(\mathbf{0})\|_2 + \|\nabla F(\mathbf{1})\|_2) \|\mathbf{y} - \mathbf{x}\|_2
\end{aligned}
$$

The above inequality demonstrates that the DR-submodular function is continuous with the constant $L_0 = \|\nabla F(\mathbf{0})\|_2 + \|\nabla F(\mathbf{1})\|_2$. The first inequality follows from Cauchy-Schwarz inequality and the second is from the property of DR-submodular functions. Combining this inequality with results in Section 3.2, we illustrate that the quantum zero-order algorithm, i.e., $\mathbb{E}[A(\mathbf{x}(t_j), t_j)] \leq \sigma$, can achieve the performance of classical first-order methods in both approximation ratio and complexity.

**Theorem 3.** *For $L_0$-Lipschitz continuous and $L_1$-smooth DR-submodular functions $F$, when we adopt the algorithm in Theorem 2 to estimate the gradient for $F$, quantum zero-order algorithms in Appendix C.1 achieve approximation ratios in Theorem 1 after $O(\epsilon^{-1})$ iteration complexity.*

## 4.2 CLASSICAL ZERO-ORDER ALGORITHM

When only the function value oracle is available for queries, the following smooth technique is widely used in classical algorithms, which is the expectation for sampling some points through a unit ball centered at a give point.

$$\hat{F}(\mathbf{x}) = \mathbb{E}_{\mathbf{u} \sim \mathbb{B}}[F(\mathbf{x} + r\mathbf{u})]$$

where $\mathbb{B} = \{\mathbf{v} \in \mathbb{R}^d | \|\mathbf{v}\|_2 \leq 1\}$. A key advantage of this smooth technique is that an unbiased estimation of the gradient for after smoothing function can be obtained by sampling the original function values through a unit sphere. Since Shamir (2017) demonstrated that two-point estimation yields a smaller variance, in our work, we also adopt the two-point estimation.

$$\nabla \hat{F}(\mathbf{x}) = \frac{d}{r}\mathbb{E}_{\mathbf{u} \sim \mathbb{S}}[F(\mathbf{x} + r\mathbf{u})\mathbf{u}] = \frac{d}{2r}\mathbb{E}_{\mathbf{u} \sim \mathbb{S}}[(F(\mathbf{x} + r\mathbf{u}) - F(\mathbf{x} - r\mathbf{u}))\mathbf{u}]$$

where $\mathbb{S} = \{\mathbf{v} \in \mathbb{R}^d | \|\mathbf{v}\|_2 = 1\}$. Leveraging the unbiased gradient for $\hat{F}(\mathbf{x}(t))$, most existing works convert the optimized objective function into the smooth function $\hat{F}(\mathbf{x})$, whose distance for the original function can be bounded by the $L_0$-continuous constant of the original function. However, when we directly consider the original function, the above gradient estimation is a biased estimation for $\nabla F(\mathbf{x}(t))$. Next, we illustrate both the bias and the noise are bounded as follows:

**Lemma 2.** *If $F$ is $L_1$-smooth, then $\hat{F}$ is $L_1$-smooth and*

$$
\begin{aligned}
\|\mathbf{b}(\mathbf{x}(t))\|_2 &= \left\|\nabla \hat{F}(\mathbf{x}) - \nabla F(\mathbf{x})\right\|_2 \leq \frac{d L_1 r}{2} \\
\mathbb{E}[\|\mathbf{n}(\mathbf{x}(t))\|_2] &= \mathbb{E}\left[\left\|\nabla \hat{F}(\mathbf{x}) - \mathbb{E}[\nabla \hat{F}(\mathbf{x})]\right\|_2\right] \leq \sqrt{16\sqrt{2\pi}d L_0^2}
\end{aligned}
$$

To mitigate the above variance in the algorithm, we adopt the variance reduction technique in Chen et al. (2018); Mokhtari et al. (2020), which introduced a momentum term and based the unbiased stochastic gradient. Next, we prove analogous results for the case of stochastic biased gradients.

**Theorem 4.** *For a $L_1$-smooth function $F$, $\mathbf{g}(\mathbf{x}(t))$ is the gradient of $\mathbf{x}(t)$ returned by a stochastic biased gradient oracle with Assumptions 1 and 2, i.e., $\mathbf{g}(\mathbf{x}(t)) = \nabla F(\mathbf{x}(t)) + \mathbf{b}(\mathbf{x}(t)) + \mathbf{n}(\mathbf{x}(t))$. If $\|\mathbf{x}(t_{j+1}) - \mathbf{x}(t_j)\|_2 \leq \frac{\sqrt{D}}{N}$, the variance of the $\mathbf{d}_{t_j} = (1 - \rho_{t_j})\mathbf{d}_{t_{j-1}} + \rho_t \mathbf{g}(\mathbf{x}(t_j))$ is as follows:*

$$\mathbb{E}[\|\nabla F(\mathbf{x}(t_j)) - \mathbf{d}_{t_j}\|_2^2] \leq \frac{Q(t_j)}{(j + 9)^{2/3}}$$

*where* $Q(t_j) = \max\left\{5L_1^2 D + 16 \max_{k \in [j]} A^2(\mathbf{x}(t_k), t_k) + 2L_1\sqrt{D}\max_{k \in [j]}\|\mathbf{b}(\mathbf{x}(t_k))\|_2 + 16(j+8)^{2/3}\max_{k \in [j]}\|\mathbf{b}(\mathbf{x}(t_k))\|_2^2, \|\nabla F(\mathbf{x}(t_0)) - \mathbf{d}_{t_0}\|_2^2 9^{2/3}\right\}$

Combining the above variance reduction technique and results in Section 3.2, the theoretical guarantee for classical zero-order algorithms with stochastic biased gradients is as follows:

**Theorem 5.** *For $L_0$-Lipschitz continuous and $L_1$-smooth DR-submodular functions $F$, when we adopt the algorithm in Theorem 4 to estimate the gradient for $F$, there are some classical algorithms with the stochastic biased gradient oracle achieving approximation ratios in Theorem 1 after $O(\epsilon^{-3})$ iteration complexity.*

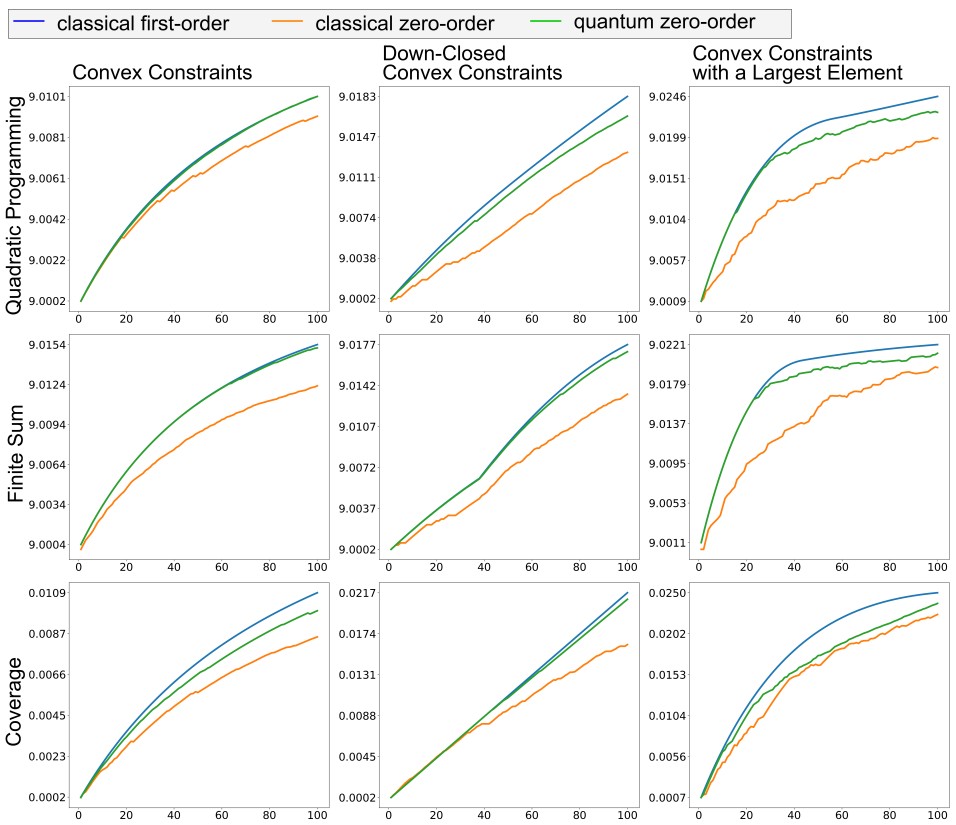

Figure 1: Comparison of classical first-order algorithm(blue), quantum zero-order algorithm(green), and classical zero-order algorithm(orange) with $d = 3$. The x-axis is the number of iterations. The y-axis is the function value.

## 5 NUMERICAL EXPERIMENTS

In this section, we evaluate our algorithms proposed in Section 3.2 and Section 4 on three standard test problems: DR-submodular quadratic programming, finite sums of DR-submodular quadratic functions, and regular coverage functions. The tests consider three constraint settings: convex constraints, down-closed convex constraints, and convex constraints with a largest element. These problems are widely used benchmarks in DR-submodular maximization (Bian et al., 2017b; Chen et al., 2023a; Lian et al., 2024b) with practical applications such as scheduling (Skutella (2001)) and demand forecasting (Ito & Fujimaki (2016)). Due to the high computational cost of classical simulation of quantum algorithms, we only set $d = 3$. For settings of the problem, parameters of the experiment, and more experimental results, please refer to Appendix D.

In Fig.1, under general convex constraints, our quantum zero-order algorithm exhibits faster convergence than the classical zero-order algorithm, while achieving solution quality comparable to that of the classical first-order algorithm in DR-submodular quadratic programming and its finite-sum. For

Table 2: The rough analysis of clock times for phase estimations on quantum gradient algorithms

| Dimension | Errors | Depth | Qubits | Clock Time | Dimension of Vector |
|---|---|---|---|---|---|
| | $\epsilon_{ALG} = 0.1$ | $\approx 203$ | $\approx 130$ | $\approx 20\mu s$ | $\approx 2^{130}$ |
| $d = 10$ | $\epsilon_{ALG} = 0.01$ | $\approx 394$ | $\approx 190$ | $\approx 39\mu s$ | $\approx 2^{190}$ |
| | $\epsilon_{ALG} = 0.001$ | $\approx 543$ | $\approx 230$ | $\approx 54\mu s$ | $\approx 2^{230}$ |
| | $\epsilon_{ALG} = 0.1$ | $\approx 512$ | $\approx 1600$ | $\approx 51\mu s$ | $\approx 2^{1600}$ |
| $d = 100$ | $\epsilon_{ALG} = 0.01$ | $\approx 796$ | $\approx 2300$ | $\approx 79\mu s$ | $\approx 2^{2300}$ |
| | $\epsilon_{ALG} = 0.001$ | $\approx 1002$ | $\approx 2600$ | $\approx 100\mu s$ | $\approx 2^{2600}$ |

down-closed convex constraints, the quantum zero-order algorithm retains faster convergence than the classical zero-order algorithm. It trails the classical first-order algorithm slightly in solution quality for DR-submodular quadratic programming but still outperforms the classical zero-order algorithm. Under convex constraints with the largest element, the quantum zero-order algorithm attains convergence speed close to the classical first-order algorithm, with both outperforming the classical zero-order algorithm in iteration complexity and solution quality for DR-submodular quadratic programming and its finite-sum. Although a degradation in solution quality is observed for regular coverage maximization, the convergence rate still surpasses that of classical zero-order algorithm. Numerical experiments demonstrate quantum accelerations.

To obtain a general yet rough understanding of the clock-time performance for quantum gradient estimation algorithms, we first assume that the time-consuming for single/double qubit gate is approximately 50/100ns, an achievable assumption with the existing quantum technology. Because a quantum algorithm implemented on quantum circuits consists of several single/double qubit gates, and gates at the same depth can be executed in parallel, we can construct a rough relationship between clock times for quantum algorithms and depth for their corresponding quantum circuits. Numerical details are in Tab.2. Thus, the clock time for phase estimation is about $100\mu s$ when the dimension $d = 100$ and the numerical errors $\epsilon_{ALG} = 0.001$. Notably, preparing quantum states required as input for gradient estimation algorithms depends on the depth of the quantum circuit for the value oracle $F$, which varies by case and cannot be uniformly characterized. If we consider the quadratic function, the depth for value oracle $F$ on quantum circuits is roughly $O(d)$. Assuming that the copies of $|\psi\rangle$, required as input for the quantum algorithm in Theorem 2, can be prepared simultaneously across multiple quantum computers, the total clock time for quantum gradients estimation algorithm is about a few hundred $\mu s$ when the dimension $d = 100$ and the numerical errors $\epsilon_{ALG} = 0.001$. With the development of quantum hardware, the time-consuming of this quantum gradient estimation algorithm is expected to be further reduced.

## 6 CONCLUSION

In this work, we study the continuous DR-submodular maximization problem with stochastic biased gradients. First, we extend the Lyapunov framework originally developed for exact gradients to stochastic biased gradients by characterizing the adverse impacts of bias and the noise. These insights are further employed to design approximation algorithms for DR-submodular maximization under three constraints. For a novel constraint, convex sets with a largest element, we propose an $1/e$ approximation algorithm, surpassing the prior $1/4$ hardness result under general convex constraints by leveraging the largest element. In addition, we consider zero-order DR-submodular maximization as a direct application for stochastic biased gradient oracles. Our quantum zero-order method attains the iteration complexity $O(\epsilon^{-1})$, outperforming the classical iteration complexity $O(\epsilon^{-3})$. Notably, it also matches the performance of classical first-order methods in both approximation ratio and complexity, demonstrating substantial quantum acceleration.

ACKNOWLEDGMENTS

This work was supported in part by Quantum Science and Technology-National Science and Technology Major Project under Grant No. 2024ZD0300500 and the National Natural Science Foundation of China Grant Nos. 92465202, 12501450, 12571382, and 62272441.

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

THE USE OF LARGE LANGUAGE MODELS

We acknowledge the assistance of the large language model in polishing the English expression of this article.

## A  THE PROOF FOR SECTION 2

In this section, we prove some important inequalities for DR-submodular functions that are essential ingredients for the design of approximation algorithms. First, we re-state the definition of submodular functions and DR-submodular functions.

**Definition 3.** *The submodular function $F$ defined over $[0,1]^d$, i.e., $F : [0,1]^d \to \mathbb{R}$, satisfies the following inequality:*

$$F(\mathbf{x} \vee \mathbf{y}) + F(\mathbf{x} \wedge \mathbf{y}) \leq F(\mathbf{x}) + F(\mathbf{y}) \quad \forall \mathbf{x}, \mathbf{y} \in [0,1]^d$$

*The above inequality can be rewrite as the weakly diminishing return property. For $\forall \mathbf{x} \leq \mathbf{y} \in [0,1]^d$, $\forall k \in \mathbb{R}_{\geq 0}$ and $\forall i \in [d]$ with $x_i = y_i$ such that $\mathbf{x} + k\mathbf{1}_i, \mathbf{y} + k\mathbf{1}_i \in [0,1]^d$, the inequality holds:*

$$F(\mathbf{x} + k\mathbf{1}_i) - F(\mathbf{x}) \geq F(\mathbf{y} + k\mathbf{1}_i) - F(\mathbf{y}) \quad \forall i \in V = \{j \in [d] | x_j = y_j\}$$

*When $F$ is twice-differentiable, its Hessian matrix possesses the following properties*

$$\frac{\partial^2 F(\mathbf{x})}{\partial x_i \partial x_j} \leq 0 \quad \forall i \neq j, i, j \in [d]$$

**Definition 4.** *The DR-submodular function $F$ defined over $[0,1]^d$, i.e., $F : [0,1]^d \to \mathbb{R}$, for $\forall \mathbf{x} \leq \mathbf{y} \in [0,1]^d$, $\forall k \in \mathbb{R}_{\geq 0}$ and $\forall i \in [d]$ such that $\mathbf{x} + k\mathbf{1}_i, \mathbf{y} + k\mathbf{1}_i \in [0,1]^d$, the inequality holds:*

$$F(\mathbf{x} + k\mathbf{1}_i) - F(\mathbf{x}) \geq F(\mathbf{y} + k\mathbf{1}_i) - F(\mathbf{y})$$

*When $F$ is twice-differentiable, its Hessian matrix possesses the following properties*

$$\frac{\partial^2 F(\mathbf{x})}{\partial x_i \partial x_j} \leq 0 \quad \forall i, j \in [d]$$

Based on the above definitions, it is evident that DR-submodularity is stronger than submodularity. In some scenarios, the submodularity can be equivalently described as the weak DR-submodularity, which retains diminishing returns in dimensions with the same component. Leveraging the diminishing return and the submodularity, the followings illustrate some inequalities for approximation algorithm design.

**Lemma 3.** *If $F$ is a monotone DR-submodular function, then*

$$F(\mathbf{x}^*) \leq F(\mathbf{x}^* \vee \mathbf{x}(t)) \leq F(\mathbf{x}(t)) + \langle \nabla F(\mathbf{x}(t)), \mathbf{x}(t) \vee \mathbf{x}^* - \mathbf{x}(t) \rangle \tag{15}$$

*If $F$ is a non-monotone DR-submodular function, then*

$$(1 - \theta(t))F(\mathbf{x}^*) \leq F(\mathbf{x}^* \vee \mathbf{x}(t)) \leq F(\mathbf{x}(t)) + \langle \nabla F(\mathbf{x}(t)), \mathbf{x}(t) \vee \mathbf{x}^* - \mathbf{x}(t) \rangle \tag{16}$$

*Proof of Lemma 3.* Because all entries of the Hessian matrix are non-positive for DR-submodular functions, for all non-negative vector $\mathbf{v} \in \mathbb{R}_{\geq 0}^d$, we have:

$$\mathbf{v}^T \nabla^2 F(\mathbf{x}) \mathbf{v} \leq 0 \Rightarrow F(\lambda \mathbf{x} + (1 - \lambda)\mathbf{y}) \geq \lambda F(\mathbf{x}) + (1 - \lambda)F(\mathbf{y})$$

where $\mathbf{x} \leq \mathbf{y} \in [0,1]^n$. This inequality implies the concavity along with the non-negative directions for DR-submodular functions. By leveraging this partial concavity, we have:

$$F(\mathbf{x}(t) \vee \mathbf{x}^*) = F(\mathbf{x}(t) + \mathbf{x}(t) \vee \mathbf{x}^* - \mathbf{x}(t)) \leq F(\mathbf{x}(t)) + \langle \nabla F(\mathbf{x}(t)), \mathbf{x}(t) \vee \mathbf{x}^* - \mathbf{x}(t) \rangle$$

where $\mathbf{v}(t) = \mathbf{x}(t) \vee \mathbf{x}^* - \mathbf{x}(t)$ is a non-negative direction, i.e., $F$ is concavity along with $\mathbf{v}(t)$. If $F$ is monotone, then we have:

$$F(\mathbf{x}^*) \leq F(\mathbf{x}(t) \vee \mathbf{x}^*) \leq F(\mathbf{x}(t)) + \langle \nabla F(\mathbf{x}(t)), \mathbf{x}^* \vee \mathbf{x}(t) - \mathbf{x}(t) \rangle$$

If $F$ is non-monotone, then we have:

$$F(\mathbf{x}(t)) + \langle \nabla F(\mathbf{x}(t)), \mathbf{x}^* \vee \mathbf{x}(t) - \mathbf{x}(t) \rangle \geq F(\mathbf{x}^* \vee \mathbf{x}(t))$$

$$= F\Big((1 - \theta(t))\mathbf{x}^* + \theta(t)(\mathbf{x}^* + \frac{\mathbf{x}(t) \vee \mathbf{x}^* - \mathbf{x}^*}{\theta(t)}))\Big)$$

$$\geq (1 - \theta(t))F(\mathbf{x}^*) + \theta(t)F\Big(\mathbf{x}^* + \frac{\mathbf{x}(t) \vee \mathbf{x}^* - \mathbf{x}^*}{\theta(t)}\Big)$$

$$\geq (1 - \theta(t))F(\mathbf{x}^*)$$

where $\theta(t) = \|\mathbf{x}(t)\|_\infty$. If $\theta(t) \leq \|\mathbf{x}(t)\|_\infty$, it will cause $\mathbf{x}^* + \frac{\mathbf{x}(t) \vee \mathbf{x}^* - \mathbf{x}^*}{\theta(t)}$ to be outside the domain $[0, 1]^d$, making the function value undefined. $\qquad\square$

When we have much detailed description for $\mathbf{x}(t)$, i.e., the upper bound for the $\mathbf{x}(t)$ can be divided into a partition $V, U$, i.e., $V \cap U = \emptyset$ and $V \cup U = [d]$, we can obtain another inequality to estimate the upper bound for the optimal solution.

**Lemma 4.** *Denote $\theta_V(t) = \|\mathbf{x}_V(t)\|_\infty = \max_{i \in V} |x_i(t)|$, $\theta_U(t) = \|\mathbf{x}_U(t)\|_\infty = \max_{i \in U} |x_i(t)|$. If $\theta_V(t) \leq \theta_U(t)$, then, for a non-monotone DR-submodular function $F$, we have*

$$u(t)(1 - \theta_V(t))F(\mathbf{x}^*) \leq F(\mathbf{x}(t)) + \Big\langle \nabla F(\mathbf{x}(t)), \mathbf{x}(t) \vee \Big(\mathbf{x}^* \wedge (u(t)\mathbf{1}_U \vee v(t)\mathbf{1}_V)\Big) - \mathbf{x}(t) \Big\rangle$$

$$+ \Big(1 - v(t)\Big)\Big(1 - \theta_V(t)\Big)F(\mathbf{x}^* \wedge \mathbf{1}_V)$$

$$+ \Big(\theta_U(t) - \theta_V(t)\Big)F\Big((\mathbf{x}^* \wedge (u(t)\mathbf{1}_U \vee v(t)\mathbf{1}_V)) \vee \mathbf{1}_V\Big)$$

(17)

*where $u(t), v(t)$ are parameter functions in these two parts $U, V$.*

*Proof of Lemma 4.* For the sake of description, we set $\theta_U = \theta_U(t)$, $\theta_V = \theta_V(t)$ and denote $\mathbf{z} = \mathbf{x}^* \wedge (u(t)\mathbf{1}_U \vee v(t)\mathbf{1}_V)$. Considering the concavity along with the non-negative direction in these two components respectively, we have:

$$F(\mathbf{x}(t) \vee \mathbf{z}) \geq (1 - \theta_U) F(\mathbf{z}) + \theta_U F\left(\mathbf{z} + \frac{1}{\theta_U}(\mathbf{x}(t) \vee \mathbf{z} - \mathbf{z})\right)$$

$$= (1 - \theta_U) F(\mathbf{z}) + \theta_U F(\mathbf{p}_{B_1} \vee \mathbf{p}_{B_2} \vee \mathbf{p}_{B_3} \vee \mathbf{z}_{B_2})$$

$$\geq (1 - \theta_U) F(\mathbf{z}) + \theta_U \frac{\theta_U - \theta_V}{\theta_U} F(\mathbf{p}_{B_1} \vee \mathbf{p}_{B_3} \vee \mathbf{z}_{B_2})$$

$$\geq (1 - \theta_U) F(\mathbf{z}) + (\theta_U - \theta_V)(F(\mathbf{z}) - F(\mathbf{z} \vee \mathbf{1}_V))$$

$$= (1 - \theta_V) F(\mathbf{z}) - (\theta_U - \theta_V) F(\mathbf{z} \vee \mathbf{1}_V),$$

where $B_1, B_2, B_3$ is the partition for the dimension set $[d]$, i.e., $B_1 = \{i \in [d] : z_i \geq x_i\}$, $B_2 = \left\{i \in [d] : x_i > z_i, p_i \leq \frac{\theta_V}{\theta_U}\right\}$, $B_3 = \left\{i \in [d] : x_i > z_i, 1 \geq p_i > \frac{\theta_V}{\theta_U}\right\}$ and $\mathbf{p} = \mathbf{z} + \frac{1}{\theta_U}(\mathbf{x}(t) \vee \mathbf{z} - \mathbf{z}) = \mathbf{p}_{B_1} \vee \mathbf{p}_{B_2} \vee \mathbf{p}_{B_3}$. The first equality comes from $\mathbf{p}_{B_1} = \mathbf{z}_{B_1}, \mathbf{p}_{B_2} \geq \mathbf{z}_{B_2}, \mathbf{p}_{B_3} \geq \mathbf{z}_{B_3}$, i.e., $\mathbf{p}_{B_1} \vee \mathbf{p}_{B_2} \vee \mathbf{p}_{B_3} = \mathbf{p}_{B_1} \vee \mathbf{p}_{B_2} \vee \mathbf{p}_{B_3} \vee \mathbf{z}_{B_2}$. The first and the third inequality comes from concavity along the non-negative direction. The last inequality comes from the definition of DR-submodular function, i.e., $F(\mathbf{p}_{B_1} \vee \mathbf{z}_{B_2} \vee \mathbf{p}_{B_3}) + F(\mathbf{z} \vee \mathbf{1}_V) \geq F(\mathbf{z}) + F(\mathbf{z} \vee \mathbf{1}_V \vee \mathbf{p}_{B_1} \vee \mathbf{z}_{B_2} \vee \mathbf{p}_{B_3})$. Replacing $\mathbf{z}$ with $\mathbf{x}^* \wedge (u(t)\mathbf{1}_U \vee v(t)\mathbf{1}_V)$, we have:

$$F\Big(\mathbf{x}(t) \vee (\mathbf{x}^* \wedge (u(t)\mathbf{1}_U \vee v(t)\mathbf{1}_V))\Big) \geq \Big(1 - \theta_V\Big)F(\mathbf{x}^* \wedge (u(t)\mathbf{1}_U \vee v(t)\mathbf{1}_V))$$

$$- \Big(\theta_U - \theta_V\Big)F\Big((\mathbf{x}^* \wedge (u(t)\mathbf{1}_U \vee v(t)\mathbf{1}_V)) \vee \mathbf{1}_V\Big)$$

$$\geq \Big(1 - \theta_V\Big)\Big[F\big(\mathbf{x}^* \wedge (u(t)\mathbf{1}_U \vee v(t)\mathbf{1}_V) \wedge \mathbf{1}_V\big) + F\Big(\mathbf{x}^* \wedge (\mathbf{1}_V \vee u(t)\mathbf{1}_U)\Big) - F(\mathbf{x}^* \wedge \mathbf{1}_V)\Big]$$

$$- \Big(\theta_U - \theta_V\Big)F\Big((\mathbf{x}^* \wedge (u(t)\mathbf{1}_U \vee v(t)\mathbf{1}_V)) \vee \mathbf{1}_V\Big)$$

$$\geq u(t) \cdot \Big(1 - \theta_V\Big)F(\mathbf{x}^*) - \Big(1 - v(t)\Big) \cdot \Big(1 - \theta_V\Big)F(\mathbf{x}^* \wedge \mathbf{1}_V)$$

$$- \Big(\theta_U - \theta_V\Big)F\Big((\mathbf{x}^* \wedge (u(t)\mathbf{1}_U \vee v(t)\mathbf{1}_V)) \vee \mathbf{1}_V\Big)$$

The second inequality comes from $F(\mathbf{x}^* \wedge \mathbf{1}_V) + F(\mathbf{x}^* \wedge (u(t)\mathbf{1}_U \vee v(t)\mathbf{1}_V)) \geq F\big(\mathbf{x}^* \wedge (u(t)\mathbf{1}_U \vee v(t)\mathbf{1}_V) \wedge \mathbf{1}_V\big) + F\big(\mathbf{x}^* \wedge (\mathbf{1}_V \vee u(t)\mathbf{1}_U)\big)$. The third inequality comes from the concavity along with the non-negative direction. Therefore, we have

$$
\begin{aligned}
u(t)(1 - \theta_V(t))F(\mathbf{x}^*) \leq & F(\mathbf{x}(t)) + \Big\langle \nabla F(\mathbf{x}(t)), \mathbf{x}(t) \vee \Big(\mathbf{x}^* \wedge (u(t)\mathbf{1}_U \vee v(t)\mathbf{1}_V)\Big) - \mathbf{x}(t)\Big\rangle \\
& + \Big(1 - v(t)\Big)\Big(1 - \theta_V(t)\Big)F(\mathbf{x}^* \wedge \mathbf{1}_V) \\
& + \Big(\theta_U(t) - \theta_V(t)\Big)F\Big((\mathbf{x}^* \wedge (u(t)\mathbf{1}_U \vee v(t)\mathbf{1}_V)) \vee \mathbf{1}_V\Big)
\end{aligned}
$$

$\square$

The above inequality only consider the non-negative direction $\mathbf{x}(t) \vee \mathbf{x}^* - \mathbf{x}(t)$. When we consider the another non-negative direction $\mathbf{x}(t) - \mathbf{x}(t) \wedge \mathbf{x}^*$ simultaneously, we can obtain the following upper bound for optimal solutions.

**Lemma 5.** *For a non-monotone DR-submodular function $F$, then*

$$
(1 - \theta(t))F(\mathbf{x}^*) \leq F(\mathbf{x}^* \vee \mathbf{x}(t)) \leq 2F(\mathbf{x}(t)) + \langle \nabla F(\mathbf{x}(t)), \mathbf{x}^* - \mathbf{x}(t)\rangle \tag{18}
$$

*In addition, if $\mathbf{x}(t)$ is a stationary point, combining two non-negative directions $\mathbf{x}(t) \vee \mathbf{x}^* - \mathbf{x}(t)$ and $\mathbf{x}(t) - \mathbf{x}(t) \wedge \mathbf{x}^*$, we have:*

$$
\alpha F(\mathbf{x}(t) \vee \mathbf{y}) + (1 - \alpha)F(\mathbf{x}(t) \wedge \mathbf{y}) \leq F(\mathbf{x}(t)), \forall \alpha \in [0, \frac{1}{2}], \mathbf{y} \in \mathcal{C} \tag{19}
$$

*Proof of Lemma 5.* When we consider the following two non-negative direction $\mathbf{x} \vee \mathbf{x}^* - \mathbf{x}$ and $\mathbf{x} - \mathbf{x} \wedge \mathbf{x}^*$ simultaneously, we have

$$
\begin{aligned}
\langle \nabla F(\mathbf{x}(t)), \mathbf{x}(t) \vee \mathbf{x}^* - \mathbf{x}(t)\rangle &\geq F(\mathbf{x}(t) \vee \mathbf{x}^*) - F(\mathbf{x}(t)) \\
\langle \nabla F(\mathbf{x}(t)), \mathbf{x}(t) - \mathbf{x}^* \wedge \mathbf{x}(t)\rangle &\leq F(\mathbf{x}(t)) - F(\mathbf{x}(t) \wedge \mathbf{x}^*)
\end{aligned}
$$

where the inequality comes from the concavity along with the non-negative direction. Combining the above two inequalities, we have

$$
\langle \nabla F(\mathbf{x}(t)), \mathbf{x}^* - \mathbf{x}(t)\rangle \geq F(\mathbf{x}(t) \vee \mathbf{x}^*) + F(\mathbf{x}(t) \wedge \mathbf{x}^*) - 2F(\mathbf{x}(t))
$$

Because the non-negativity for the function $F$, we have

$$
\begin{aligned}
\langle \nabla F(\mathbf{x}(t)), \mathbf{x}^* - \mathbf{x}(t)\rangle + 2F(\mathbf{x}(t)) &\geq F(\mathbf{x}(t) \vee \mathbf{x}^*) + F(\mathbf{x}(t) \wedge \mathbf{x}^*) \\
&\geq F(\mathbf{x}(t) \vee \mathbf{x}^*) \geq (1 - \theta(t))F(\mathbf{x}^*)
\end{aligned}
$$

If $\mathbf{x}(t)$ is a stationary point for DR-submodular maximization, i.e., $\forall \mathbf{y} \in \mathcal{C}, 0 \geq \langle \nabla F(\mathbf{x}(t)), \mathbf{y} - \mathbf{x}(t)\rangle$, leveraging the above inequality, we have

$$
F(\mathbf{x}(t) \vee \mathbf{y}) + F(\mathbf{x}(t) \wedge \mathbf{y}) \leq 2F(\mathbf{x}(t))
$$

In addition, when the constraint is down-closed and $\mathbf{x}(t)$ is a stationary point, i.e., $\forall \mathbf{y} \in \mathcal{C}$ we have $\mathbf{x}(t) \wedge \mathbf{y} \in \mathcal{C}$, it implies

$$
F(\mathbf{x}(t) \vee (\mathbf{x}(t) \wedge \mathbf{y})) + F(\mathbf{x}(t) \wedge \mathbf{x}(t) \wedge \mathbf{y}) \leq 2F(\mathbf{x}(t))
$$

The convex combination for the above two inequality, we have

$$
\alpha F(\mathbf{x}(t) \vee \mathbf{y}) + (1 - \alpha)F(\mathbf{x}(t) \wedge \mathbf{y}) \leq F(\mathbf{x}(t)), \forall \alpha \in [0, \frac{1}{2}]
$$

$\square$

## B  DETAILED DESCRIPTION IN SECTION 3

In this section, we first provide a detailed description of the Lyapunov framework. We then demonstrate how to apply it to design approximation algorithms for DR-submodular maximization using stochastic biased gradients.

### B.1 Detailed Descriptions in Lyapunov Framework

The primary goal in designing approximation algorithms is to find an algorithm that achieves the highest possible approximation ratio. This objective is referred to as the ratio maximization problem Eq.(7). First, we illustrate the relationship between the ratio maximization problem and the difference maximization problem.

**Lemma 6.** *If* $\mathbf{x}_A(T)$ *and* $\alpha$ *are the optimal solution and the optimal value of* $\max_{\dot{\mathbf{x}}(t)} \frac{\mathbb{E}[F(\mathbf{x}(T))]}{F(\mathbf{x}^*)}$, *respectively, then* $\mathbf{x}_A(T)$ *is the optimal solution of* $\max_{\dot{\mathbf{x}}(t)} \mathbb{E}[F(\mathbf{x}(T))] - \alpha F(\mathbf{x}^*)$.

*Proof of Lemma 6.* Denote $\mathbf{x}_A(T)$ and $\alpha$ are the optimal solution and the optimal value for ratio maximization problem, i.e., $\mathbf{x}_A = \arg\max_{\mathbf{x}(T) \in \mathcal{ALG}} \frac{\mathbb{E}[F(\mathbf{x}(T))]}{F(\mathbf{x}^*)}$, $\alpha = \max \frac{\mathbb{E}[F(\mathbf{x}(T))]}{F(\mathbf{x}^*)}$. Suppose $\mathbf{x}_A$ is not the optimal solution for $\max_{\mathbf{x}(T) \in \mathcal{ALG}} \mathbb{E}[F(\mathbf{x}(T))] - \alpha F(\mathbf{x}^*)$. Then, there is a solution $\mathbf{x}_B(T) \in \mathcal{ALG}$ such that

$$\mathbb{E}[F(\mathbf{x}_B(T))] - \alpha F(\mathbf{x}^*) > \mathbb{E}[F(\mathbf{x}_A(T))] - \alpha F(\mathbf{x}^*) = 1 \Rightarrow \frac{\mathbb{E}[F(\mathbf{x}_B(T))]}{F(\mathbf{x}^*)} > \alpha$$

That is a contradiction with the optimal value $\alpha$. Therefore, $\mathbf{x}_A(T)$ is the optimal solution for $\max_{\mathbf{x}(T) \in \mathcal{ALG}} \mathbb{E}[F(\mathbf{x}(T))] - \alpha F(\mathbf{x}^*)$. $\square$

Leveraging Lemma 6 and regarding the objective value of Eq.(7) as $\alpha(t)$ that depends on the current time $t$, we reformulate the ratio maximization problem as the difference maximization given in Eq.(20).

$$\begin{aligned}
\max \quad & a(T)\mathbb{E}[F(\mathbf{x}(T))] - b(T)F(\mathbf{x}^*) \\
\text{s.t.} \quad & \dot{\mathbf{x}}(t) = \mathbf{f}(\mathbf{x}(t), \mathbf{g}(\mathbf{x}(t))) \\
& \mathbf{x}(T) \in \mathcal{C}, \mathbf{x}(0) = \mathbf{x}_{initial}
\end{aligned} \qquad (20)$$

where we write $\alpha(t)$ as $\frac{a(t)}{b(t)}$ and $\mathbf{x}(0)$ is the initial solution chosen before algorithm design. For obtaining the explicit solution of the Eq.(20), we construct the Lyapunov function in Definition 1, inspired by the objective function, and design algorithms by remaining non-decreasing Lyapunov function. Considering the derivative of the Lyapunov function:

$$\begin{aligned}
\dot{E}(\mathbf{x}(t)) \geq & \dot{a}(t)F(\mathbf{x}(t)) - \dot{b}(t) \cdot p(\mathbf{x}(t)) \cdot F(\mathbf{x}(t)) - \dot{b}(t)q(\mathbf{x}(t)) \|\mathbf{n}(\mathbf{x}(t)) + \mathbf{b}(\mathbf{x}(t))\|_2 \|v(\mathbf{x}(t))\|_2 \\
& + a(t)\langle \nabla F(\mathbf{x}(t)), \dot{\mathbf{x}}(t)\rangle - \dot{b}(t)q(\mathbf{x}(t))\langle \mathbf{g}(\mathbf{x}(t)), v(\mathbf{x}(t))\rangle \\
\geq & \dot{a}(t)F(\mathbf{x}(t)) - \dot{b}(t) \cdot p(\mathbf{x}(t)) \cdot F(\mathbf{x}(t)) \\
& + a(t)\langle \mathbf{g}(\mathbf{x}(t)), \dot{\mathbf{x}}(t)\rangle - \dot{b}(t)q(\mathbf{x}(t))\langle \mathbf{g}(\mathbf{x}(t)), v(\mathbf{x}(t))\rangle \\
& - a(t)\|\mathbf{n}(\mathbf{x}(t)) + \mathbf{b}(\mathbf{x}(t))\|_2 \|\dot{\mathbf{x}}(t)\|_2 \\
& - \dot{b}(t)q(\mathbf{x}(t))] \|\mathbf{n}(\mathbf{x}(t)) + \mathbf{b}(\mathbf{x}(t))\|_2 \|v(\mathbf{x}(t))\|_2
\end{aligned}$$

The first inequality is obtained by substituting the upper bound of the optimal solution from Eq.(9). Since the bias and noise are from gradient estimations and cannot be neglected within approximation algorithms, taking the expectation $\mathbb{E}_t[\cdot]$ both on two sides, we establish a sufficient condition such that the Lyapunov function remains approximately non-decreasing:

$$\begin{cases} \dot{a}(t) - \dot{b}(t) \cdot p(\mathbf{x}(t)) \geq 0 \\ \dot{\mathbf{x}}(t) = \frac{\dot{b}(t)q(\mathbf{x}(t))}{a(t)}v(\mathbf{x}(t)) \end{cases} \Rightarrow \dot{E}(\mathbf{x}(t)) \geq -2\dot{b}(t)q(\mathbf{x}(t))\mathbb{E}\left[A(\mathbf{x}(t), t) \|v(\mathbf{x}(t))\|_2\right]$$

The above errors come from stochastic biased gradients which is bounded by Assumption 1 and 2. When we ensure that the Lyapunov function is approximately non-decreasing, we have:

$$E(\mathbf{x}(T)) \geq E(\mathbf{x}(0)) - \int_0^T 2\dot{b}(t)q(\mathbf{x}(t))\mathbb{E}\left[A(\mathbf{x}(t), t) \|v(\mathbf{x}(t))\|_2\right] dt$$
$$\Downarrow$$
$$\begin{aligned}
\mathbb{E}[F(\mathbf{x}(T))] \geq & \frac{b(T) - b(0)}{a(T)}F(\mathbf{x}^*) + \frac{a(0)}{a(T)}F(\mathbf{x}(0)) \\
& - \frac{1}{a(T)}\int_0^T 2\dot{b}(t)q(\mathbf{x}(t))\mathbb{E}\left[A(\mathbf{x}(t), t) \|v(\mathbf{x}(t))\|_2\right] dt
\end{aligned}$$

The above procedure implies that the approximation ratio is $\frac{b(T)-b(0)}{a(T)}$ when ensuring the approximately non-decreasing Lyapunov function. Consequently, we reconstruct the variational optimization problem in Eq.(20) as follows: the target is to maximize the approximation ratio while minimizing errors arising from stochastic biased gradients, with the constraint that the Lyapunov function is approximate non-decreasing.

$$\max \quad \left\{ \frac{b(T)-b(0)}{a(T)}, -\frac{1}{a(T)}\int_0^T 2\dot{b}(t)q(\mathbf{x}(t))\mathbb{E}\left[A(\mathbf{x}(t),t)\,\|v(\mathbf{x}(t))\|_2\right]dt \right\}$$

$$\text{s.t. } \dot{a}(t) - \dot{b}(t)\cdot p(\mathbf{x}(t)) \geq 0$$

$$\dot{\mathbf{x}}(t) = \frac{\dot{b}(t)q(\mathbf{x}(t))}{a(t)}v(\mathbf{x}(t))$$

The above procedure is formulated in continuous time, serving as a theoretical construction rather than a directly implementable algorithm. Next, we illustrate how the potential function introduced in Definition 2 can be employed in the design of discrete-time algorithms. Considering the difference between adjacent iterations in algorithm $\mathbf{x}(t_j)$:

$$
\begin{aligned}
P(\mathbf{x}(t_{j+1})) - P(\mathbf{x}(t_j)) =& \, a(t_{j+1})[F(\mathbf{x}(t_{j+1})) - F(\mathbf{x}(t_j))] + [a(t_{j+1}) - a(t_j)]F(\mathbf{x}(t_j)) \\
& - [b(t_{j+1}) - b(t_j)]F(\mathbf{x}^*) \\
\geq & \, a(t_{j+1})[\langle\nabla F(\mathbf{x}(t_j)), \mathbf{x}(t_{j+1}) - \mathbf{x}(t_j)\rangle - \frac{L_1}{2}\|\mathbf{x}(t_{j+1}) - \mathbf{x}(t_j)\|_2^2] \\
& + [a(t_{j+1}) - a(t_j)]F(\mathbf{x}(t_j)) - [b(t_{j+1}) - b(t_j)]F(\mathbf{x}^*) \\
= & \, a(t_{j+1})\mathbb{E}[\langle\mathbf{g}(\mathbf{x}(t_j)), \mathbf{x}(t_{j+1}) - \mathbf{x}(t_j)\rangle - \frac{L_1}{2}\|\mathbf{x}(t_{j+1}) - \mathbf{x}(t_j)\|_2^2] \\
& + [a(t_{j+1}) - a(t_j)]F(\mathbf{x}(t_j)) - [b(t_{j+1}) - b(t_j)]F(\mathbf{x}^*) \\
& - a(t_{j+1})\langle\mathbf{b}(\mathbf{x}(t_j)) + \mathbf{n}(\mathbf{x}(t_j)), \mathbf{x}(t_{j+1}) - \mathbf{x}(t_j)\rangle \\
\geq & \, \langle\mathbf{g}(\mathbf{x}(t_j)), a(t_{j+1})[\mathbf{x}(t_{j+1}) - \mathbf{x}(t_j)] - [b(t_{j+1}) - b(t_j)]q(\mathbf{x}(t_j))v(\mathbf{x}(t_j))\rangle \\
& - \frac{L_1 a(t_{j+1})}{2}\|\mathbf{x}(t_{j+1}) - \mathbf{x}(t_j)\|_2^2 \\
& + [a(t_{j+1}) - a(t_j)]F(\mathbf{x}(t_j)) - [b(t_{j+1}) - b(t_j)]p(\mathbf{x}(t_j))F(\mathbf{x}(t_j)) \\
& - a(t_{j+1})\langle\mathbf{b}(\mathbf{x}(t_j)) + \mathbf{n}(\mathbf{x}(t_j)), \mathbf{x}(t_{j+1}) - \mathbf{x}(t_j)\rangle \\
& - [b(t_{j+1}) - b(t_j)]q(\mathbf{x}(t_j))\,\|\mathbf{b}(\mathbf{x}(t_j)) + \mathbf{n}(\mathbf{x}(t_j))\|_2\,\|v(\mathbf{x}(t_j))\|_2
\end{aligned}
$$

Taking the expectation both on two sides, there is a sufficient condition to ensure the potential function in Definition 2 is approximately non-decreasing:

$$
\begin{cases}
a(t_{j+1}) - a(t_j) - [b(t_{j+1}) - b(t_j)]\cdot p(\mathbf{x}(t_j)) \geq 0 \\
\mathbf{x}(t_{j+1}) - \mathbf{x}(t_j) = \frac{[b(t_{j+1})-b(t_j)]q(\mathbf{x}(t_j))}{a(t_{j+1})}v(\mathbf{x}(t_j))
\end{cases}
$$
$$\Downarrow$$
$$
\begin{aligned}
P(\mathbf{x}(t_{j+1})) - P(\mathbf{x}(t_j)) \geq & -2[b(t_{j+1}) - b(t_j)]\mathbb{E}[q(\mathbf{x}(t_j))A(\mathbf{x}(t_j),t_j)\,\|v(\mathbf{x}(t_j))\|_2] \\
& - \frac{L_1}{2}\mathbb{E}[a(t_{j+1})\,\|\mathbf{x}(t_{j+1}) - \mathbf{x}(t_j)\|_2^2]
\end{aligned}
$$

When we ensure that the potential function is approximately non-decreasing, we have:

$$
\begin{aligned}
P(\mathbf{x}(t_N)) \geq & \, P(\mathbf{x}(t_0)) - \sum_{j=0}^{N-1}\Big[2[b(t_{j+1}) - b(t_j)]\mathbb{E}[q(\mathbf{x}(t_j))A(\mathbf{x}(t_j),t_j)\,\|v(\mathbf{x}(t_j))\|_2] \\
& + \frac{L_1}{2}\mathbb{E}[a(t_{j+1})\,\|\mathbf{x}(t_{j+1}) - \mathbf{x}(t_j)\|_2^2]\Big]
\end{aligned}
$$
$$\Downarrow$$
$$\mathbb{E}[F(\mathbf{x}(t_N))] \geq \frac{b(t_N) - b(t_0)}{a(t_N)}F(\mathbf{x}^*) + \frac{a(t_0)}{a(t_N)}F(\mathbf{x}(t_0))$$

$$- \frac{\sum_{j=0}^{N-1}\left[2[b(t_{j+1}) - b(t_j)]\mathbb{E}[q(\mathbf{x}(t_j))A(\mathbf{x}(t_j),t_j)\,\|v(\mathbf{x}(t_j))\|_2] + \frac{L_1}{2}\mathbb{E}[a(t_{j+1})\,\|\mathbf{x}(t_{j+1}) - \mathbf{x}(t_j)\|_2^2]\right]}{a(t_N)}$$

Similarly, the above procedure implies that the approximation ratio is $\frac{b(t_N)-b(t_0)}{a(t_N)}$ when ensuring the approximately non-decreasing potential function. Accordingly, we reconstruct the sequence problem as follows: the target is to maximize the approximation ratio while minimizing errors arising from stochastic biased gradients and the $L_1$-smooth condition, with the constraint that the potential function is approximate non-decreasing.

$$\max \quad \left\{ \frac{b(t_N)-b(t_0)}{a(t_N)}, -\frac{1}{a(t_N)} \sum_{j=0}^{N-1} \left[ 2[b(t_{j+1})-b(t_j)]\mathbb{E}[q(\mathbf{x}(t_j))A(\mathbf{x}(t_j),t_j)\,\|v(\mathbf{x}(t_j))\|_2] \right. \right.$$
$$\left. \left. + \frac{L_1}{2}\mathbb{E}[a(t_{j+1})\,\|\mathbf{x}(t_{j+1})-\mathbf{x}(t_j)\|_2^2] \right] \right\}$$
$$\text{s.t. } a(t_{j+1})-a(t_j)-[b(t_{j+1})-b(t_j)]\cdot p(\mathbf{x}(t_j)) \geq 0$$
$$\mathbf{x}(t_{j+1})-\mathbf{x}(t_j) = \frac{[b(t_{j+1})-b(t_j)]q(\mathbf{x}(t_j))}{a(t_{j+1})}v(\mathbf{x}(t_j))$$

### B.2 DETAILED DESCRIPTION FOR DR-SUBMODULAR MAXIMIZATION WITH STOCHASTIC BIASED GRADIENTS

We investigate the DR-submodular maximization with stochastic biased gradients subject to three constraint classes: general constraints, down-closed convex constraints, and convex constraints with the largest element. By leveraging inequalities in Section 2, our approximation algorithms are constructed through solving the variational problem Eq.(10) and the sequence optimization problem Eq.(13). Our algorithm achieves an approximation ratio $(1-e^{-1})$ for monotone DR-submodular maximization subject to general convex constraints, matching the hardness result for monotone DR-submodular maximization under the above three constraints. Consequently, in our work, we focus exclusively on general convex constraints when addressing monotone DR-submodular functions.

**Convex Constraints.** As a warm-up, we first consider general convex constraints. When $F$ is a monotone DR-submodular function, in light of Eq.(15), we can obtain the explicit formula for the upper bound of the optimal solution Eq.(9).

$$F(\mathbf{x}^*) \leq F(\mathbf{x}^* \vee \mathbf{x}(t)) \leq F(\mathbf{x}(t)) + \langle \nabla F(\mathbf{x}(t)), \mathbf{x}(t) \vee \mathbf{x}^* - \mathbf{x}(t) \rangle$$
$$\leq F(\mathbf{x}(t)) + \langle \nabla F(\mathbf{x}(t)), \mathbf{x}^* \rangle$$
$$= F(\mathbf{x}(t)) + \langle \mathbf{g}(\mathbf{x}(t)) - \mathbf{n}(\mathbf{x}(t)) - \mathbf{b}(\mathbf{x}(t)), \mathbf{x}^* \rangle$$
$$\leq F(\mathbf{x}(t)) + \langle \mathbf{g}(\mathbf{x}(t)), \mathbf{v}_{\max}(t) \rangle + \|\mathbf{n}(\mathbf{x}(t)) + \mathbf{b}(\mathbf{x}(t))\|_2\,\sqrt{D}$$

where the first and the third inequality comes from the monotonicity. The last inequality comes from Cauchy-Schwarz inequality and $\mathbf{v}_{\max} = \arg\max_{\mathbf{v}\in\mathcal{C}}\langle \mathbf{g}(\mathbf{x}(t)), \mathbf{v} \rangle$, $\|\mathbf{x}^*\|_2 \leq \sqrt{D} = \max_{\mathbf{x},\mathbf{y}\in\mathcal{C}}\|\mathbf{x}-\mathbf{y}\|_2$. When $F$ is non-monotone DR-submodular function, in light of Eq.(18), the upper bound of the optimal solution Eq.(9) can be rewrite as follows:

$$(1-\theta(t))F(\mathbf{x}^*) \leq F(\mathbf{x}^* \vee \mathbf{x}(t)) \leq 2F(\mathbf{x}(t)) + \langle \nabla F(\mathbf{x}(t)), \mathbf{x}^* - \mathbf{x}(t) \rangle$$
$$\leq 2F(\mathbf{x}(t)) + \langle \mathbf{g}(\mathbf{x}(t)) - \mathbf{b}(\mathbf{x}(t)) - \mathbf{n}(\mathbf{x}(t)), \mathbf{x}^* - \mathbf{x}(t) \rangle$$
$$\leq 2F(\mathbf{x}(t)) + \max_{\mathbf{v}\in\mathcal{C}}\langle \mathbf{g}(\mathbf{x}(t)), \mathbf{v} - \mathbf{x}(t) \rangle + \|\mathbf{n}(\mathbf{x}(t)) + \mathbf{b}(\mathbf{x}(t))\|_2\,\sqrt{D}$$

Simultaneously taking expectations $\mathbb{E}_t[\cdot]$ on both sides of the equation, we can obtain the explicit formula in Eq.(10) as follows:

$$\begin{cases} \textbf{if monotone} & \begin{cases} \dot{a}(t)-\dot{b}(t)\cdot p(\mathbf{x}(t))=0 \\ \dot{\mathbf{x}}(t)=\frac{\dot{b}(t)q(\mathbf{x}(t))}{a(t)}v(\mathbf{x}(t)) \\ p(\mathbf{x}(t))=q(\mathbf{x}(t))=1 \end{cases} \Rightarrow \begin{cases} \dot{\mathbf{x}}(t)=\frac{\dot{a}(t)}{a(t)}v(\mathbf{x}(t)) \\ v(\mathbf{x}(t))=\mathbf{v}_{\max}(t) \\ \mathbf{v}_{\max}(t)=\arg\max_{\mathbf{v}\in\mathcal{C}}\langle \mathbf{g}(\mathbf{x}(t)),\mathbf{v}\rangle \end{cases} \\[2em] \textbf{if non-monotone} & \begin{cases} \dot{a}(t)-\dot{b}(t)\cdot p(\mathbf{x}(t))=0 \\ \dot{\mathbf{x}}(t)=\frac{\dot{b}(t)q(\mathbf{x}(t))}{a(t)}v(\mathbf{x}(t)) \\ p(\mathbf{x}(t))=2q(\mathbf{x}(t))=\frac{2}{1-\theta(t)} \end{cases} \Rightarrow \begin{cases} \dot{\mathbf{x}}(t)=\frac{\dot{a}(t)}{2a(t)}v(\mathbf{x}(t)) \\ v(\mathbf{x}(t))=\mathbf{v}_{\max}(t)-\mathbf{x}(t) \\ \mathbf{v}_{\max}(t)=\arg\max_{\mathbf{v}\in\mathcal{C}}\langle \mathbf{g}(\mathbf{x}(t)),\mathbf{v}\rangle \end{cases} \end{cases}$$
$$(21)$$

For monotone cases, we choose $\mathbf{x}(0) = \mathbf{0}$, which means that $\mathbf{x}(T)$ can be represented by the convex combination of solutions in the convex feasible domain for $T = 1$. There is a natural choice $a(t) = e^t$. Similarly, for non-monotone cases, if $\mathbf{x}(0) \in \mathcal{C}$ and $\mathbf{v}_{\max}(t) \in \mathcal{C}$, $\mathbf{x}(T) \in \mathcal{C}$ because $\mathbf{x}(T)$ is also the convex combination of solutions in the convex feasible domain.

$$
\begin{cases}
\textbf{if monotone} & \mathbf{x}(T) - \mathbf{x}(0) = \int_0^1 \mathbf{v}_{\max}(t) d\ln a(t) \\
\textbf{if non-monotone} & \lim_{\epsilon \to 0} \mathbf{x}(t+\epsilon) - \mathbf{x}(t) = \lim_{\epsilon \to 0} \int_t^{t+\epsilon} \frac{\dot{a}(s)}{2a(s)} v(\mathbf{x}(s)) ds = \lim_{\epsilon \to 0} \frac{\dot{a}(t)}{2a(t)}[\mathbf{v}_{\max}(t) - \mathbf{x}(t)]\epsilon
\end{cases}
$$

Additionally, because of $\mathbf{v}_{\max}(t) \le \mathbf{1}$, for non-monotone cases, we have the following upper bound for $\mathbf{x}(t)$ to obtain the explicit formula for $\theta(t)$:

$$
\dot{\mathbf{x}}(t) \le \frac{\dot{a}(t)}{2a(t)}(\mathbf{1} - \mathbf{x}(t)) \Rightarrow \mathbf{x}(t) \le \mathbf{1} - (\mathbf{1} - \mathbf{x}(0))e^{-\int_0^t \frac{\dot{a}(s)}{2a(s)} ds} = \mathbf{1} - (\mathbf{1} - \mathbf{x}(0))\sqrt{\frac{a(0)}{a(t)}}
$$

Combining Eq.(21) with $p(\mathbf{x}(t)) = 1$ for monotone cases and $p(\mathbf{x}(t)) = \frac{2}{\|\mathbf{1} - \mathbf{x}(0)\|_\infty \sqrt{a(0)/a(t)}}$, we have:

$$
\begin{cases}
\dot{a}(t) = \dot{b}(t) \Rightarrow \dfrac{b(T) - b(0)}{a(T)} = \dfrac{a(T) - a(0)}{a(T)} = \dfrac{e - 1}{e} & \textbf{if monotone} \\
\|\mathbf{1} - \mathbf{x}(0)\|_\infty \dot{a}(t)\sqrt{\dfrac{a(0)}{a(t)}} = 2\dot{b}(t) \Rightarrow \dfrac{b(T) - b(0)}{a(T)} = \|\mathbf{1} - \mathbf{x}(0)\|_\infty \left[\sqrt{\dfrac{a(0)}{a(T)}} - \dfrac{a(0)}{a(T)}\right] & \textbf{if non-monotone} \\
\qquad\qquad\qquad\qquad\qquad\qquad\qquad\qquad \le \dfrac{\|\mathbf{1} - \mathbf{x}(0)\|_\infty}{4}
\end{cases}
$$

The explicit formula is as follows, when $T = 1$:

$$
\begin{cases}
a(t) = e^t, b(t) = e^t, & \textbf{if monotone} \\
a(t) = (t+1)^2, b(t) = \|\mathbf{1} - \mathbf{x}(0)\|_\infty t, & \textbf{if non-monotone}
\end{cases}
$$

---

**Algorithm 1** Frank-Wolfe Variant for Convex Constraints

---

**Input:** $F$: objective function, $\mathcal{C}$: feasible domain
**Output:** the solution $\mathbf{x}(1)$

1: Initialize $\mathbf{x}(0) = \mathbf{0}$, $t_0 = 0$
2: **for** $j = 0$ to $N - 1$ **do**
3:     Estimate the gradient $\mathbf{g}(\mathbf{x}(t_j))$
4:     $\mathbf{v}(t_j) = \arg\max_{\mathbf{v} \in \mathcal{C}} \langle \mathbf{g}(\mathbf{x}(t_j)), \mathbf{v} \rangle$
5:     $\mathbf{x}(t_{j+1}) = \begin{cases} \mathbf{x}(t_j) + (1 - e^{\frac{-1}{N}})\mathbf{v}(t_j) & \textbf{if monotone} \\ (1 - \frac{j+N}{(j+1+N)^2})\mathbf{x}(t_j) + \frac{j+N}{(j+1+N)^2}\mathbf{v}(t_j) & \textbf{if non-monotone} \end{cases}$
6: **end for**

---

The aforementioned procedure focuses on continuous-time settings, which cannot be directly implemented on computers. Next, we introduce the analysis procedure based on the potential function. Instead of directly solving Eq.(13), we exploit the explicit formula for parameter functions in the Lyapunov function to the potential function. Therefore, the constraint in Eq.(13) is as follows:

$$
\begin{aligned}
& a(t_{j+1}) - a(t_j) - [b(t_{j+1}) - b(t_j)]p(\mathbf{x}(t_j)) \\
&= \begin{cases} e^{t_{j+1}} - e^{t_j} - [e^{t_{j+1}} - e^{t_j}] \cdot 1 = 0 & \textbf{if monotone} \\ (t_{j+1} + 1)^2 - (t_j + 1)^2 - 2(t_j + 1)[t_{j+1} - t_j] = (t_{j+1} - t_j)^2 \ge 0 & \textbf{if non-monotone} \end{cases}
\end{aligned}
$$

This implies that parameter functions within Lyapunov function also satisfies the constraint in Eq.(13). Consequently, we can directly consider the discrete time for parameter functions in the Lyapunov function while retaining the same approximation ratio:

$$
\textbf{If monotone:} \frac{b(t_N) - b(t_0)}{a(t_N)} = 1 - e^{-1}, \quad \textbf{If non-monotone:} \frac{b(t_N) - b(t_0)}{a(t_N)} = \frac{1}{4}\|\mathbf{1} - \mathbf{x}(0)\|_\infty
$$

Therefore, we can analyze the second objective function in Eq.(13) to obtain the complexity of our algorithms, which maintains the above approximation ratio. Specially, when we consider a constant time step, i.e., the time sequence is given by $t_j = \frac{j}{N}$, we have:

$$\sum_{j=0}^{N-1} \left[ 2[b(t_{j+1}) - b(t_j)]\mathbb{E}[q(\mathbf{x}(t_j))A(\mathbf{x}(t_j), t_j)\,\|v(\mathbf{x}(t_j))\|_2] + \frac{L_1}{2}\mathbb{E}[a(t_{j+1})\,\|\mathbf{x}(t_{j+1}) - \mathbf{x}(t_j)\|_2^2] \right]$$

$$\leq \begin{cases} \sum_{j=0}^{N-1} 2(e^{t_{j+1}} - e^{t_j})\sqrt{D}\mathbb{E}[A(\mathbf{x}(t_j), t_j)] + \frac{L_1(e^{t_{j+1}} - e^{t_j})^2}{2e^{t_{j+1}}}D & \text{if monotone} \\ \sum_{j=0}^{N-1} 2(t_{j+1} - t_j)(t_j + 1)\sqrt{D}\mathbb{E}[A(\mathbf{x}(t_j), t_j)] + \frac{L_1(t_{j+1} - t_j)^2(t_j+1)^2}{2(t_{j+1}+1)^2}D & \text{if non-monotone} \end{cases}$$

$$= \begin{cases} \sum_{j=0}^{N-1} 2e^{\frac{j+1}{N}}(1 - e^{-\frac{1}{N}})\sqrt{D}\mathbb{E}[A(\mathbf{x}(t_j), t_j)] + \frac{L_1}{2}e^{\frac{j+1}{N}}(1 - e^{-\frac{1}{N}})^2 D & \text{if monotone} \\ \sum_{j=0}^{N-1} 2\frac{j+N}{N^2}\mathbb{E}[A(\mathbf{x}(t_j), t_j)] + \frac{L_1(N+j)^2}{2N^2(N+j+1)^2}D & \text{if non-monotone} \end{cases}$$

$$\leq \begin{cases} \sum_{j=0}^{N-1} \frac{2e^{\frac{j+1}{N}}\sqrt{D}}{N}\mathbb{E}[A(\mathbf{x}(t_j), t_j)] + \frac{L_1 D}{N} & \text{if monotone} \\ \sum_{j=0}^{N-1} 2\frac{j+N}{N^2}\sqrt{D}\mathbb{E}[A(\mathbf{x}(t_j), t_j)] + \frac{L_1 D}{2N} & \text{if non-monotone} \end{cases}$$

If the stochastic biased gradient is strengthened to the exact gradient, the first item will be reduced to 0. Specifically, when $N = \frac{L_1 D}{\epsilon}$, this recovers the approximation algorithm for DR-submodular maximization with the exact gradient subject to convex constraint. Therefore, for DR-submodular maximization with the stochastic biased gradient, we obtain approximation algorithms with the following theoretical guarantees. The iteration complexity can be determined by the total variance of the stochastic bias gradient and the $L_1$-smooth condition.

**Approximation Ratio:** $\begin{cases} 1 - e^{-1} - O(\epsilon) & \textbf{If monotone} \\ \frac{1}{4} - O(\epsilon) & \textbf{If non-monotone} \end{cases}$

**Algorithm:** $\mathbf{x}(t_{j+1}) = \begin{cases} \mathbf{x}(t_j) + (1 - e^{\frac{-1}{N}})\mathbf{v}(t_j) & \textbf{if monotone} \\ (1 - \frac{j+N}{(j+1+N)^2})\mathbf{x}(t_j) + \frac{j+N}{(j+1+N)^2}\mathbf{v}(t_j) & \textbf{if non-monotone} \end{cases}$

**Iteration Complexity:** $\begin{cases} \sum_{j=0}^{N-1} \frac{2e^{\frac{j+1}{N}}\sqrt{D}}{N}\mathbb{E}[A(\mathbf{x}(t_j), t_j)] + \frac{L_1 D}{N} & \textbf{if monotone} \\ \sum_{j=0}^{N-1} 2\frac{j+N}{N^2}\sqrt{D}\mathbb{E}[A(\mathbf{x}(t_j), t_j)] + \frac{L_1 D}{2N} & \textbf{if non-monotone} \end{cases} \leq O(\epsilon)$

**Down-Closed and Convex Constraints.** On the basis of the convex set, we consider the constraint set is also down-closed. If $\mathbf{1}_V$ in Eq.(17) comes from a stationary point $\mathbf{y}$, i.e., $i \in V$ with the probability $y_i$, then we have the following upper bound of the optimal solution:

$$F(\mathbf{x}^*) \leq \frac{F(\mathbf{x}(t)) + \max_{\mathbf{v} \in \mathcal{C}, \mathbf{v} \leq (\mathbf{1} - \mathbf{x}(t)) \odot (u(t)\mathbf{1}_U \vee v(t)\mathbf{1}_V)} \langle \mathbf{g}(\mathbf{x}(t)), \mathbf{v}\rangle - \sqrt{D}\,\|\mathbf{n}(\mathbf{x}(t)) + \mathbf{b}(\mathbf{x}(t))\|_2}{u(t)(1 - \theta_V(t))}$$

$$+ \frac{(1 - v(t))(1 - \theta_V(t))F(\mathbf{x}^* \wedge \mathbf{1}_V)}{u(t)(1 - \theta_V(t))}$$

$$+ \frac{(\theta_U(t) - \theta_V(t))F\left((\mathbf{x}^* \wedge (u(t)\mathbf{1}_U \vee v(t)\mathbf{1}_V)) \vee \mathbf{1}_V\right)}{u(t)(1 - \theta_V(t))}$$

$$\leq \frac{F(\mathbf{x}(t)) + \max_{\mathbf{v} \in \mathcal{C}, \mathbf{v} \leq (\mathbf{1} - \mathbf{x}(t)) \odot (u(t)\mathbf{1}_U \vee v(t)\mathbf{1}_V)} \langle \mathbf{g}(\mathbf{x}(t)), \mathbf{v}\rangle - \sqrt{D}\,\|\mathbf{n}(\mathbf{x}(t)) + \mathbf{b}(\mathbf{x}(t))\|_2}{u(t)(1 - \theta_V(t))}$$

$$+ \left[\frac{1 - v(t)}{u(t)v(t)} + \frac{(\theta_U(t) - \theta_V(t))}{u(t)(1 - \theta_V(t))}\right]F(\mathbf{1}_V)$$

Since the upper bound of the optimal solution incorporates information about the stationary point, we consider the following Lyapunov function and the upper bound of the optimal solution that combines the stationary point.

$$E(\mathbf{x}(t)) = a(t)\mathbb{E}[F(\mathbf{x}(t))] + c(t)F(\mathbf{y}) - b(t)F(\mathbf{x}^*)$$

$$F(\mathbf{x}^*) \leq p(\mathbf{x}(t))F(\mathbf{x}(t)) + w(\mathbf{x}(t))F(\mathbf{y}) + q(\mathbf{x}(t))\mathbb{E}_t[\langle \mathbf{g}(\mathbf{x}(t)), v(\mathbf{x}(t))\rangle]$$
$$+ q(\mathbf{x}(t))\mathbb{E}_t[\|\mathbf{n}(\mathbf{x}(t)) + \mathbf{b}(\mathbf{x}(t))\|_2\,\|v(\mathbf{x}(t))\|_2]$$

When the Lyapunov function is non-decreasing, we have:

$$\frac{a(T)\mathbb{E}[F(\mathbf{x}(T))]}{a(T) + c(T) - c(0)} + \frac{[c(T) - c(0)]F(\mathbf{y})}{a(T) + c(T) - c(0)} \geq \frac{[b(T) - b(0)]F(\mathbf{x}^*)}{a(T) + c(T) - c(0)} + \frac{a(0)F(\mathbf{x}(0))}{a(T) + c(T) - c(0)}$$

It can be regarded as the algorithm returning the solution $\mathbf{x}(T)$ with the probability $\frac{a(T)}{a(T)+c(T)-c(0)}$ and the stationary point with the probability $\frac{c(T)-c(0)}{a(T)+c(T)-c(0)}$. Considering the derivative of the Lyapunov function when taking expectations $\mathbb{E}_t[\cdot]$ on both sides:

$$\begin{aligned}
\dot{E}(\mathbf{x}(t)) \geq &[\dot{a}(t) - \dot{b}(t) \cdot p(\mathbf{x}(t))]F(\mathbf{x}(t)) + [\dot{c}(t) - \dot{b}(t) \cdot w(\mathbf{x}(t))]F(\mathbf{y}) \\
&+ a(t)\langle \mathbf{g}(\mathbf{x}(t)), \dot{\mathbf{x}}(t)\rangle - \dot{b}(t)q(\mathbf{x}(t))\langle \mathbf{g}(\mathbf{x}(t)), v(\mathbf{x}(t))\rangle \\
&- a(t)\|\mathbf{n}(\mathbf{x}(t)) + \mathbf{b}(\mathbf{x}(t))\|_2 \|\dot{\mathbf{x}}(t)\|_2 \\
&- \dot{b}(t)q(\mathbf{x}(t))]\|\mathbf{n}(\mathbf{x}(t)) + \mathbf{b}(\mathbf{x}(t))\|_2 \|v(\mathbf{x}(t))\|_2
\end{aligned}$$

Similarly, by combining coefficients of identical terms in the derivative of the Lyapunov function, we have the following sufficient condition. Given that the total variance of stochastic biased gradients is non-negligible, this condition guarantees the Lyapunov function is approximately non-decreasing.

$$\begin{cases}
\dot{a}(t) - \dot{b}(t) \cdot p(\mathbf{x}(t)) = 0 \\
\dot{c}(t) - \dot{b}(t) \cdot w(\mathbf{x}(t)) = 0 \\
\dot{\mathbf{x}}(t) = \frac{\dot{b}(t)q(\mathbf{x}(t))}{a(t)}v(\mathbf{x}(t)) \\
p(\mathbf{x}(t)) = q(\mathbf{x}(t)) = \dfrac{1}{u(t)(1 - \theta_V(t))} \\
w(\mathbf{x}(t)) = \dfrac{1 - v(t)}{u(t)v(t)} + \dfrac{(\theta_U(t) - \theta_V(t))}{u(t)(1 - \theta_V(t))}
\end{cases} \tag{22}$$

$$\Downarrow$$

$$\begin{cases}
\dot{\mathbf{x}}(t) = \dfrac{\dot{a}(t)}{a(t)}v(\mathbf{x}(t)) \\
v(\mathbf{x}(t)) = \arg\max_{\mathbf{v}\in\mathcal{C}, \mathbf{v}\leq(\mathbf{1}-\mathbf{x}(t))\odot(u(t)\mathbf{1}_U \vee v(t)\mathbf{1}_V)}\langle \mathbf{g}(\mathbf{x}(t)), \mathbf{v}\rangle
\end{cases}$$

Where $p(\mathbf{x}(t)), q(\mathbf{x}(t)), w(\mathbf{x}(t))$ comes from the expectations $\mathbb{E}_t[\cdot]$ on both sides of the upper bound for the optimal solution. Formally, if we choose $\mathbf{x}(0) = \mathbf{0}$, $a(t) = e^t$, and $T = 1$, $\mathbf{x}(1)$ is the convex combination of feasible solutions.

$$\mathbf{x}(T) - \mathbf{x}(0) = \int_0^T v(\mathbf{x}(T))d\ln(a(t))$$

Additionally, according to the upper bound of the optimal solution, we can obtain the explicit formula $\theta_V(t)$ and $\theta_U(t)$:

$$\dot{\mathbf{x}}(t) \leq (\mathbf{1} - \mathbf{x}(t)) \odot (u(t)\mathbf{1}_U \vee v(t)\mathbf{1}_V) \Rightarrow \begin{cases} x_i(t) \leq 1 - e^{-\int_0^t u(\tau)d\tau} & i \in U \\ x_i(t) \leq 1 - e^{-\int_0^t v(\tau)d\tau} & i \in V \end{cases}$$

Therefore, with the constraints for the approximation algorithm design given in Eq.(22), the corresponding objective function for the algorithm design is as follows:

$$c(1) - c(0) = \int_0^1 e^{t-\int_0^t v(\tau)d\tau}\left[\frac{1}{v(t)} - e^{-\int_0^t u(\tau)-v(\tau)d\tau}\right]dt$$

$$\max_{\text{Eq.(22)}} \frac{b(T) - b(0)}{a(T) + c(T) - c(0)} = \max_{\text{Eq.(22)}} \frac{\int_0^1 e^{t-\int_0^t v(\tau)d\tau}u(t)dt}{e + \int_0^1 e^{t-\int_0^t v(\tau)d\tau}\left[\frac{1}{v(t)} - e^{-\int_0^t u(\tau)-v(\tau)d\tau}\right]dt}$$

To obtain the explicit solution for the above objective function, we first consider the derivative for $u(t)$ in numerator and denominator.

$$\frac{\partial e^{t-\int_0^t v(\tau)d\tau}u(t)}{\partial u} = e^{t-\int_0^t v(\tau)d\tau} \geq \frac{\partial - e^{t-\int_0^t u(\tau)d\tau}}{\partial u} = e^{t-\int_0^t u(\tau)d\tau}$$

$$\Downarrow$$

$$u(t) = 1 \forall t \in [0, 1] \text{ such that our target is as large as possible.}$$

Table 3: Parameter functions in the Lyapunov function

| | $a(t)$ | $c(t)$ | $b(t)$ | $p(\mathbf{x}(t))$ | $q(\mathbf{x}(t))$ | $w(\mathbf{x}(t))$ |
|---|---|---|---|---|---|---|
| $t \in [0, s]$ | $e^t$ | $2e^t - t$ | $e^t$ | $1$ | $1$ | $2 - e^{-t}$ |
| $t \in (s, 1]$ | $e^t$ | $e^s \cdot (t + 2 - s) - t$ | $e^s \cdot (t + 1 - s)$ | $e^{t-s}$ | $e^{t-s}$ | $1 - e^{-s}$ |

When $u(t)$ is chosen to be large, the contribution for the numerator is greater than that for the denominator. This implies that selecting large possible $u(t)$ results in a larger value of the objective function. Leveraging this observation, we can simplify $F\Big((\mathbf{x}^* \wedge (\mathbf{1}_U \vee v(t)\mathbf{1}_V)) \vee \mathbf{1}_V\Big) = F(\mathbf{x}^* \vee \mathbf{1}_V)$, which implies the objective function can be rewrote as follows:

$$\max_{\text{Eq.(22)}} \frac{\int_0^1 e^{t - \int_0^t v(\tau)d\tau} dt}{e + \int_0^1 e^{t - \int_0^t v(\tau)d\tau} \left[2 - v(t) - e^{-t + \int_0^t v(\tau)d\tau dt}\right] dt} = \max_{\text{Eq.(22)}} \frac{1}{\frac{e - 2 + e^{1 - \int_0^1 v(\tau)d\tau}}{\int_0^1 e^{t - \int_0^t v(\tau)d\tau} dt} + 1}$$

Therefore, we aim to minimize $e^{1 - \int_0^1 v(\tau)d\tau}$ while maximizing $\int_0^1 e^{t - \int_0^t v(\tau)d\tau}$. This in turn implies that there is a $s \in [0, 1]$ for which the explicit formula for $v(t)$ can be derived.

$$v(t) = 0 \forall t \in [0, s]; v(t) = 1 \forall t \in (s, 1]$$

Accordingly, parameter functions in Lyapunov function are determined as shown in Tab.3 and the variational problem can be formulated as a numerical optimization problem. Solving the following numerical optimization problem, we have:

$$\max_{s \in [0,1]} \frac{(2 - s) \cdot e^s - 1}{e - 3 + (3 - s)e^s} \Rightarrow s \approx 0.372, \quad \frac{[b(T) - b(0)]}{a(T) + c(T) - c(0)} \approx 0.385$$

Next, we introduce the analysis procedure based on the potential function. Correspondingly, we consider the following potential function and the upper bound of the optimal solution in discrete time:

$$P(\mathbf{x}(t_j)) = a(t_j)\mathbb{E}[F(\mathbf{x}(t_j))] + c(t_j)F(\mathbf{y}) - b(t_j)F(\mathbf{x}^*)$$
$$F(\mathbf{x}^*) \leq p(\mathbf{x}(t_j))F(\mathbf{x}(t_j)) + w(\mathbf{x}(t_j))F(\mathbf{y}) + q(\mathbf{x}(t_j))\mathbb{E}_{t_j}[\langle \mathbf{g}(\mathbf{x}(t_j)), v(\mathbf{x}(t_j))\rangle]$$
$$+ q(\mathbf{x}(t_j))\mathbb{E}_{t_j}[\|\mathbf{n}(\mathbf{x}(t_j)) + \mathbf{b}(\mathbf{x}(t_j))\|_2 \|v(\mathbf{x}(t_j))\|_2]$$

If the potential function is non-decreasing, we have:

$$\frac{a(t_N)\mathbb{E}[F(\mathbf{x}(t_N))]}{a(t_N) + c(t_N) - c(t_0)} + \frac{[c(t_N) - c(t_0)]F(\mathbf{y})}{a(t_N) + c(t_N) - c(t_0)} \geq \frac{[b(t_N) - b(t_0)]F(\mathbf{x}^*)}{a(t_N) + c(t_N) - c(t_0)} + \frac{a(t_0)F\big(\mathbf{x}(t_0)\big)}{a(t_N) + c(t_n) - c(t_0)}$$

To ensure that the non-decreasing potential function, we consider the difference between adjacent iterations in algorithm $\mathbf{x}(t_j)$ when taking expectations $\mathbb{E}_t[\cdot]$ on both sides:

$$P(\mathbf{x}(t_{j+1})) - P(\mathbf{x}(t_j)) \geq \langle \mathbf{g}(\mathbf{x}(t_j)), a(t_{j+1})[\mathbf{x}(t_{j+1}) - \mathbf{x}(t_j)] - [b(t_{j+1}) - b(t_j)]q(\mathbf{x}(t_j))v(\mathbf{x}(t_j))\rangle$$
$$- \frac{L_1 a(t_{j+1})}{2}\|\mathbf{x}(t_{j+1}) - \mathbf{x}(t_j)\|_2^2$$
$$+ [a(t_{j+1}) - a(t_j)]F(\mathbf{x}(t_j)) - [b(t_{j+1}) - b(t_j)]p(\mathbf{x}(t_j))F(\mathbf{x}(t_j))$$
$$+ [c(t_{j+1}) - c(t_j)]F(\mathbf{y}) - [b(t_{j+1}) - b(t_j)]w(\mathbf{x}(t_j))F(\mathbf{y})$$
$$- a(t_{j+1})\langle \mathbf{b}(\mathbf{x}(t_j)) + \mathbf{n}(\mathbf{x}(t_j)), \mathbf{x}(t_{j+1}) - \mathbf{x}(t_j)\rangle$$
$$- [b(t_{j+1}) - b(t_j)]q(\mathbf{x}(t_j))\|\mathbf{b}(\mathbf{x}(t_j)) + \mathbf{n}(\mathbf{x}(t_j))\|_2 \|v(\mathbf{x}(t_j))\|_2$$

There are two types of errors: (1) error from stochastic biased gradients, (2) error from $L_1$ smooth condition. Combining coefficients of identical terms in the difference of the potential function, we have the following sufficient condition to ensure that the potential function is approximate non-decreasing.

$$\begin{cases} a(t_{j+1}) - a(t_j) - [b(t_{j+1}) - b(t_j)] \cdot p(\mathbf{x}(t_j)) \geq 0 \\ c(t_{j+1}) - c(t_j) - [b(t_{j+1}) - b(t_j)]w(\mathbf{x}(t_j)) \geq 0 \\ \mathbf{x}(t_{j+1}) - \mathbf{x}(t_j) = \frac{[b(t_{j+1}) - b(t_j)]q(\mathbf{x}(t_j))}{a(t_{j+1})}v(\mathbf{x}(t_j)) \end{cases}$$

To substituting the explicit formula of the parameter functions of the Lyapunov function into the potential function, we first check these parameter functions also satisfy the constraints for approximately non-decreasing potential functions. Since we partition the parameter functions of the Lyapunov function into two segments, the difference in the potential function between adjacent iterations is also split into two corresponding parts:

$$a(t_{j+1}) - a(t_j) - [b(t_{j+1}) - b(t_j)]p(\mathbf{x}(t_j))$$

$$= \begin{cases} e^{t_{j+1}} - e^{t_j} - [e^{t_{j+1}} - e^{t_j}] \cdot 1 = 0 & \text{if } j \leq 0.372N \\ e^{t_{j+1}} - e^{t_j} - e^{t_j}[t_{j+1} - t_j] \geq 0 & \text{if } j > 0.372N \end{cases}$$

$$c(t_{j+1}) - c(t_j) - [b(t_{j+1}) - b(t_j)]w(\mathbf{x}(t_j))$$

$$= \begin{cases} 2[e^{t_{j+1}} - e^{t_j}] - [t_{j+1} - t_j] - [e^{t_{j+1}} - e^{t_j}](2 - e^{-t_j}) \geq 0 & \text{if } j \leq 0.372N \\ e^s[t_{j+1} - t_j] - [t_{j+1} - t_j] - e^s[t_{j+1} - t_j](1 - e^{-s}) = 0 & \text{if } j > 0.372N \end{cases}$$

Therefore, we only need to analyze the remaining terms in the difference of the potential function to determine the complexity of the algorithm that is consistent with the second objective function in Eq.(13).

$$\sum_{j=0}^{N-1} \left[ 2[b(t_{j+1}) - b(t_j)]\mathbb{E}[q(\mathbf{x}(t_j))A(\mathbf{x}(t_j), t_j) \|v(\mathbf{x}(t_j))\|_2] + \frac{L_1}{2}\mathbb{E}[a(t_{j+1}) \|\mathbf{x}(t_{j+1}) - \mathbf{x}(t_j)\|_2^2] \right]$$

$$\leq \sum_{j=0}^{0.372N-1} 2(e^{t_{j+1}} - e^{t_j})\sqrt{D}\mathbb{E}[A(\mathbf{x}(t_j), t_j)] + \frac{L_1}{2}\frac{(e^{t_{j+1}} - e^{t_j})^2}{e^{t_{j+1}}}D$$

$$+ \sum_{j=0.372N}^{N-1} 2e^{t_j}(t_{j+1} - t_j)\sqrt{D}\mathbb{E}[A(\mathbf{x}(t_j), t_j)] + \frac{L_1}{2}\frac{e^{2t_j}(t_{j+1} - t_j)^2}{e^{t_{j+1}}}D$$

$$= \sum_{j=0}^{0.372N-1} 2e^{\frac{j+1}{N}}(1 - e^{-\frac{1}{N}})\sqrt{D}\mathbb{E}[A(\mathbf{x}(t_j), t_j)] + \frac{L_1}{2}e^{\frac{j+1}{N}}(1 - e^{-\frac{1}{N}})^2 D$$

$$+ \sum_{j=0.372N}^{N-1} 2\frac{e^{\frac{j}{N}}\sqrt{D}}{N}\mathbb{E}[A(\mathbf{x}(t_j), t_j)] + \frac{L_1}{2}\frac{e^{\frac{j-1}{N}}}{N^2}D$$

$$\leq \sum_{j=0}^{0.372N-1} \frac{2e^{\frac{j+1}{N}}\sqrt{D}}{N}\mathbb{E}[A(\mathbf{x}(t_j), t_j)] + \sum_{j=0.372N}^{N-1} 2\frac{j+1}{N^2}\sqrt{D}\mathbb{E}[A(\mathbf{x}(t_j), t_j)] + \frac{L_1 D(e-1)}{2N}$$

When the stochastic biased gradient is strengthened to the exact gradient, our method recovers the approximation algorithm for DR-submodular maximization under down-closed convex constraints with exact gradients with $N = \frac{L_1 D(e-1)}{2\epsilon}$. Therefore, for DR-submodular maximization with the stochastic biased gradient, we obtain an approximation algorithm with the following theoretical guarantees. The iteration complexity can be determined by the total variance of the stochastic bias gradient and the $L_1$-smooth condition.

**Approximation Ratio:** $0.385 - O(\epsilon)$

**Algorithm:** $\mathbf{x}(t_{j+1}) = \begin{cases} \mathbf{x}(t_j) + (1 - e^{\frac{-1}{N_1+N_2}})\mathbf{v}(t_j) & \text{If } j \leq 0.372N \\ \mathbf{x}(t_j) + \frac{1}{(N_1+N_2)e^{\frac{1}{N_1+N_2}}}\mathbf{v}(t_j) & \text{If } j > 0.372N \end{cases}$

$\mathbf{v}(t_j) = \begin{cases} \arg\max_{\mathbf{v}\in\mathcal{P}, \mathbf{v}\leq(1-\mathbf{x}(t_j))\odot\mathbf{1}_U} \langle\nabla F(\mathbf{x}(t_j)), \mathbf{v}\rangle & \text{If } j \leq 0.372N \\ \arg\max_{\mathbf{v}\in\mathcal{P}, \mathbf{v}\leq 1-\mathbf{x}(t_j)} \langle\nabla F(\mathbf{x}(t_j)), \mathbf{v}\rangle & \text{If } j > 0.372N \end{cases}$

**Iteration Complexity:** $\sum_{j=0}^{0.372N-1} \frac{2e^{\frac{j+1}{eN}}\sqrt{D}}{N}\mathbb{E}[A(\mathbf{x}(t_j), t_j)] + \sum_{j=0.372N}^{N-1} 2\frac{j+1}{eN^2}\sqrt{D}\mathbb{E}[A(\mathbf{x}(t_j), t_j)]$

$+ \frac{L_1 D(e-1)}{2N} \leq O(\epsilon)$

---

**Algorithm 2** Aided Frank-Wolfe Variant for Down-closed Convex Constraints

---

**Input:** $F$: objective function, $\mathcal{C}$: feasible domain, $s \in [0,1]$: parameter.
**Output:** $\mathbf{x}(1)$ with probability $1 - p$ and $\mathbf{y}$ with probability $p$

1: $\mathbf{y} \leftarrow$ Frank-Wolfe $(F, \mathcal{P})$
2: $V$ is a random subset of $[d]$. For all $i \in [d]$, $i \in V$ with probability of $y_i$ independently.
3: Initialize $\mathbf{x}(0) = \mathbf{0}$
4: **for** $j = 0$ to $N$ **do**
5:     **if** $j \leq s \cdot N - 1$ **then**
6:         $\mathbf{v}(t_j) = \arg\max_{\mathbf{v} \in \mathcal{C}, \mathbf{v} \leq (1 - \mathbf{x}(t_j)) \odot \mathbf{1}_U} \langle \nabla F(\mathbf{x}(t_j)), \mathbf{v} \rangle$
7:         $\mathbf{x}(t_{j+1}) = \mathbf{x}(t_j) + (1 - e^{\frac{-1}{N_1 + N_2}}) \mathbf{v}(t_j)$
8:     **else**
9:         $\mathbf{v}(t_j) = \arg\max_{\mathbf{v} \in \mathcal{P}, \mathbf{v} \leq 1 - \mathbf{x}(t_j)} \langle \nabla F(\mathbf{x}(t_j)), \mathbf{v} \rangle$
10:         $\mathbf{x}(t_{j+1}) = \mathbf{x}(t_j) + \frac{1}{(N_1 + N_2)e^{\frac{1}{N_1 + N_2}}} \mathbf{v}(t_j)$
11:     **end if**
12: **end for**

---

## C     Detailed Description in Section 4

In this section, we first introduce the total variance for the quantum gradient estimation algorithm. Next, we demonstrate the bias and the noise when the smoothing technique is adopted to estimate the gradient that is a classical algorithm. Since this constitutes a biased estimation of $\nabla F(\mathbf{x})$, we also extend the prior variance reduction technique originally developed for unbiased gradient estimations and provide a corresponding analysis.

### C.1     Quantum Zero-Order Algorithm

Jordan Algorithm is the gradient estimation algorithm on quantum computers whose query complexity is 1 for the function value oracle, instead of the query complexity is $O(d)$ for classical computers. In (van Apeldoorn et al., 2023), authors proposed an improved Jordan algorithm that adopt the state tomograph technique to reduce the bias for the Jordan algorithm to estimate the gradient.

**Lemma 7** (Suppressed-Bias Gradient Estimation (van Apeldoorn et al., 2023)). *Let $\epsilon, \delta \in (0, \frac{1}{6}]$ and $\mathbf{g} \in \mathbb{R}^d$ such that $\|\mathbf{g}\|_\infty \leq \frac{1}{3}$. Let $b = \lceil \log_2 \frac{2}{\epsilon} \rceil$ and $B = 2^b$. If*

$$\left\| |\psi\rangle - \frac{1}{\sqrt{B^d}} \sum_{\mathbf{y} \in G_b^d} e^{2\pi i B \langle \mathbf{g}, \mathbf{y} \rangle} |\mathbf{y}\rangle \right\|_2 \leq \frac{\delta}{24\lceil \ln \frac{6d}{\delta} \rceil + 3}$$

*given access to $8\lceil \ln \frac{6d}{\delta} \rceil + 1$ copies of $|\psi\rangle$, we can compute $\mathbf{k} \in [-\frac{1}{2}, \frac{1}{2}]^d$ satisfying*

$$Pr[\|\mathbf{k} - \mathbf{g}\|_\infty \geq \epsilon] \leq \delta, \quad \|\mathbb{E}[\mathbf{k}] - \mathbf{g}\|_\infty \leq \delta$$

*The procedure has a gate complexity $O(d \log \frac{d}{\delta} \log \frac{1}{\epsilon} \log(\frac{d}{\delta} \log \frac{1}{\epsilon}))$ and requires a corresponding circuit depth of $O(\log \frac{1}{\epsilon} \log(\frac{d}{\delta} \log \frac{1}{\epsilon}))$.*

The above lemma shows that the bias for the quantum gradient is $\delta$. Therefore, the total variance can be obtained by the following procedure.

$$\mathbb{E}[\|\mathbf{k} - \mathbf{g}\|_\infty] \leq Pr[\|\mathbf{k} - \mathbf{g}\|_\infty \geq \epsilon] \cdot \frac{5}{6} + Pr[\|\mathbf{k} - \mathbf{g}\|_\infty < \epsilon] \cdot \epsilon$$

However, when preparing the state $|\psi\rangle$, truncation errors arise in the function value $F$ due to the finite number of qubits. Leveraging the above lemma, next, we prove the total variance when accounting for the finite-bit precision of $F$. For convenience, we denote $G(\mathbf{x} + r\mathbf{y}) = F(\mathbf{x} + r\mathbf{y}) - F(\mathbf{x})$, $G_b = \{\frac{j}{2^b} - \frac{1}{2} + 2^{-(b+1)} : j \in \{0, ..., 2^b - 1\}\}$ and $G_b^d = \otimes^d G_b$. In Algorithms, we denote $|\psi\rangle = \frac{1}{\sqrt{2^{bd}}} \sum_{\mathbf{y}=\mathbf{0}}^{\mathbf{2^b}-\mathbf{1}} e^{i2\pi B G_\beta(\mathbf{x}+r\mathbf{y})} |\mathbf{y}\rangle = \frac{1}{\sqrt{2^{bd}}} \sum_{y_1=0}^{2^b-1} \cdots \sum_{y_d=0}^{2^b-1} e^{i2\pi B G_\beta(\mathbf{x}+r\mathbf{y})} | \otimes_{j=1}^d y_j \rangle$

---

**Algorithm 3** Suppressed-Bias Phase Estimation

---

**Input:** $|\psi\rangle = \frac{1}{\sqrt{2^b}} \sum_{y_j=0}^{2^b-1} e^{i2\pi BG_\beta(\mathbf{x}+r\mathbf{y})}|y_j\rangle$ and $n \in \mathbb{N}$

1: Sample a uniformly random $n$-digit binary number $u \in [0,1)$ and define $\xi = \frac{2\pi u}{2^b}$
2: Apply multi-phase gate $\sum_{j=0}^{2^{b-1}} e^{-i\langle \xi,j \rangle}|j\rangle\langle j|$ to $|\psi\rangle$
3: Perform inverse Fourier transform over $\mathbb{Z}_M$ and measure the statement, yielding outcome $j$
4: **return** $\varphi = \frac{2\pi j}{2^b} + \xi$

---

**Algorithm 4**

---

**Input:** $2m+1$ copeis of $|\psi\rangle = \frac{1}{\sqrt{2^{bd}}} \sum_{\mathbf{y=0}}^{\mathbf{2^b-1}} e^{i2\pi BG_\beta(\mathbf{x}+r\mathbf{y})}|\mathbf{y}\rangle$

1: **while** for each $s \in [d]$ **do**
2:     **for** $j \in [2m+1]$ **do**
3:         Suppressed-Bias Phase Estimation($|\psi_s(j)\rangle$,$n$), where $|\psi_s(j)\rangle$ the $s$-th register of the $j$-th copy of $|\psi\rangle$
4:     **end for**
5:     **for** $j \in [2m+1]$ **do**
6:         Compute $d_j$ the $m$-th smallest distance in the (multi)set $\{|\varphi_j - \varphi_k|_{2\pi} : k \in [2m+1]\setminus\{j\}\}$

7:         Define $w_j = e^{-\frac{m2^b}{4}d_j}$
8:     **end for**
9:     $\bar{\varphi}_s = \varphi_j$ with probability $\frac{2_j}{\sum_{j\in[2m+1]} w_j}$
10: **end while**
11: **return** $\bar{\varphi} = (\bar{\varphi}_1,...,\bar{\varphi}_d)$

---

*Proof of Theorem 2.* In the finite precision scenario, i.e., with a finite number of qubits, we can only query the $\beta$ approximation function value $|G_\beta(\mathbf{y}) - G(\mathbf{y})| \leq \beta$. Consequently, we only have $|\psi\rangle = \frac{1}{\sqrt{B^d}} \sum_{\mathbf{y}\in G_b^d} e^{2\pi i BG_\beta(\mathbf{x}+r\mathbf{y})}|\mathbf{y}\rangle$. Next, we establish the relationship between $\beta$ and $\delta$. First, for $L_0$ Lipschitz continuous and $L_1$ smooth functions, we have $\|\nabla F(\mathbf{x})\| \leq L_0$. This implies $\left\|\frac{\nabla F(\mathbf{x})}{3L_0}\right\|_\infty \leq \frac{1}{3}$. In light of the above theorem, we have:

$$\left\||\psi\rangle - \frac{1}{\sqrt{B}} \sum_{\mathbf{y}\in G_b^d} e^{2\pi iB\langle \frac{\nabla F(\mathbf{x})}{3L_0}, \mathbf{y}\rangle}|\mathbf{y}\rangle\right\|_2 = \left\|\frac{1}{\sqrt{B}} \sum_{\mathbf{y}\in G_b^d} e^{2\pi i\frac{BG_\beta(\mathbf{x}+r\mathbf{y})}{3rL_0}}|\mathbf{y}\rangle - \frac{1}{\sqrt{B}} \sum_{\mathbf{y}\in G_b^d} e^{2\pi iB\langle \frac{\nabla F(\mathbf{x})}{3L_0}, \mathbf{y}\rangle}|\mathbf{y}\rangle\right\|_2$$

$$= \left\|\frac{1}{\sqrt{B}} \sum_{\mathbf{y}\in G_b^d} \left[e^{2\pi i\frac{BG_\beta(\mathbf{x}+r\mathbf{y})}{3rL_0}} - e^{2\pi iB\langle \frac{\nabla F(\mathbf{x})}{3L_0}, \mathbf{y}\rangle}\right]|\mathbf{y}\rangle\right\|_2 \leq \frac{\sqrt{B^d}}{\sqrt{B^d}} \left\|\sum_{\mathbf{y}\in G_b^d} \left[e^{2\pi i\frac{BG_\beta(\mathbf{x}+r\mathbf{y})}{3rL_0}} - e^{2\pi iB\langle \frac{\nabla F(\mathbf{x})}{3L_0}, \mathbf{y}\rangle}\right]|\mathbf{y}\rangle\right\|_\infty$$

$$\leq \max_{\mathbf{y}\in G_b^d} \left|e^{2\pi iB\frac{G_\beta(\mathbf{x}+r\mathbf{y})}{3rL_0}} - e^{2\pi iB\langle \frac{\nabla F(\mathbf{x})}{3L_0}, \mathbf{y}\rangle}\right| \leq \frac{2\pi B}{3rL_0} \max_{\mathbf{y}\in G_b^d} |G_\beta(\mathbf{x}+r\mathbf{y}) - \langle \nabla F(\mathbf{x}), r\mathbf{y}\rangle|$$

$$\leq \frac{2\pi B}{3rL_0} \max_{\mathbf{y}\in G_b^d} |G_\beta(\mathbf{x}+r\mathbf{y}) - G(\mathbf{x}+r\mathbf{y})| + |G(\mathbf{x}+r\mathbf{y}) - \langle \nabla F(\mathbf{x}), r\mathbf{y}\rangle|$$

$$\leq 2\pi B\frac{\beta + \frac{L_1}{2}dr^2}{3rL_0}$$

If we choose $\beta = \frac{L_1dr^2}{2}$, then we have:

$$2\pi B\frac{\beta + \frac{L_1}{2}dr^2}{3rL_0} = \frac{4\pi B\beta}{3\sqrt{\frac{2\beta}{L_1d}}L_0} \leq \frac{\delta}{24\lceil\ln\frac{6d}{\delta}\rceil + 3}$$

Therefore, if we choose:

$$\beta \leq \frac{L_0^2\delta^2}{8\pi^2 B^2 L_1 d(8\lceil\ln\frac{6d}{\delta}\rceil + 1)^2}$$

We can compute $\mathbf{k} \in [-\frac{1}{2}, \frac{1}{2}]^d$ satisfying

$$Pr\left[\left\|\mathbf{k} - \frac{\nabla F(\mathbf{x})}{3L_0}\right\|_\infty \geq \epsilon\right] \leq \delta, \quad \left\|\mathbb{E}[\mathbf{k}] - \frac{\nabla F(\mathbf{x})}{3L_0}\right\|_\infty \leq \delta$$

Next, we show the variance. If we choose $\epsilon = \delta = \frac{\sigma}{6L_0}$, then, we have:

$$\mathbb{E}\left[\left\|\mathbf{k} - \frac{\nabla F(\mathbf{x})}{3L_0}\right\|_\infty\right] \leq Pr\left[\epsilon \leq \left\|\mathbf{k} - \frac{\nabla F(\mathbf{x})}{3L_0}\right\|_\infty\right] \times \frac{5}{6} + Pr\left[\left\|\mathbf{k} - \frac{\nabla F(\mathbf{x})}{3L_0}\right\|_\infty \leq \epsilon\right] \times \epsilon$$

$$= \frac{5\delta}{6} + \epsilon(1-\delta) \leq \epsilon + \delta = \frac{\sigma}{3L_0}$$

Therefore, we have

$$\mathbb{E}\left[\|\mathbf{k} - \nabla F(\mathbf{x})\|_\infty\right] \leq \sigma$$

$\square$

The above proof demonstrates that the algorithm designed using quantum gradient estimation exhibits the following total variance in each iteration. Using this variance, we can derive the complexity of quantum zero-order algorithm.

$$\mathbb{E}[A(\mathbf{x}(t_j), t_j)] = \mathbb{E}\left[(m+M)\left\|\nabla F(\mathbf{x}(\frac{j}{N}))\right\|_2 + \eta_b(t_j) + \eta_n(t_j)\right] \leq \sigma$$

*Proof of Theorem 3.* By substituting the above total variance for the stochastic biased gradient into Section 3.2, we obtain the iteration complexity of the quantum zero-order algorithm as follows, where we restrict both the first term and the second term to be $\frac{\epsilon}{2}$.

**Convex constraint.**

$$\begin{cases} \sum_{j=0}^{N-1} \frac{2e^{\frac{j+1}{N}}\sqrt{D}}{eN}\mathbb{E}[A(\mathbf{x}(t_j), t_j)] + \frac{L_1 D}{N} & \text{if monotone} \\ \sum_{j=0}^{N-1} 2\frac{j+N}{N^2}\sqrt{D}\mathbb{E}[A(\mathbf{x}(t_j), t_j)] + \frac{L_1}{2}\frac{1}{N}D & \text{if non-monotone} \end{cases}$$

$$\leq \begin{cases} \frac{2(e-1)(N+1)\sigma\sqrt{Dd}}{N} + \frac{L_1 D}{N} \\ \frac{(N+1)\sigma\sqrt{Dd}}{N} + \frac{L_1}{2}\frac{1}{N}D \end{cases} = O(\epsilon) \Rightarrow \begin{cases} \sigma = O(\frac{L_1 D\epsilon}{2(e-1)\sqrt{Dd}(2L_1 D+\epsilon)}), N = O(\frac{L_1 D}{\epsilon}) & \text{if monotone} \\ \sigma = O(\frac{L_1 D\epsilon}{2\sqrt{Dd}(L_1 D+\epsilon)}), N = O(\frac{L_1 D}{\epsilon}) & \text{if non-monotone} \end{cases}$$

**Down-closed and convex constraints**

$$\sum_{j=0}^{0.372N-1} \frac{2e^{\frac{j+1}{eN}}\sqrt{D}}{N}\mathbb{E}[A(\mathbf{x}(t_j), t_j)] + \sum_{j=0.372N}^{N-1} 2\frac{j+1}{eN^2}\sqrt{D}\mathbb{E}[A(\mathbf{x}(t_j), t_j)] + \frac{L_1 D(e-1)}{2eN}$$

$$\leq \frac{2(e^{0.372}-1)(N+1)\sigma\sqrt{Dd}}{N} + \frac{0.628(1.372N+1)\sigma\sqrt{Dd}}{N} + \frac{L_1 D(e-1)}{2N}$$

$$\leq \frac{3(N+1)\sigma\sqrt{Dd}}{N} + \frac{L_1 D(e-1)}{2N} = O(\epsilon)$$

$$\Downarrow$$

$$\sigma = O(\frac{L_1 D(e-1)\epsilon}{6\sqrt{Dd}(L_1 D(e-1)+\epsilon)}), N = O(\frac{L_1 D(e-1)}{\epsilon})$$

**Convex constraints with the largest element**

$$\sum_{j=0}^{N-1} \frac{2e^{\frac{j}{N}}\sqrt{D}}{N}\mathbb{E}[A(\mathbf{x}(t_j), t_j)] + \frac{L_1 D(e-1)}{2N} \leq \frac{2(N+1)\sigma\sqrt{Dd}}{N} + \frac{L_1 D(e-1)}{2N} = O(\epsilon)$$

$$\Downarrow$$

$$\sigma = O(\frac{(e-1)L_1 D\epsilon}{4\sqrt{Dd}((e-1)L_1 D+\epsilon)}), N = O(\frac{L_1 D(e-1)}{\epsilon})$$

$\square$

## C.2 CLASSICAL ZERO-ORDER ALGORITHM

The smoothing technique involving sampling over a unit ball centered at a given point is widely adopted in zero-order algorithms. While sampling over a unit sphere centered at the given point yields an unbiased gradient estimation for smoothing function $\hat{F}$, it is a biased estimation for $\nabla F(\mathbf{x})$. Next, we illustrate the bias and the noise can be bounded as follows:

*Proof of Lemma 2.* If $F$ is $L_0$-continuous, we have

$$|\hat{F}(\mathbf{x}) - \hat{F}(\mathbf{y})| = \mathbb{E}_{\mathbf{u}\sim\mathbb{B}}[F(\mathbf{x} + r\mathbf{u}) - F(\mathbf{y} + r\mathbf{u})] \le \mathbb{E}_{\mathbf{u}\sim\mathbb{B}}[L_0 \|\mathbf{x} - \mathbf{y}\|_2] = L_0 \|\mathbf{x} - \mathbf{y}\|_2$$

It implies that $\hat{F}(\mathbf{x})$ is $L_0$-continuous. In addition, combining the $L_0$-continuous and the radius $r$, the difference between $\hat{F}(\mathbf{x})$ and $F(\mathbf{x})$ can be bounded as follows:

$$|\hat{F}(\mathbf{x}) - F(\mathbf{x})| = |\mathbb{E}_{\mathbf{u}\sim\mathbb{B}}[F(\mathbf{x} + r\mathbf{u}) - F(\mathbf{x})]| \le \mathbb{E}_{\mathbf{u}\sim\mathbb{B}}[L_0 r \|\mathbf{u}\|_2] \le L_0 r$$

When $F$ is $L_1$-smooth, the following inequality implies $\hat{F}$ is also $L_1$-smooth:

$$\left\|\nabla\hat{F}(\mathbf{x}) - \nabla\hat{F}(\mathbf{y})\right\|_2 = \|\mathbb{E}_{\mathbf{u}\sim\mathbb{S}}[\nabla F(\mathbf{x} + r\mathbf{u}) - \nabla F(\mathbf{y} + r\mathbf{u})]\|_2 = \mathbb{E}_{\mathbf{u}\sim\mathbb{S}}[L_1 \|\mathbf{x} - \mathbf{y}\|_2] \le L_1 \|\mathbf{x} - \mathbf{y}\|_2$$

The above gradient estimation corresponds to a single-point estimation scheme. Since Shamir (2017) demonstrated that two-point estimation yields a smaller variance, next, we prove the bias $\mathbf{b}(\mathbf{x})$ and the noise $\mathbf{n}(\mathbf{x})$ of the two-point estimation can be bounded as follows:

$$
\begin{aligned}
\|\mathbf{b}(\mathbf{x})\|_2 = \left\|\nabla\hat{F}(\mathbf{x}) - \nabla F(\mathbf{x})\right\|_2 &= \left\|\frac{d}{2r}\mathbb{E}_{\mathbf{u}\sim\mathbb{S}}[(F(\mathbf{x} + r\mathbf{u}) - F(\mathbf{x} - r\mathbf{u}))\mathbf{u}] - \nabla F(\mathbf{x})\right\|_2 \\
&= \left\|\frac{d}{2r}\mathbb{E}_{\mathbf{u}\sim\mathbb{S}}[(F(\mathbf{x} + r\mathbf{u}) - F(\mathbf{x} - r\mathbf{u}))\mathbf{u} - \langle\nabla F(\mathbf{x}), 2r\mathbf{u}\rangle\mathbf{u}]\right\|_2 \\
&\le \frac{d}{2r}\mathbb{E}_{\mathbf{u}\sim\mathbb{S}}[\|(F(\mathbf{x} + r\mathbf{u}) - F(\mathbf{x} - r\mathbf{u}))\mathbf{u} - \langle\nabla F(\mathbf{x}), 2r\mathbf{u}\rangle\mathbf{u}\|_2] \\
&= \frac{d}{2r}\mathbb{E}_{\mathbf{u}\sim\mathbb{S}}[|(F(\mathbf{x} + r\mathbf{u}) - F(\mathbf{x} - r\mathbf{u})) - \langle\nabla F(\mathbf{x}), 2r\mathbf{u}\rangle| \|\mathbf{u}\|_2] \\
&= \frac{d}{2r}\mathbb{E}_{\mathbf{u}\sim\mathbb{S}}[|\int_0^1 \langle\nabla(F(\mathbf{x} - r\mathbf{u} + 2tr\mathbf{u})) - \nabla F(\mathbf{x}), 2r\mathbf{u}\rangle dt| \|\mathbf{u}\|_2] \\
&\le d\mathbb{E}_{\mathbf{u}\sim\mathbb{S}}[\int_0^1 \|\nabla(F(\mathbf{x} - r\mathbf{u} + 2tr\mathbf{u})) - \nabla F(\mathbf{x})\|_2 dt] \\
&\le dL_1 r\mathbb{E}_{\mathbf{u}\sim\mathbb{S}}[\|\mathbf{u}\|_2]\int_0^1 |2t - 1|dt \le \frac{dL_1 r}{2}
\end{aligned}
$$

The second equality comes from $\mathbb{E}_{\mathbf{u}\sim\mathbb{S}}[\langle\nabla F(\mathbf{x}), \mathbf{u}\rangle\mathbf{u}] = \mathbb{E}_{\mathbf{u}\sim\mathbb{S}}[\mathbf{u}\mathbf{u}^T]\nabla F(\mathbf{x}) = \frac{1}{d}\mathbf{I}\nabla F(\mathbf{x})$. The third inequality comes from $L_1$-smooth.

$$
\begin{aligned}
\mathbb{E}\left[\|\mathbf{n}(\mathbf{x})\|_2^2\right] &= \mathbb{E}\left[\left\|\nabla\hat{F}(\mathbf{x}) - \mathbb{E}[\nabla\hat{F}(\mathbf{x})]\right\|_2^2\right] \\
&= \mathbb{E}\left[\langle\nabla\hat{F}(\mathbf{x}) - \mathbb{E}[\nabla\hat{F}(\mathbf{x})], \nabla\hat{F}(\mathbf{x}) - \mathbb{E}[\nabla\hat{F}(\mathbf{x})]\rangle\right] \\
&= \mathbb{E}\left[\left\|\nabla\hat{F}(\mathbf{x})\right\|_2^2\right] - \left\|\mathbb{E}[\nabla\hat{F}(\mathbf{x})]\right\|_2^2 \le \mathbb{E}\left[\left\|\nabla\hat{F}(\mathbf{x})\right\|_2^2\right] \le 16\sqrt{2\pi}dL_0^2
\end{aligned}
$$

The last inequality comes from (Lin et al., 2022). □

Using the bias and noise results from Lemma 2, we can obtain the total variance for the gradient estimation when employing the smoothing technique.

$$
\begin{aligned}
\mathbb{E}[A(\mathbf{x}(t_j), t_j)] = \mathbb{E}\|\mathbf{g}(\mathbf{x}(t_j)) - \nabla F(\mathbf{x}(t_j))\|_2 &= \mathbb{E}\|\mathbf{n}(\mathbf{x}(t_j)) + \mathbf{b}(\mathbf{x}(t_j))\|_2 \\
&\le \mathbb{E}\|\mathbf{n}(\mathbf{x}(t_j))\|_2 + \mathbb{E}\|\mathbf{b}(\mathbf{x}(t_j))\|_2 \le \frac{dL_1 r}{2} + \sqrt{16\sqrt{2\pi}dL_0^2}
\end{aligned}
$$

---

**Algorithm 5** The Estimation of Gradient with Variance Reduction

---

**Input:** Step-size $\rho_{t_j} = \frac{4}{(j+s)^{2/3}}$, Current gradient $\mathbf{d}_{t_{j-1}}$ and solution $\mathbf{x}(t_j)$

1: $\mathbf{g}(\mathbf{x}(t_j))$ is the gradient of $\mathbf{x}(t_j)$ returned by the stochastic biased gradient oracle
2: $\mathbf{d}_{t_j} = (1 - \rho_{t_j})\mathbf{d}_{t_{j-1}} + \rho_{t_j}\mathbf{g}(\mathbf{x}(t_j))$
3: **return** $\mathbf{d}_{t_j}$

---

When we directly employ the above gradient estimation to design algorithm, the total variance cannot be controlled by the algorithm behavior. To reduce the variance, we introduce the momentum term that was mentioned in (Mokhtari et al., 2020). Specifically, we initialize the momentum term $\mathbf{d}_{t_0} = \mathbf{0}$. In each iteration, the discrete time algorithm adopt the gradient with momentum as follows:

$$\mathbf{d}_{t_j} = (1 - \rho_{t_j})\mathbf{d}_{t_{j-1}} + \rho_{t_j}\mathbf{g}(\mathbf{x}(t_j))$$

where the discrete time is $t_j = \frac{j}{N}$, $\mathbf{g}(\mathbf{x}(t_j))$ is the gradient estimation obtained via the smoothing technique at $\mathbf{x}(t_j)$, and $\rho_{t_j} = \frac{4}{(j+s)^{2/3}}$. Through the above procedure, the variance between the $\nabla F(\mathbf{x}(t_j))$ and $\mathbf{d}_{t_j}$ decreases as the number of iterations increases. While Chen et al. (2018); Mokhtari et al. (2020) proved the upper bound for the total variance, their proof relies on the assumption that $\mathbf{g}(\mathbf{x}(t_j))$ constitutes an unbiased estimation of $\nabla F(\mathbf{x}(t_j))$. By combining the Lemma 2, which illustrates the bounded bias and the noise for the $\mathbf{g}(\mathbf{x}(t_j))$, next, we prove the variance reduction statement for the biased gradient estimation.

*Proof of Theorem 4.*

$$\mathbb{E}[\|\nabla F(\mathbf{x}(t_j)) - \mathbf{d}_{t_j}\|_2^2] = \mathbb{E}[\|\nabla F(\mathbf{x}(t_j)) - (1 - \rho_{t_j})\mathbf{d}_{t_{j-1}} - \rho_{t_j}\mathbf{g}(\mathbf{x}(t_j))\|_2^2]$$

$$= \mathbb{E}[\|\rho_{t_j}[\nabla F(\mathbf{x}(t_j)) - \mathbf{g}(\mathbf{x}(t_j))] + (1 - \rho_{t_j})[\nabla F(\mathbf{x}(t_{j-1})) - \mathbf{d}_{t_{j-1}}] + (1 - \rho_{t_j})[\nabla F(\mathbf{x}(t_j)) - \nabla F(\mathbf{x}(t_{j-1}))]\|_2^2]$$

$$= \rho_{t_j}^2 \mathbb{E}[\|\nabla F(\mathbf{x}(t_j)) - \mathbf{g}(\mathbf{x}(t_j))\|^2] + (1 - \rho_{t_j})^2 \|\nabla F(\mathbf{x}(t_{j-1})) - \mathbf{d}_{t_{j-1}}\|^2 + (1 - \rho_{t_j})^2 \|\nabla F(\mathbf{x}(t_j)) - \nabla F(\mathbf{x}(t_{j-1}))\|^2$$

$$+ 2(1 - \rho_{t_j})^2 \langle \nabla F(\mathbf{x}(t_j)) - \nabla F(\mathbf{x}(t_{j-1})), \nabla F(\mathbf{x}(t_{j-1})) - \mathbf{d}_{t_{j-1}} \rangle$$

$$+ 2(1 - \rho_{t_j})\rho_{t_j} \langle \mathbf{b}(\mathbf{x}(t_j)), \nabla F(\mathbf{x}(t_{j-1})) - \mathbf{d}_{t_{j-1}} \rangle + 2(1 - \rho_{t_j})\rho_{t_j} \langle \mathbf{b}(\mathbf{x}(t_j)), \nabla F(\mathbf{x}(t_j)) - \nabla F(\mathbf{x}(t_{j-1})) \rangle$$

$$\leq \rho_{t_j}^2 \|\nabla F(\mathbf{x}(t_j)) - \mathbf{g}(\mathbf{x}(t_j))\|_2^2 + (1 - \rho_{t_j})^2 \|\nabla F(\mathbf{x}(t_{j-1})) - \mathbf{d}_{t_{j-1}}\|_2^2 + (1 - \rho_{t_j})^2 \|\nabla F(\mathbf{x}(t_j)) - \nabla F(\mathbf{x}(t_{j-1}))\|_2^2$$

$$+ (1 - \rho_{t_j})^2[\beta_1(t_j) \|\nabla F(\mathbf{x}(t_{j-1})) - \mathbf{d}_{t_{j-1}}\|_2^2 + \frac{L_1^2 D}{\beta_1(t_j)N^2}]$$

$$+ (1 - \rho_{t_j})\rho_{t_j}[\beta_2(t_j) \|\nabla F(\mathbf{x}(t_{j-1})) - \mathbf{d}_{t_{j-1}}\|_2^2 + \frac{\|\mathbf{b}(\mathbf{x}(t_j))\|_2^2}{\beta_2(t_j)}]$$

$$+ 2(1 - \rho_{t_j})\rho_{t_j}\frac{L_1\sqrt{D}}{N}\|\mathbf{b}(\mathbf{x}(t_j))\|_2$$

The inequality comes from the Young's inequality, i.e., $2\langle \mathbf{a}, \mathbf{b} \rangle \leq \beta \|\mathbf{a}\|^2 + \|\mathbf{b}\|^2/\beta$. In addition, we have $\|\nabla F(\mathbf{x}(t_j)) - \nabla F(\mathbf{x}(t_{j-1}))\|_2 \leq L_1 \|\mathbf{x}(t_j) - \mathbf{x}(t_{j-1})\|_2 \leq \frac{L_1\sqrt{D}}{N}$, when $\|\mathbf{x}(t_j) - \mathbf{x}(t_{j-1})\|_2 \leq \frac{\sqrt{D}}{N}$. Setting $\beta_1(t_j) = \frac{\rho_{t_j}}{4}, \beta_2(t_j) = \frac{1}{4}$, we have

$$\begin{cases} (1 - \rho_{t_j})[(1 - \rho_{t_j})(1 + \beta_1(t_j)) + \rho_{t_j}\beta_2(t_j)] \leq 1 - \frac{\rho_{t_j}}{2} \\ (1 - \rho_{t_j})^2(1 + \frac{1}{\beta_1(t_j)})\frac{L_1^2 D}{N^2} \leq (1 + \frac{1}{\beta_1(t_j)})\frac{L_1^2 D}{N^2} \leq (1 + (j+s)^{2/3})\frac{4L_1^2 D}{(j+s)^2} \leq \frac{5L_1^2 D}{(j+s)^{4/3}} \\ (1 - \rho_{t_j})\rho_{t_j}\frac{\|\mathbf{b}(\mathbf{x}(t_j))\|_2^2}{\beta_2(t_j)} + 2(1 - \rho_{t_j})\rho_{t_j}\frac{L_1\sqrt{D}}{N}\|\mathbf{b}(\mathbf{x}(t_j))\|_2 \leq \frac{8L_1\sqrt{D}\|\mathbf{b}(\mathbf{x}(t_j))\|_2 + 16(j+s)^{2/3}\|\mathbf{b}(\mathbf{x}(t_j))\|_2^2}{(t+s)^{4/3}} \end{cases}$$

The above inequalities are because of $j \leq N$ and $8 \leq s \leq N$. Setting:

$$Q(t_j) = \max \Big\{ \|\nabla F(\mathbf{x}(t_0)) - \mathbf{d}_{t_0}\|_2^2 (s + 1)^{2/3},$$

$$5L_1^2 D + 16 \max_{k \in [j]} A^2(\mathbf{x}(t_k), t_k) + 8L_1\sqrt{D} \max_{k \in [j]} \|\mathbf{b}(\mathbf{x}(t_k))\|_2 + 16(t + s)^{2/3} \max_{k \in [j]} \|\mathbf{b}(\mathbf{x}(t_k))\|_2^2 \Big\}$$

Clearly, $Q(t_{j-1})$ is non-decreasing, i.e., $Q(t_{j-1}) \leq Q(t_j)$. If $\|\mathbf{b}(\mathbf{x}(t))\|_2^2 = 0$, it is degenerated to Chen et al. (2018); Mokhtari et al. (2020). Next, we adopt the induction to prove the variance

$$\mathbb{E}[\|\nabla F(\mathbf{x}(t_j)) - \mathbf{d}_{t_j}\|_2^2] \leq \frac{Q(t_j)}{(j + s + 1)^{2/3}} \tag{23}$$

Considering $t_0$, we have

$$(s + 1)^{2/3}\mathbb{E}[\|\nabla F(\mathbf{x}(t_0)) - \mathbf{d}_{t_0}\|_2^2] \leq Q(t_0)$$

The above inequality implies Eq.(23) holds. Assume the Eq.(23) for the variance holds when $t_{k-1}$:

$$\mathbb{E}[\|\nabla F(\mathbf{x}(t_k)) - \mathbf{d}_{t_k}\|_2^2] \leq (1 - \frac{2}{(k + s)^{2/3}})\mathbb{E}[\|\nabla F(\mathbf{x}(t_{k-1})) - \mathbf{d}_{t_{k-1}}\|_2^2] + \frac{Q(t_{k-1})}{(k + s)^{4/3}}$$

$$\leq (1 - \frac{2}{(k + s)^{2/3}})\frac{Q(t_{k-1})}{(k + s)^{2/3}} + \frac{Q(t_{k-1})}{(k + s)^{4/3}}$$

$$\leq (1 - \frac{2}{(k + s)^{2/3}})\frac{Q(t_{k-1})}{(k + s)^{2/3}} + \frac{Q(t_{k-1})}{(k + s)^{4/3}}$$

$$\leq \frac{((k + s)^{2/3} - 1)Q(t_k)}{(k + s)^{4/3}} \leq \frac{Q(t_k)}{(k + s)^{2/3} + 1} \leq \frac{Q(t_k)}{(k + s + 1)^{2/3}}$$

Therefore, the proof for Eq.(23) is complete. $\qquad\square$

Using Theorem 4, we can obtain the variance between the gradient in algorithm and the original gradient $\nabla F(\mathbf{x}(t_j))$. For the convenience of algorithmic analysis, we further rewrite the aforementioned inequality as follows, where $q(t_j) = \max\{\|\nabla F(\mathbf{x}(t_0)) - \mathbf{d}_{t_0}\|_2 9^{1/3}, \sqrt{5D}L_1 + 4\max_{k \in [j]} A(t_k) + \sqrt{8L_1\sqrt{D}\max_{k \in [j]} \|\mathbf{b}(\mathbf{x}(t_k))\|_2}\}$.

$$\mathbb{E}[A(\mathbf{x}(t_j), t_j)] = \mathbb{E}\left[(m + M)\left\|\nabla F(\mathbf{x}(\frac{j}{N}))\right\|_2 + \eta_b(t_j) + \eta_n(t_j)\right] \leq \frac{\sqrt{Q(t_j)}}{(j + 9)^{1/3}}$$

$$\leq \max\Big\{\frac{\|\nabla F(\mathbf{x}(t_0)) - \mathbf{d}_{t_0}\|_2 9^{1/3}}{(j + 9)^{1/3}},$$

$$\frac{\sqrt{5D}L_1 + 4\max_{k \in [j]} A(\mathbf{x}(t_k), t_k) + \sqrt{8L_1\sqrt{D}\max_{k \in [j]} \|\mathbf{b}(\mathbf{x}(t_k))\|_2} + 4(j + 8)^{1/3}\max_{k \in [j]} \|\mathbf{b}(\mathbf{x}(t_k))\|_2}{(j + 9)^{1/3}}\Big\}$$

$$\leq \max\Big\{\frac{\|\nabla F(\mathbf{x}(t_0)) - \mathbf{d}_{t_0}\|_2 9^{1/3}}{(j + 9)^{1/3}},$$

$$\frac{\sqrt{5D}L_1 + 4\max_{k \in [j]} A(\mathbf{x}(t_k), t_k) + \sqrt{8L_1\sqrt{D}\max_{k \in [j]} \|\mathbf{b}(\mathbf{x}(t_k))\|_2}}{(j + 9)^{1/3}} + 4\max_{k \in [j]} \|\mathbf{b}(\mathbf{x}(t_k))\|_2\Big\}$$

$$\leq \max\Big\{\frac{\|\nabla F(\mathbf{x}(t_0)) - \mathbf{d}_{t_0}\|_2 9^{1/3}}{(j + 9)^{1/3}}, \frac{\sqrt{5D}L_1 + 4\max_{k \in [j]} A(\mathbf{x}(t_k), t_k) + \sqrt{8L_1\sqrt{D}\max_{k \in [j]} \|\mathbf{b}(\mathbf{x}(t_k))\|_2}}{(j + 9)^{1/3}}\Big\}$$

$$+ 4\max_{k \in [j]} \|\mathbf{b}(\mathbf{x}(t_k))\|_2$$

$$= \frac{q(t_j)}{(j + 9)^{1/3}} + 4\max_{k \in [j]} \|\mathbf{b}(\mathbf{x}(t_k))\|_2$$

When we adopt the smoothing technique in Section 4.2, we have:

$$q(t_j) = \max\{\|\nabla F(\mathbf{x}(t_0)) - \mathbf{d}_{t_0}\|_2 9^{1/3}, \sqrt{5D}L_1 + 2dL_1 r + 16L_0\sqrt{\sqrt{2\pi}d} + \sqrt{4L_1^2 dr\sqrt{D}}\} = Q$$

The above inequality shows that $q(t_j)$ is independent of the discrete time. For notational convenience, we rewrite $q(t_j)$ as $Q$, which implies the total variance is given by:

$$\mathbb{E}[A(\mathbf{x}(t_j), t_j)] = \mathbb{E}\left[(m + M)\left\|\nabla F(\mathbf{x}(\frac{j}{N}))\right\|_2 + \eta_b(t_j) + \eta_n(t_j)\right] \leq \frac{Q}{(j + 9)^{1/3}} + 2dL_1 r$$

By substituting the above total variance into the algorithmic analysis, we can derive the iteration complexity of the classical zero-order algorithm as follows:

*Proof of Theorem 5.* Since the term $2dL_1r$ in the total variance is independent in the iteration step, it can only be reduced by controlling the radius $r$ in the algorithm design. Therefore, we restrict the first term and the third term in the following inequalities are $\frac{\epsilon}{2}$, respectively, which can be reduced by the iteration step.

**Convex constraints.**

$$\begin{cases} \sum_{j=0}^{N-1} \frac{2e^{\frac{j+1}{N}}\sqrt{D}}{N}\mathbb{E}[A(\mathbf{x}(t_j),t_j)] + \frac{L_1D}{N} & \text{if monotone} \\ \sum_{j=0}^{N-1} 2\frac{j+1}{N^2}\sqrt{D}\mathbb{E}[A(\mathbf{x}(t_j),t_j)] + \frac{L_1}{2}\frac{1}{N}D & \text{if non-monotone} \end{cases}$$

$$\leq \begin{cases} \sum_{j=0}^{N-1} \frac{2e^{\frac{j+1}{N}}\sqrt{D}}{N}\frac{\sqrt{Q(t_j)}}{(j+9)^{1/3}} + \frac{L_1D}{N} & \text{if monotone} \\ \sum_{j=0}^{N-1} 2\frac{j+1}{N^2}\sqrt{D}\frac{\sqrt{Q(t_j)}}{(j+9)^{1/3}} + \frac{L_1}{2}\frac{1}{N}D & \text{if non-monotone} \end{cases}$$

$$\leq \begin{cases} \sum_{j=0}^{N-1} \frac{2e\sqrt{D}}{N}[\frac{Q}{(j+9)^{1/3}} + 2dL_1r] + \frac{L_1D}{N} & \text{if monotone} \\ \sum_{j=0}^{N-1} 2\frac{1}{N}\sqrt{D}[\frac{Q}{(j+9)^{1/3}} + 2dL_1r] + \frac{L_1}{2}\frac{1}{N}D & \text{if non-monotone} \end{cases}$$

$$\leq \begin{cases} \frac{6e\sqrt{D}QN^{2/3}}{N} + 4e\sqrt{D}dL_1r + \frac{L_1D}{N} & \text{if monotone} \\ \frac{6\sqrt{D}QN^{2/3}}{N} + 4\sqrt{D}dL_1r + \frac{L_1}{2}\frac{1}{N}D & \text{if non-monotone} \end{cases} = O(\epsilon) \Rightarrow N = O(\frac{1}{\epsilon^3})$$

**Down-closed and convex constraints.**

$$\sum_{j=0}^{0.372N-1} \frac{2e^{\frac{j+1}{N}}\sqrt{D}}{N}\mathbb{E}[A(\mathbf{x}(t_j),t_j)] + \sum_{j=0.372N}^{N-1} 2\frac{j+1}{N^2}\sqrt{D}\mathbb{E}[A(\mathbf{x}(t_j),t_j)] + \frac{L_1D(e-1)}{2N}$$

$$\leq \sum_{j=0}^{0.372N-1} \frac{2e^{0.372}\sqrt{D}}{N}\frac{\sqrt{Q(t_j)}}{(j+9)^{1/3}} + \sum_{j=0.372N}^{N-1} \frac{2\sqrt{D}}{N}\frac{\sqrt{Q(t_j)}}{(j+9)^{1/3}} + \frac{L_1D(e-1)}{2N}$$

$$\leq \frac{6e\sqrt{D}QN^{2/3}}{N} + 4e\sqrt{D}dL_1r + \frac{L_1D(e-1)}{2N} = O(\epsilon) \Rightarrow N = O(\frac{1}{\epsilon^3})$$

**Convex constraints with the largest elements.**

$$\sum_{j=0}^{N-1} \frac{2e^{\frac{j}{N}}\sqrt{D}}{N}\mathbb{E}[A(\mathbf{x}(t_j),t_j)] + \frac{L_1D(e-1)}{2N} \leq \sum_{j=0}^{N-1} \frac{2e\sqrt{D}}{N}\frac{\sqrt{Q(t_j)}}{(j+9)^{1/3}} + \frac{L_1D(e-1)}{2N}$$

$$\leq \frac{26e\sqrt{D}QN^{2/3}}{N} + 4e\sqrt{D}dL_1r + \frac{L_1D(e-1)}{2N} = O(\epsilon) \Rightarrow N = O(\frac{1}{\epsilon^3})$$

$\square$

# D NUMERICAL EXPERIMENTS

This section details the settings of test problems, the parameter configurations of experiments, and additional specifics regarding the numerical tests.

## D.1 SETTINGS

Continuous DR-submodular functions possess a wide range of applications in domains including machine learning, resource allocation, and among others. We employ the following examples to evaluate our algorithms.

- Non-convex/concave quadratic programming (NQP): This type of problem arises in many applications, such as demand forecasting (Ito & Fujimaki (2016)). The goal is to maximize a quadratic DR-function under some constraints. The formal statement of objective function is as follows:

$$F(\mathbf{x}) = \frac{1}{2}\mathbf{x}^T\mathbf{H}\mathbf{x} + \mathbf{h}^T\mathbf{x} + c$$

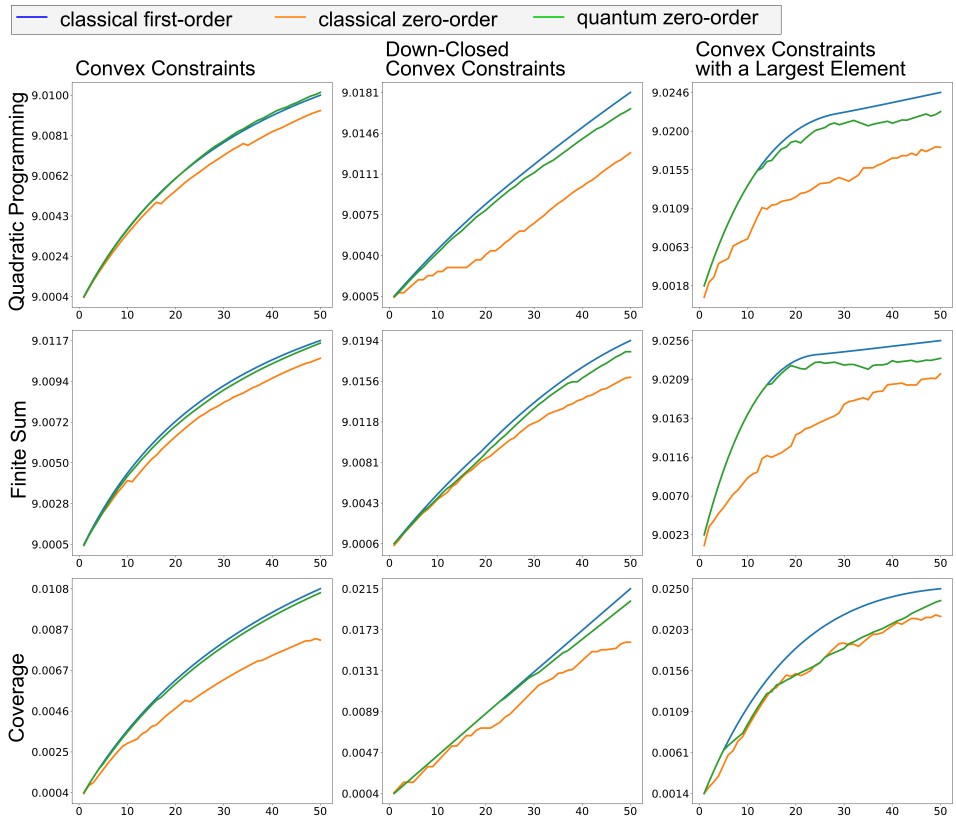

Figure 2: Comparison of classical first-order algorithm(blue), quantum zero-order algorithm(green), and classical zero-order algorithm(orange). The x-axis is the number of iterations. The y-axis is the function value.

In our numerical experiments, we choose $d = 3$ and consider a randomly generated matrix $\mathbf{H} \in \mathbb{R}^{d \times d}_{\leq 0}$ and vector $\mathbf{h} \in \mathbb{R}^d_{\geq 0}$. The constant $c$ is set to a sufficiently large value to guarantee that $F(\mathbf{x})$ is non-negative.

- Finite sums of DR-submodular quadratic programming: Building on the DR-submodular quadratic programming, we consider a finite-sum formulation that arises in applications like multi-resolution data summarization (Bian et al., 2017b). In our numerical experiments, the formal statement of the objective function is as follows:

$$F(\mathbf{x}) = \sum_{j=1}^{m} F_j(\mathbf{x}) = \sum_{j=1}^{m} \frac{1}{2}\mathbf{x}^T\mathbf{H}_j\mathbf{x} + \mathbf{h}_j^T\mathbf{x} + c_j$$

where we choose $m = 10$ and $d = 3$. The matrix $\mathbf{H}_j$, the vector $\mathbf{h}_j$, and the constant $c$ are generated in the same manner as in the DR-submodular quadratic programming case.

- Regular coverage maximization: A coverage function is a well-known monotone submodular set function, and its multilinear extension is a monotone DR-submodular function. In our numerical experiment, we consider a special case based on the regular coverage function from (Chen et al., 2023a). For a ground set $V = \{1, 2, ..., n\}$ and a subset collection $\{S_1, ..., S_n\}$ where $S_i \subseteq V$, the regular coverage function is defined as $f(X) = |\bigcup_{i \in X} S_i| - |X|$, whose multilinear extension is as follows:

$$F(\mathbf{x}) = k + 1 - (1 - x_{2k+1})\prod_{i=1}^{k}(1 - x_i) - (1 - x_{2k+1})\left(k - \sum_{i=1}^{k} x_i\right) - \sum_{i=1}^{k} x_i - x_{2k+1}$$

Since in our theoretical analysis, we consider three type constraints: general convex constraints, down-closed convex constraints, and convex constraints with the largest element, we consider the

following constraints in our numerical experiments that can be easily to generate the aforementioned three type constraints:

$$\mathcal{C} = \{\mathbf{x} \in [0,1]^d | \mathbf{A}\mathbf{x} \leq \mathbf{b}\}$$

If $\mathbf{A} \in \mathbb{R}^{c \times d}$ and $\mathbf{b} \in \mathbb{R}^c$, it is easy to generate an example such that $\mathcal{C}$ is general convex. When $\mathbf{A} \in \mathbb{R}_{\geq 0}^{c \times d}$ and $\mathbf{b} \in \mathbb{R}_{\geq 0}^c$, $\mathcal{C}$ is down-closed convex. If $\mathbf{A} \in \mathbb{R}_{\leq 0}^{c \times d}$ and $\mathbf{b} \in \mathbb{R}_{\leq 0}^c$, $\mathcal{C}$ is convex set with a largest element. All matrices and vectors here are randomly generated. Due to the high computational cost of classical simulation of quantum algorithms, we only set $d = 3$ on DR-submodular quadratic programming and its finite-sum and $k = 1$ on regular coverage maximization. For the number of the total iteration, we set $N = 50, 100$ respectively.

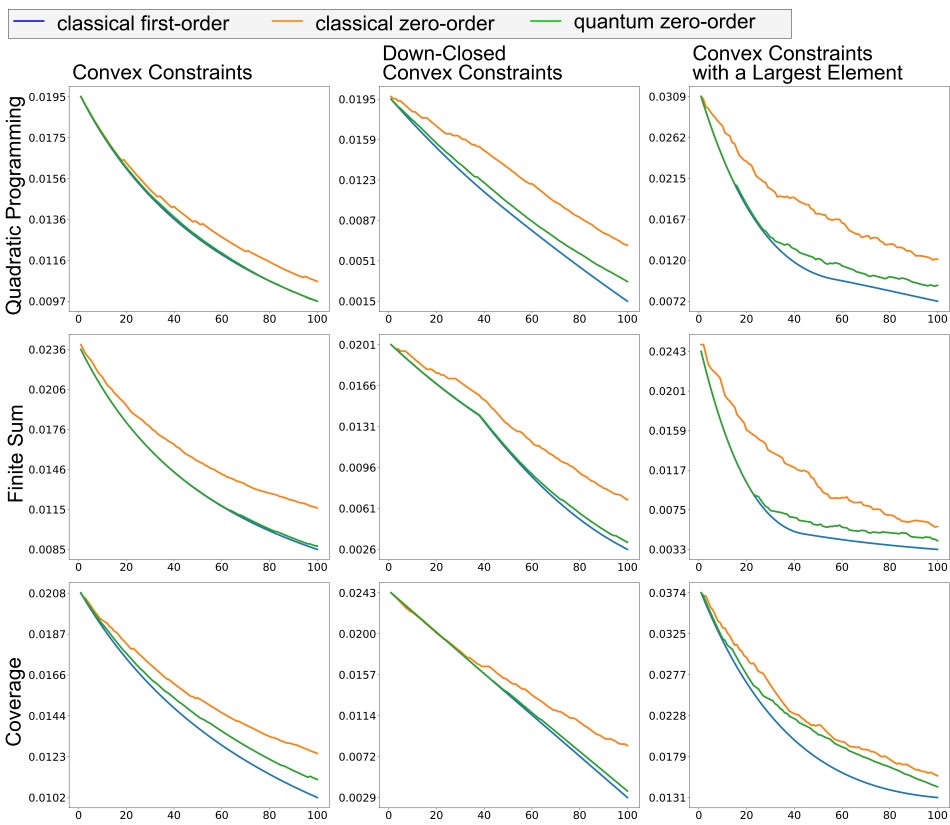

Figure 3: Comparison of classical first-order algorithm(blue), quantum zero-order algorithm(green), and classical zero-order algorithm(orange). The x-axis is the number of iterations. The y-axis is the difference between the function value and the baseline value.

## D.2 RESULTS

In this subsection, we present numerical experiments conducted with $N = 50$. Fig.2 depicts the quality of the current solution, exhibiting experimental phenomena consistent with those in Fig.1. Fig.3 illustrates the difference between the current solution and the baseline value.

For DR-submodular quadratic programming, the quality of the solution returned by quantum zero-order algorithm approximately approaches that of the solution returned by classical first-order algorithm. Although the quality returned by quantum zero-order algorithm has a slight degeneration under down-closed convex constraints and convex constraints with the largest element, it still significantly outperforms the classical zero-order. For the convergence rate, the quantum zero-order algorithm illustrate the significant acceleration for classical zero-order algorithm, achieving the approximate convergence rate for classical first-order algorithm.

For finite sums of DR-submodular quadratic programming, under both down-closed convex constraints and convex constraints with the largest element, the quantum zero-order algorithm demonstrates significant improvements in both solution quality and convergence rate compared to the clas-

sical zero-order algorithm. Moreover, its performance approaches that of the classical first-order algorithm across all constraint types.

When considering regular coverage maximization, the quantum zero-order algorithm performs excellently under general convex constraints and down-closed convex constraints, outperforming the classical zero-order algorithm and achieving performance close to that of the classical first-order algorithm. However, under convex constraints with the largest element, its performance degrades to the same level as the classical zero-order algorithm. This is primarily attributed to the excessive specificity of regular coverage maximization when the problem dimension is extremely small, i.e., $k = 1$. $V = \{1, 2, 3\}$, $S_1 = \{1, 3\}$, $S_2 = \{2\}$, $S_3 = \{1, 3\}$. Then, $f(\{1\}) = 1$, $f(\{2\}) = 0$, $f(\{3\}) = 1$, $f(\{1, 2\}) = 1$, $f(\{1, 3\}) = 0$, $f(\{2, 3\}) = 1$, $f(\{1, 2, 3\}) = 0$. According to results in DR-submodular quadratic programming, we believe that when the dimension is large, quantum zero-order algorithm can outperform classical zero-order algorithm both in the quality of solution and the convergence rate.

Totally, the numerical experiments with $N = 50$ exhibit similar phenomena to those with $N = 100$, confirming the quantum acceleration of the quantum zero-order algorithm relative to the classical zero-order algorithm.

