# OpenReview forum: "DR-Submodular Maximization with Stochastic Biased Gradients: Classical and Quantum Gradient Algorithms"
_ICLR.cc/2026/Conference — ICLR 2026 Poster_

### Official Review · Reviewer_3kKo · 2025-10-30

**Soundness:** 3
**Presentation:** 2
**Contribution:** 2
**Rating:** 6
**Confidence:** 2

**Summary:**

The paper studies a quite general optimization problem, namely maximizing a function F on [0,1]^n under some contraints defining a feasible set $\cal C \in [0,1]^n$. The function is DR-submodular. The feasible set is convex. Such a problem is typically solved using gradient ascend, and there is a huge literature on these techniques.

There are 3 contributions.

1. Extending the Lyapunov framework, to allow gradients to be imprecise, with a bias and a noise.
2. Proposition an 1/e approximation for non-monotone DR-submodular maximization over a convex set. The novelty is the assumption on a largest feasible point. This overcomes a 1/4 upper bound on the general optimization problem.
3. Providing a quantum algorithm, based on the improved quantum Jordan algorithm from 2023, which has the same performance guarantees as its classical counterpart but improves in cubic iteration time.

Some performance guarantees are proven, and experiments are conducted with standard benchmarks.

My background is too weak in this area to judge the results.

**Strengths:**

The domain is important and has a huge literature. The paper has 3 contributions, all of which seem central and important. The paper has a theoretical and a practical side.

**Weaknesses:**

The paper seems to be hard to follow for an outsider.

**Questions:**

- at some moment it would be good to say what DR stands for (diminishing return)
- Page 3 line 127. I think you mean $\textbf{x} \vee \textbf{y} = (\max\\{x_i, y_i\\})_{i\in [d]}$
- Also here dimension is d and later it is n
- Page 3, line 150. I could not understand the difference of a bias and a noise. Is the noise consistent, in the sense that n is a deterministic function? What is the domain of $\xi$? What is known to the algorithm? Does it know the functions $b,n$ and the parameter $\xi$ or only some of them? Does the algorithm knows the assumed bounds $m,\eta_b, M, \eta_n$?
- Page 4 line 205. It wasn't clear to me before that the function x(t) is an algorithm.
- Page 5 line 228. What does it mean to maximize two values? Or do you mean to maximize the maximum of the two values?
- Page 6 line 277: which we have newly introduced -> Potential author name revealing
- Page 8 line 418 Lipschiz -> Lipschitz
- Page 10. Some acronyms should be uppercase, i.e. in curly brackets in the bibtex file. Such as DR, SGD.

---

> ### Author Response · Authors · 2025-11-20
>
> Thank you for your careful works and significant comments. Based on your suggestions, we make every effort to improve our description and answer your relevant questions.
>
> First, we correct typos in our revised version. Additionally, we fix some imprecise mathematical symbols (e.g., defining $\mathbf{x}\vee \mathbf{y} = (\max \{x_i,y_i\})_{i \in [d]}$ and correcting $n$ to $d$). Finally, we recompiled the bibtex file to resolve the issue of acronyms. We sincerely apologize for the negligence and thank you for pointing out these typos for us. The updated manuscript is available in Supplementary Material and will continue to be revised..
>
> Next, we provide some explanations and try our best to answer your questions.
>
> The concepts of bias and noise describe the gradient from different perspectives. The key distinction is that the bias is deterministic and not related to randomness, whereas noise is inherently stochastic. The bias usually arises from numerical errors (e.g., the finite-difference to approach gradients) and limitations in computational precision (e.g., representing real numbers with finite bits). Once the difference step length and the number of bits is given, the bias is determined. Conversely, the noise usually comes from sampling techniques (e.g., sampling data with a certain distribution) or probabilistic procedures (e.g., algorithms that achieve desired performance with high probability), and is normally dependent on the random variable $\xi$. Therefore, in our work, we adopt $\mathbf{b}(\mathbf{x}(t))$ to represent the bias that is determined and $\mathbf{n}(\mathbf{x}(t),\xi)$ to represent the noise that depends on the random variable $\xi$. In the context of iterative update algorithms, the upper bounds for both the bias $\mathbf{b}(\mathbf{x}(t))$ and the variance for the noise $\mathbf{n}(\mathbf{x}(t),\xi)$ are known in advance, as they are determined by the specific gradient estimation algorithm employed. For example, in Lemma 2, we provide the upper bound of the bias and the noise for the gradient estimation algorithm on classical model, i.e., $m=M=0$, $\eta_b=\frac{dL_1r}{2}$, and $\eta_n = \sqrt{16\sqrt{2\pi}dL_0^2}$. These upper bound are independent on the iterative algorithm for updating the current solution and is determined when the gradient estimation algorithm is given.
>
> Considering an iteration algorithm from the initial solution $\mathbf{x}(0)$ to the final solution $\mathbf{x}(T)$, they are a series of discrete points. The initial solution $\mathbf{x}(0)$ serves as the input of iteration algorithms, and different algorithms yield different trajectories of intermediate and final solution. In each iteration, algorithms move the current solution along a direction with a certain step length, i.e., $\mathbf{x}(t_{j+1})-\mathbf{x}(t_j) = \alpha(t_j) \mathbf{v}(t_j)$. Potentially, this difference equation can be viewed as a numerical discretization for a continuous dynamic system, i.e., $\dot{\mathbf{x}}(t) = \frac{d \mathbf{x}(t)}{d t}=\tilde{\mathbf{v}}(t)$. In these interpretations, the curve $\mathbf{x}(t)$ is corresponds with the trajectory of an algorithm when $\tilde{\mathbf{v}}(t)$ and $\mathbf{x}(0)$ are given. Different discretization methods cause different positions for all intermediate solutions. And the numerical errors from the continuous procedure to the discrete procedure directly implies the iteration complexity. Therefore, we adopt a continuous curve $\mathbf{x}(t)$ to conceptually represent an algorithm for continuous times.
>
> On Page 5 (Line 228), our target is simultaneously to optimize two values. The first value is the approximation ratio. The second value is the errors from stochastic biased gradients. To formulate this as a unified maximization problem, we introduce a negative sign before the errors term.
>
> Thank you for your details works and pointing out the concern about Page 6 (Line 277). According to the context "as a demonstration, our main text focuses on convex constraints with the largest element, which we have newly introduced", our intention was simply to illustrate our methodology using the new setting considered in this work as an example for illustration. No relevant information about the author has been disclosed here. To eliminate this ambiguity, we have polished the description of this sentence.

---

### Official Review · Reviewer_VyE8 · 2025-10-31

**Soundness:** 3
**Presentation:** 3
**Contribution:** 3
**Rating:** 6
**Confidence:** 2

**Summary:**

This paper studies continuous DR-submodular maximization under stochastic biased gradient oracles. It extends Du's Lyapunov framework to handle bias and variance in gradient estimators. Based on this, it provides approximation algorithms under three constraint classes: general convex, down-closed convex, and convex sets with a largest element. The paper develops zeroth-order algorithms, where the classical version achieves $O(\epsilon^{-3})$ while the quantum version achieves $O(\epsilon^{-1})$ iteration complexity, matching the performance of classical first-order methods.

**Strengths:**

- Extending Du's Lyapunov framework to stochastic biased gradients seems interesting. The framework explicitly characterize the effect of bias and noise, and the resulting analysis looks useful beyond this problem setting.
- The constraint set given by a convex set with a largest element is well-motivated. Bridging the convex and down-closed settings leads to a provable $1/e$ guarantee.
- The paper shows that quantum gradient estimation can close the gap to first-order methods in ratio and complexity.

**Weaknesses:**

- The experiments are limited to $d=3$ due to simulating quantum algorithms in classical computers. To improve empirical evidence, larger-scale tests would make the claim of quantum acceleration more convincing.

**Questions:**

- How tight is the $1/e$ approximation for convex sets with a largest element?
- Would it be possible to provide alternative constructions for $a(t)$ and $b(t)4 that possibly improve the current setup?

---

> ### Author Response · Authors · 2025-11-20
>
> Thank you for your diligent reviews and helpful comments. Following your insightful comments, we exert our best efforts to revise the manuscript and address your question.
>
> Technically, the hardness result for non-monotone DR-submodular maximization over convex sets with a largest element is an interesting open problem that we investigate further in the future. At this stage, we only have preliminary intuitions. In Qi (2024), it was shown that the approximation ratio for maximizing non-monotone submodular set functions under a matroid/cardinality constraint cannot exceed 0.478. It implies that the hardness result of non-monotone DR-submodular maximization under down-closed convex sets is also bounded above by 0.478 because the multilinear extension of submodular set functions is a special DR-submodular function and the matroid constraint is a special down-closed convex constraint. Additionally, the down-closed convex constraint is a special convex set with a smallest element, which in some sense is symmetric to convex constraints with a largest element. It gives us an intuition: the hardness result for non-monotone DR-submodular maximization under convex sets with a largest element is bounded above by 0.478. In addition, inspired by approximation algorithms for non-monotone DR-submodular maximization with down-closed convex sets (0.385 approximation (Chen et al., 2023) and 0.401 approximation (Buchbinder and Feldman, 2024)), we prefer to believe that the approximation ratio for this problem can be further improved.
>
> Considering the Lyapunov function $E(t)= a(t) F(\mathbf{x}(t))-b(t)F(\mathbf{x}^*)$, different $a(t)$ and $b(t)$ potentially implies different approximation ratios. The following table illustrates some concrete choices for $a(t),b(t)$ with corresponding approximation ratios.
>
> | $a(t)$| $b(t)$ | Func/Cons | Approx Ratio |
> | :- | :- | :-: | :-: |
> | $a(t)=e^{t}$| $b(t)=e^{t}$ | Monotone/Convex| $1-e^{-1}$|
> | $a(t)=(t+1)^2$| $b(t)=\|\|\mathbf{1}-\mathbf{x}(0)\|\|_{\infty} t$ | Non-Monotone/ Convex| $\frac{\|\|\mathbf{1}-\mathbf{x}(0)\|\|_{\infty}}{4}$|
> | $a(t)=e^{t}$| $b(t)=t$ | Non-Monotone/Down-closed Convex| $e^{-1}$|
> | $a(t)=e^t$| $b(t)=\|\|\mathbf{1}-\mathbf{x}(0)\|\|_{\infty} t$ | Non-Monotone/ Convex + a Largest Element| $\frac{\|\|\mathbf{1}-\mathbf{x}(0)\|\|_{\infty}}{e}$|
>
> If tighter upper bounds for the optimal solution can be constructed, it potentially induces alternative choices for $a(t)$ and $b(t)$, which in turn yields improved approximation algorithms. However, as discussed in Section 3.2, under the current upper bound of $(1-\theta(t)) F(\mathbf{x}^*) \le F(\mathbf{x}(t)) +<\nabla F(\mathbf{x}(t)), \mathbf{x}^* \vee \mathbf{x}(t)-\mathbf{x}(t) >$, different $a(t)$ and $b(t)$ cannot improve the approximation ratio.
>
> For experiments, simulating quantum algorithms on classical computers is normally expensive for space and time. Although it is challenging to simulate quantum algorithms on classical computers for large-scale datasets, we attempt to illustrate the potential quantum acceleration as follows:
> 1. Quantum gates with the same circuit depth can be executed in parallel.
> 2. In each depth, the time-consuming is approximately 100ns on current superconducting platforms.
> 3. As a toy example, we consider the quadratic function $F(\mathbf{x})=\mathbf{x}^T \mathbf{H} \mathbf{x}+\mathbf{h}^T \mathbf{x}+c$ as the function value oracle. The quantum circuit depth for this oracle is about $O(d)$, where $d$ is the dimension of the function $F$.
> 4. For convex constraints with a largest element, the quantum circuits depth for phase estimation is $O(\log \frac{24L_0 d}{\epsilon_{ALG}} \log ( \frac{24L_0 d^2}{\epsilon_{ALG}}\log \frac{24L_0 d}{\epsilon_{ALG}}))$.
> 5. Totally, the quantum circuit depth is approximately $O( d + \log \frac{24L_0 d}{\epsilon_{ALG}} \log ( \frac{24L_0 d^2}{\epsilon_{ALG}}\log \frac{24L_0 d}{\epsilon_{ALG}}))$ assuming that the copies of $|\psi\rangle$ can be prepared simultaneously across multiple quantum computers.
>
> Consequently, we can roughly estimate the time-consuming for the quantum gradient estimation algorithm, as summarized in the table below. The revised manuscript is in Supplementary Material and will continue to be revised.
>
> | Dimensions | Errors | Depth | Qubits | Clock Time| Dimension of Vector|
> | :- | :- | :-: | :-: | :-: | :-: |
> | $d=10$| $\epsilon_{ALG}=0.1$ | $\approx 213$| $\approx 130$| $\approx 21 \mu s$| $\approx 2^{130}$|
> | | $\epsilon_{ALG}=0.01$ |$\approx 404$ | $\approx 190$| $\approx 40 \mu s$ | $\approx 2^{190}$|
> | | $\epsilon_{ALG}=0.001$ |$\approx 553$ | $\approx 230$| $\approx 55 \mu s$ | $\approx 2^{230}$|
> | $d=100$| $\epsilon_{ALG}=0.1$ | $\approx 612$| $\approx 1600$| $\approx 61 \mu s$| $\approx 2^{1600}$|
> | | $\epsilon_{ALG}=0.01$ |$\approx 896$ | $\approx 2300$| $\approx 89 \mu s$ | $\approx 2^{2300}$|
> | | $\epsilon_{ALG}=0.001$ | $\approx 1102$| $\approx 2600$| $\approx 110 \mu s$ | $\approx 2^{2600}$|

---

> > ### Comment · Reviewer_VyE8 · 2025-11-26
> >
> > Thank you for the detailed response to my comment and questions. That said, I have increase the confidence level, and I will keep my positive review.

---

### Official Review · Reviewer_AyjJ · 2025-11-05

**Soundness:** 3
**Presentation:** 2
**Contribution:** 3
**Rating:** 6
**Confidence:** 3

**Summary:**

This paper investigates the problem of continuous DR-submodular maximization under the practically relevant yet theoretically challenging setting of stochastic biased gradients. The authors extend the Lyapunov framework, traditionally developed for exact or unbiased stochastic gradients, to handle gradient estimators that contain both bias and noise, thereby characterizing their effects on convergence and approximation guarantees. They further introduce a new class of constraints, namely convex sets with a largest element, that naturally arise in resource allocation and similar applications. Under this setting, the paper proposes a $1/e$-approximation algorithm for non-monotone DR-submodular maximization, which surpasses the known $1/4$ hardness bound for general convex sets. Building upon this framework, the authors design both classical and quantum zero-order algorithms, showing that the quantum version achieves the same approximation ratio with only $O(\varepsilon^{-1})$ iteration complexity, demonstrating a quantum acceleration compared with classical zero-order methods that require $O(\varepsilon^{-3})$. Numerical experiments on quadratic and coverage-type DR-submodular functions validate the theoretical results, showing that quantum algorithms converge faster and achieve comparable solution quality to classical first-order methods. Overall, the paper provides a unified theoretical and algorithmic treatment of DR-submodular maximization in the presence of biased gradients and connects classical optimization with emerging quantum techniques.

**Strengths:**

The paper extends the Lyapunov-based analytical framework to accommodate stochastic biased gradients, a setting that more faithfully represents real-world learning and optimization scenarios where gradient estimates are noisy and biased. The work also successfully integrates quantum computation into continuous submodular optimization, showing that quantum zero-order algorithms can match the convergence rate of classical first-order methods, offering a clear demonstration of quantum speedup. The results are rigorously proven, the methodology is well grounded in prior literature, and the experimental findings, though small in scale, corroborate the theoretical analysis.

**Weaknesses:**

The paper currently lacks experimental results on runtime performance, which makes it difficult to assess the practical efficiency of the proposed algorithms.

Moreover, it would be beneficial if the authors could provide additional real-world examples or application scenarios to better justify the practical relevance of the studied problem, since DR-submodular maximization has so far attracted more attention from the theoretical community rather than from most of the ICLR audience.

Minor comment: In Figure 1, the legends in the third and sixth subplots appear partially covered by light-colored text, suggesting a plotting or rendering issue that should be corrected for clarity.

**Questions:**

Please refer to the weakness.

---

> ### Author Response · Authors · 2025-11-20
>
> Thank you for your patient works and insightful comments. Following your suggestions, we make every attempt to address your questions and improve our manuscript.
>
> First, we explain why we did not include a comparison of the running time performance in the submitted manuscript.
> 1. Classical simulation of quantum algorithms could be large overhead. Each qubit corresponds to a 2-dimensional vector in Hilbert space($\mathbb{C}^2$), so simulating $n$ qubits quantum algorithms essentially involves operations on $2^n$-dimensional vectors. If $n$ is relatively large, storing these vectors is space-expensive and executing operations for $2^n$-dimensional vectors is time-expensive.
> 2. Algorithms in our works are challenging to implement on current quantum hardware but hold optimistic in the future. On the Noisy Intermediate-Scale Quantum (NISQ), although thousands of physical qubits may be available, the number of computational qubits remains limited due to noises in single/double qubit gates. Their fidelity is currently far lower than that of gates in classical computer. With the development of quantum hardware and error correction technologies, we are optimistic for implementing our algorithms on quantum hardware in the future.
> 3. Cross-model computational comparisons may be partially unfair at this stage. Because simulating quantum algorithms on classical computers is intrinsically time-expensive, comparing the time-consuming of quantum/classical algorithms on classical computers is unfair for quantum algorithms. Conversely, comparing the time-consuming between quantum algorithms on quantum hardware with classical algorithms on CPU/GPU, it seems to be a competition between athletes and amateur players and is unfair for classical algorithms. A more equitable comparison in the future may involve the SOTA classical algorithm on the best classical computers versus the SOTA quantum algorithm on the best quantum computer for the same problem.
>
> Next, we illustrate the cost of simulating our quantum algorithms on classical computers through specific examples and numerical values. Leveraging Lemma 7 in Appendix C.1 and the algorithm procedure in Appendix B, the depth of quantum circuits and the number of qubits in each iteration for the convex set with a largest element is as follows:
> - Depth: $O(\log \frac{24L_0 d}{\epsilon_{ALG}} \log ( \frac{24L_0 d^2}{\epsilon_{ALG}}\log \frac{24L_0 d}{\epsilon_{ALG}}))$
> - Qubits: $O(bd)= O(d \lceil \log \frac{48L_0 d}{\epsilon_{ALG}} \rceil)$
> - Dimension of Vectors: $2^{O(bd)}$
>
> When $d=3$, $1/\epsilon_{ALG}=10$, and $L_0=1$, the number of qubits is approximately $33$, corresponding to a vector dimension of $2^{33}$. This is already computationally expensive for classical computers. In real-world examples, the dimension of the function is typical much larger than $d=3$. Numerical results show that simulating quantum algorithms with $d=100$ exceeds the capabilities of some classical computers. We apologize for not being able to simulate real datasets. However, we can roughly describe the clock time of our quantum gradient estimation algorithms as follows:
> 1. Single/double qubit gates operate approximately 50ns/100ns on current superconducting platforms. Quantum gates at the same circuit depth can be executed simultaneously because the depth for quantum gate essentially reflects the order for executing. That implies each depth in quantum circuits can be executed about 100ns.
> 2. Preparing quantum states for the input of gradient estimation algorithms depends on the depth of quantum circuit for the value oracle $F$ that is case by case and cannot be uniformly characterized. Thus, we only consider the runtime of phase estimations, whose depth only depends on the dimension $d$ of the function $F$ and the errors $\epsilon_{QG}$ of estimations.
> 3. Clock time: $ \approx$ Depth $\times$ 100ns($10^{-7}$ s). When $L_0=d=1/\epsilon_{ALG}=100$, the magnitude of the clock time is about 100$\mu$s. Much details on clock time are in table.
>
> At last, in our revised manuscript, we include the analysis about the clock time and redraw Fig.1 in the main text to make it clearer. The revised manuscript is in Supplementary Material and will continue to be revised.
>
> | Dimensions | Errors | Depth | Qubits | Clock Time| Dimension of Vector|
> | :- | :- | :-: | :-: | :-: | :-: |
> | $d=10$| $\epsilon_{ALG}=0.1$ | $\approx 203$| $\approx 130$| $\approx 20 \mu s$| $\approx 2^{130}$|
> | | $\epsilon_{ALG}=0.01$ |$\approx 394$ | $\approx 190$| $\approx 39 \mu s$ | $\approx 2^{190}$|
> | | $\epsilon_{ALG}=0.001$ |$\approx 543$ | $\approx 230$| $\approx 54 \mu s$ | $\approx 2^{230}$|
> | $d=100$| $\epsilon_{ALG}=0.1$ | $\approx 512$| $\approx 1600$| $\approx 51 \mu s$| $\approx 2^{1600}$|
> | | $\epsilon_{ALG}=0.01$ |$\approx 796$ | $\approx 2300$| $\approx 79 \mu s$ | $\approx 2^{2300}$|
> | | $\epsilon_{ALG}=0.001$ | $\approx 1002$| $\approx 2600$| $\approx 100 \mu s$ | $\approx 2^{2600}$|

---

> > ### Comment · Reviewer_AyjJ · 2025-11-24
> >
> > Thank you for your rebuttal. After carefully reviewing your response and considering the feedback from other reviewers, I have decided to maintain my original score.

---

### Official Review · Reviewer_bbTg · 2025-11-11

**Soundness:** 3
**Presentation:** 2
**Contribution:** 2
**Rating:** 6
**Confidence:** 2

**Summary:**

The authors consider the problem of DR-submodular maximization.  They consider a few combinations of monotone/non-monotone functions over the hypercube and classes of convex constraints (general, down-closed, largest element).  The authors first propose an extension of a Lyapunov framework from exact to stochastic and biased gradients.  The authors consider a new constraint setting (largest element) for the non-monotone setting and obtain an improved approximation ratio (over using a general convex region based method).  The authors also show significant improvements in complexity for the value oracle setting using a quantum algorithm for gradient estimation.  Lastly, the authors run several experiments to demonstrate the improvements of the quantum based method.

**Strengths:**

- The authors present a new approximation ratio for the case of non-monotone DR-submodular maximization over convex constraints with a largest element.
- The authors show that there is a notable convergence speedup using quantum algorithms for gradient estimation (for the value oracle setting).
- The authors extend the Lyapunov framework for DR-submodular maximization to handle stochastic and biased oracles.
- The authors include experiments that show improved convergence for quantum algorithms in the value oracle setting.

**Weaknesses:**

- I had some uncertainty about the extent of technical challenges and novelty for some parts
    - For the Lyapunov extension to handle stochastic biased gradients, it was unclear to me from the description in the main section what technical challenges were encountered compared to past works.   Could the authors summarize the challenges?
        - in line 047 (Du, 2022) is cited as having a unifying Lyapunov framework unifying many previous methods.  That reference is not brought up in Section 3 for the framework, though the authors are upfront that they generalize a previous framework to handle stochastic (and biased) gradients.
        - Du’s work applied the Lyapunov framework for DR-submodular, and though that work presumed exact oracles, for convex optimization at least has there been Lyapunov based approaches that handled stochastic and biased (or at least stochastic) gradients? If so, are there unique challenges in extending the Lyapunov framework for DR-submodular problems from exact to stochastic (and biased) gradients?
    - For the quantum section, the authors adopt a quantum algorithm for gradient estimation for DR-submodular maximization similar to past works in convex optimization.  In light of past works (both from classic gradient estimation in DR-submodular works and wrt Augustino et al 2025 for quantum methods in convex optimization), were there some key steps that were particularly challenging?


### Minor
- Table 1 I’d suggest listing DR-max approx. bounds (and complexities) for stochastic first order based methods for reference.

- Non-monotone max with largest element is new setting considered, so appx ratio is first, but not clear  how tight it may be (no lower bound)

- Theorem statements referencing algorithms were imprecise,
    - eg line 360 “returned by quantum algorithms satisfies” – do the bounds in Theorem 2 hold for any quantum algorithms, or the (single) specific algorithm adopted from van Apeldoorn et al 2023?
    - Theorem 3 line 377 “there are some quantum zero-order algorithms achieving”
- Fig 1 the axes’ fonts are too small

### Very Minor
- “Lipschitz” not “Lipschiz”
- line 407 “with [A]ssumption[s] 1 …”
- line 366 “The query complexity of the value oracle” should that be the complexity of the algorithm?

**Questions:**

- Do the complexity bounds depend on the dimension?
- I found Section 4 on quantum acceleration hard to follow.  I am not familiar with quantum methods.  I have some uncertainty about the specific set up and some uncertainty about whether the impressive complexity results using quantum algorithms are purely of theoretic interest or if in the (near) future there could be the potential for real-world use.
    - From line 354, just to confirm, the set up is identical in terms of the environment (the (biased) value oracle)?  A learner that has access to a quantum computer can use quantum Jordan algorithms to achieve the speedup in terms of query complexity over a learner that only has access to classical computers, but the environment itself and how they interact with the environment is identical?
    - Could the authors remark on example situations where there could be a practical benefit in terms of total run-time?  Eg Figure 1 is measured in terms of iterations.  How would that map to clock time?  I understand that in the experiments the authors were simulating a quantum algorithm on a classical computer, so there could be large overhead.
        - For readers unaware of how much overhead, just looking at Fig 1 results it might be tempting to consider using a simulated quantum algorithm even for a smaller number of iterations if the overhead is low enough.  What were the (rough) run-times?
        - Would the authors be familiar enough with current quantum computers on the market if they would be big enough to be used for this type of problem? If not, is there an educated guess for how close quantum computing is to the scale even to be used for the small $d=3$ experiments here?
        - For current/near future quantum systems, is there a rough sense for how long each iteration (for the quantum Jordan algorithm) might take in clock time?

---

> ### Author Response · Authors · 2025-11-20
>
> Thanks for your valuable comments. Leveraging your comments, we try our best to answer your questions and improve our work.
>
> Compared to previous works, the main technical challenges in extending the Lyapunov framework are as follows:
> 1. From exact gradient oracles to stochastic biased gradient oracles for DR-submodular maximization: how to integrate the algorithm design target (approximation ratios) and the effects of bias and variance from gradient oracles in the Lyapunov framework.
> 2. From convex optimization to DR-submodular maximization with stochastic gradients: how to choose a proper Lyapunov function and describe the upper bound of optimal solutions based on stochastic biased gradients.
>
> DR-submodular maximization is NP-hard such that optimal solutions with numerical errors are unavailable in polynomial time. Its Lyapunov function relies on some parameter functions (e.g. $a(t),b(t)$). By Deriving from the monotone Lyapunov function and discretizing it, we obtain the approximation ratio (variational problems) and the convergence rate from parameter functions. Different from convex optimization, tight upper bounds of the optimal solution induce algorithms with large approximation ratios (e.g. better $a(t),b(t)$), while the bias/variance of gradient couples with parameter functions to impact the convergence. Depicting these relationships in the Lyapunov framework clearly is important, yet this has not been achieved in previous works.
>
> For the computational complexity of algorithms, there are many perspectives:
> 1. Query Complexity: for 0-order algorithms (estimating gradients via function values), this complexity illustrates how many times the function value needs to be computed and is directly related to the dimension. For example, in Theorem 2, the query complexity for each quantum gradient estimation is $O(8\lceil \ln \frac{36L_0d}{\sigma}\rceil)$ where $d$ is the dimension.
> 2. Iteration Complexity: this complexity illustrates how many iterations for the algorithms. The iteration complexity for algorithms proposed in our work is independent on the dimension. This is also the complexity that we mainly consider.
>
> The upper bound of approximation ratios for convex constraints with a largest element is an interesting open problem we will pursue. Inspired by the hardness result 0.478 for down-closed convex constraints, we conjecture that the hardness result of convex constraints with a largest element is $\le$ 0.478 (the down-closed convex constraint is a special case for convex constraints with a smallest element, which is symmetric to convex constraints with a largest element to some extent).
>
> Considering experiments, we map the runtime into the clock time as below:
> 1. Single/double qubit gates are approximately 50ns/100ns on current superconducting platforms. Gates of the same-depth can be executed in parallel. Therefore, each depth in quantum circuits can be executed about 100ns.
> 2. We only consider the runtime of phase estimations (depth only depends on the dimension $d$ and the estimation errors $\epsilon_{QG}$). This is because quantum state preparations for gradient estimation algorithms varies with the value oracle $F$, leading to the depth of quantum circuit is case by case and hard-to-describe uniformly.
>
> For the convex set with a largest element(a toy example), leveraging Lemma 7 in Appendix C.1 and the algorithm procedure in Appendix B, the depth of quantum circuits and the number of qubits in each iteration is as below:
> - Depth: $O(\log \frac{24L_0 d}{\epsilon_{ALG}} \log ( \frac{24L_0 d^2}{\epsilon_{ALG}}\log \frac{24L_0 d}{\epsilon_{ALG}}))$
> - Clock Time: $ \approx$ Depth $\times$ 100ns. Setting $L_0=d=1/\epsilon_{ALG}=100$, the magnitude of the clock time is about 100$\mu$s.
> - Qubits: $O(bd)= O(d \lceil \log \frac{48L_0 d}{\epsilon_{ALG}} \rceil)$. Because each qubit is a vector in $\mathbb{C}^2$, adopting classical computers to simulate quantum algorithms relies on $2^{O(bd)}$ dimensional vector operations. When $d=3$, $1/\epsilon_{ALG}=10$, and $L_0=1$, the number of qubits is about 33, which exceeds the number of computational qubits (rather than physical qubits) currently available and is computationally expensive for classical simulations because of the $2^{33}$ dimension vector. Therefore, we choose $d=3$ in numerical experiments.
>
> At last, we correct typos in the revised version and thanks for pointing them out. We fully agree with your suggestions and list related positive results (stochastic 1-order methods) and hardness results (along with their tightness) in Tab.1, accompanied by corresponding references. We refine statements of Theorem 2 and 3 for precision: the quantum algorithm mentioned in Theorem 2 builds on van Apeldoorn et al. 2023 with recalculated parameters; algorithms in Theorem 3 are quantum 0-order algorithms proposed in our works. Also, we redraw Fig.1 to improve clarity. The revised manuscript is in Supplementary Material and will continue to be revised.

---

### Author Response · Authors · 2025-12-03

Dear Area Chair,

Below is a summary of the original reviews and corresponding responses, provided to assist in your assessment. We are grateful to both the original and newly assigned Area Chairs for their dedicated efforts, and to reviewers for their valuable comments.

All reviewers initially provided positive scores **(6,6,6,6)** before the discussion stage. During the rebuttal period, we addressed the reviewers' questions, clarified their concerns, and received updated feedback (reviewers AyjJ,VyE8 keep their positive reviews and scores). The main strengths of our works and our key responses are summarized as follows:

**Main Strengths:**
1. Present a new approximation ratio $1/e$ for the case of non-monotone DR-submodular maximization over convex constraints with a largest elements. Prior to this work, there was a lack of approximation algorithms under this constraint (Although the $1/4$ approximation algorithm for convex constraints can also be adopted for this constraint, this algorithm does not utilize the information for the largest element).
2. Extend the Lyapunov framework from exact gradient oracles to stochastic biased gradient oracles, enabling the design and analysis for approximation algorithm on DR-submodular maximization in settings that more faithfully represent real-world learning and optimization scenarios with noisy and biased gradient estimates.
3. Integrate the quantum gradient estimation algorithm into DR-submodular maximization, showing that quantum zero-order algorithms can achieve the convergence rate comparable to classical first-order methods, offering a clear demonstration of quantum speedup.
4. Combine theoretical and practical validation through rigorously proofs and numerical experiments.

**Main Responses:**
1. **Reviewers (bbTg,AyjJ,VyE8) suggest that we report the running time (the clock time) for our algorithms and test them on larger-scale functions.**

  First, we explain why running comparisons were not include in the submitted manuscript, considering both fairness of comparison and current limitations of quantum hardware. In response, we have provided a rough characterization of clock time and discussed the overhead for simulating quantum algorithms on classical computers. With the development of quantum hardware, our algorithms can be implemented in the future. Detailed descriptions can be founded in our responses to the individual reviewers.

2. **Reviewers (bbTg, VyE8) inquire about the tightness $1/e$ for the case of non-monotone DR-submodular maximization over convex constraints with a largest elements. And Reviewer (VyE8) want to know whether alternative constructions for $a(t),b(t)$ imply different approximation ratios.**

  The hardness result for convex constraints with a largest element is an interesting open problem we will pursue. Although we currently do not have a precise characterization, we offer a roughly intuition and conjecture from the hardness result on DR-submodular maximization over down-closed convex constraints. Additionally, in response to the Reviewer’s (VyE8) question regarding $a(t), b(t)$, we also illustrate that different $a(t)$ and $b(t)$ potentially implies different approximation ratios when considering the Lyapunov function $E(t)= a(t) F(\mathbf{x}(t))-b(t)F(\mathbf{x}^*)$. Detailed descriptions can be founded in our responses to the individual reviewers.

3. **Reviewers (bbTg, 3kKo) gently point out some typos, confused notations and descriptions.**

  We correct typos and clarify notational confusions. In addition, we list related positive results (stochastic 1-order methods) and hardness results in Tab.1 and refine statements of Theorem 2 and 3 for precision. Also, we redraw Fig.1 to improve clarity. The revised manuscript is included in Supplementary Material. Detailed descriptions can be founded in our responses to the individual reviewers.


Thank you once again for your diligent works.

Best regards,

---

### Meta-Review · Area_Chair_eGWt · 2026-01-06

**Summary:**

The reviews are mostly positive. The rebuttal of the authors have successfully addressed the reviewers' comments, such as the technical challenges, run time comparison, larger d for quantum experiments.

**Reviewer Concerns:**

Most comments have been addressed.

**Reviewer Scores:**

I think they will maintain their scores, which are already positive.

---

### Decision · Program_Chairs · 2026-01-26

Accept (Poster)